# CheapNet: Cross-attention on Hierarchical representations for Efficient protein-ligand binding Affinity Prediction

**Hyukjun Lim**[1]**, Sun Kim**[2,3,4,5]**, Sangseon Lee**[6] *

[1]Department of Materials Science and Engineering, Seoul National University
[2]Department of Computer Science and Engineering, Seoul National University
[3]Interdisciplinary Program in Bioinformatics, Seoul National University
[4]Interdisciplinary Program in Artificial Intelligence, Seoul National University
[5]AIGENDRUG Co., Ltd., Seoul, Republic of Korea
[6]Department of Artificial Intelligence, Inha University
{hyukjunlim, sunkim.bioinfo}@snu.ac.kr, ss.lee@inha.ac.kr

## Abstract

Accurately predicting protein-ligand binding affinity is a critical challenge in drug discovery, crucial for understanding drug efficacy. While existing models typically rely on atom-level interactions, they often fail to capture the complex, higher-order interactions, resulting in noise and computational inefficiency. Transitioning to modeling these interactions at the cluster level is challenging because it is difficult to determine which atoms form meaningful clusters that drive the protein-ligand interactions. To address this, we propose CheapNet, a novel interaction-based model that integrates atom-level representations with hierarchical cluster-level interactions through a cross-attention mechanism. By employing differentiable pooling of atom-level embeddings, CheapNet efficiently captures essential higher-order molecular representations crucial for accurate binding predictions. Extensive evaluations demonstrate that CheapNet not only achieves state-of-the-art performance across multiple binding affinity prediction tasks but also maintains prediction accuracy with reasonable computational efficiency. The code of CheapNet is available at https://github.com/hyukjunlim/CheapNet.

## 1 Introduction

Predicting protein-ligand binding affinity—the quantitative measure of interaction strength between a protein and a ligand—is a fundamental challenge in drug discovery with major implications for therapeutic development. This measure, often expressed as the dissociation constant ($K_d$) or inhibition constant ($K_i$), directly determines drug efficacy. Traditional wet-lab methods, though accurate, are time-consuming, costly, and difficult to scale (Schirle & Jenkins, 2016; Lee & Lee, 2016; Yang et al., 2022), necessitating the development of computational approaches as faster, scalable alternatives in the drug discovery pipeline. However, computational modeling of binding affinity remains highly challenging due to the intricate and variable nature of molecular interactions, presenting significant hurdles for deep learning approaches (Dhakal et al., 2022).

Recent advances in deep learning have shown promise in predicting binding affinity by learning atom-level representations of proteins and ligands (Öztürk et al., 2018; Yang et al., 2022; Jiang et al., 2021; Townshend et al., 2020; Yang et al., 2023; Feng et al., 2024), modeling their interactions as sets of atom-to-atom relationships. While this atom-centric approach captures fine-grained details of local interactions, it has notable limitations. Modeling solely at the atom level results in excessive computational complexity, as many atom pairs contribute negligibly to overall binding affinity (Nguyen et al., 2023; Tan et al., 2024; Abdelkader et al., 2023). Moreover, treating all atoms equally introduces noise, as irrelevant atoms can interfere with accurate predictions (Jin et al., 2023; Shen et al., 2024).

---

*Corresponding author.

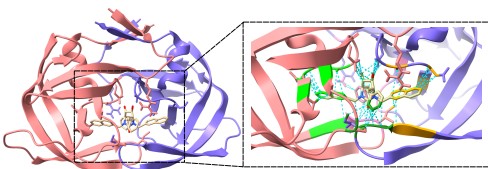

Beyond the limitations of atom-level modeling, binding mechanisms often involve hierarchical relationships that atom-level approaches alone cannot fully capture. Clusters of atoms often interact collectively with specific protein regions, such as aromatic rings targeting binding pockets to inhibit HIV protease (see Figure 1). These clusters exemplify the importance of identifying groups of atoms that act synergistically, a key factor in established binding

Figure 1: **Protein-ligand complex (PDB ID: 1HVR) of HIV protease and its inhibitor.** The ligand's aromatic ring and its corresponding contact region on the protein are highlighted in matching colors. Cyan dashed lines represent all contact points between the ligand and protein.

paradigms like the lock-and-key and induced fit models (Du et al., 2016; Zhao et al., 2021; Fatmi et al., 2009; Schatz et al., 2021; Liu et al., 2017a). The primary challenge lies in developing a mechanism that identifies meaningful clusters dynamically and ensures their relevance to the binding process.

Effectively addressing this challenge involves learning subgraphs—substructures within the protein-ligand complex—that encode both local and global structural features (Yuan & Ji, 2020). Unlike traditional methods that focus exclusively on atom-level interactions or predefined clusters with geometric constraints (Du et al., 2024; Kong et al., 2024), this requires a model capable of adaptively identifying relevant clusters based on their contributions to binding interactions. Such a model should aim to learn these clusters through end-to-end training, capturing both local interactions and their broader structural context to provide a comprehensive understanding of protein-ligand binding.

To address these limitations, we propose CheapNet, a novel interaction-based model that dynamically identifies cluster-level representations of protein-ligand complexes through end-to-end training. By leveraging a differentiable pooling mechanism, CheapNet aggregates atom-level embeddings into higher-level clusters, reducing noise and computational complexity while focusing on groups of atoms that contribute significantly to binding interactions. Next, a cross-attention mechanism is applied between protein and ligand clusters, enabling the model to focus on the most relevant inter-molecular interactions, thereby improving prediction accuracy and computational efficiency. In summary, the key contributions of CheapNet are as follows:

- We propose a hierarchical model that integrates atom-level and cluster-level interactions, improving the representation of protein-ligand complexes.

- Our model incorporates a cross-attention mechanism between protein and ligand clusters, focusing on relevant binding interactions in the cluster-level.

- CheapNet achieves state-of-the-art performance across multiple binding affinity prediction tasks while maintaining computational efficiency.

## 2 RELATED WORKS

Protein-ligand binding affinity prediction has traditionally focused on atom-level approaches. Recently, cluster-level frameworks have emerged, emphasizing the importance of capturing higher-level interactions. Additional details on representative methods are provided in the Appendix A.1.

### 2.1 ATOM-LEVEL PROTEIN-LIGAND BINDING AFFINITY PREDICTION

Atom-level approaches to protein-ligand binding affinity prediction are categorized as interaction-free or interaction-based. Interaction-free models, while computationally efficient, treat proteins and ligands independently, failing to capture critical interdependent interactions (Öztürk et al., 2018; Nguyen et al., 2021; Yang et al., 2021; Rifaioglu et al., 2021; Huang et al., 2021; Yang et al., 2022; Yuan et al., 2022). Interaction-based models address this by modeling atomic-level relationships using 3D structural data (Townshend et al., 2020; Jiang et al., 2021; Yazdani-Jahromi et al., 2022; Yang et al., 2023; Wang et al., 2023; Nguyen et al., 2023; Feng et al., 2024), but they often overlook hierarchical mechanisms, such as group-level or cluster-level interactions. Our model fills this gap by integrating a cluster-attention mechanism, capturing interactions at both atom and cluster levels for a more comprehensive representation.

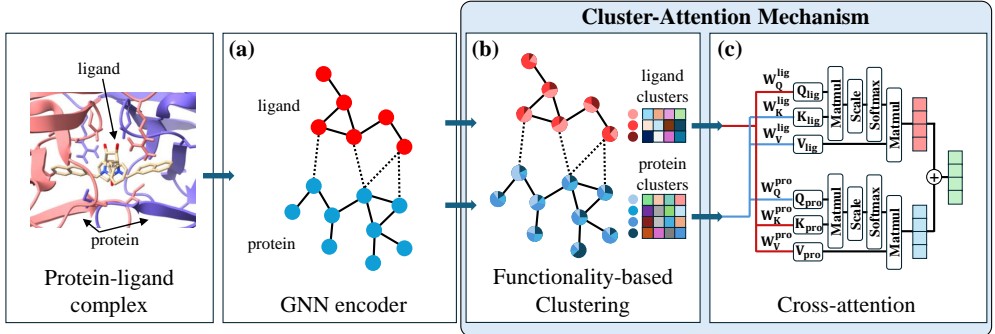

Figure 2: **Architecture of CheapNet for protein-ligand binding affinity prediction.** (a) A graph encoder learns atom-level embeddings of the protein-ligand complex. (b) A differentiable pooling mechanism clusters the embeddings into cluster-level representations. (c) A cross-attention mechanism is applied between the protein and ligand clusters to capture key interactions.

## 2.2 CLUSTER-LEVEL PROTEIN-LIGAND BINDING AFFINITY PREDICTION

Recent studies, including GemNet (Gasteiger et al., 2021), Equiformer (Liao & Smidt, 2022), LEFT-Net (Du et al., 2024), and GET (Kong et al., 2024), have leveraged geometric and hierarchical representations to enhance protein-ligand binding affinity prediction. GemNet and Equiformer focus on local and global molecular interactions using geometric and equivariant features. LEFTNet and GET build on this by incorporating hierarchical frameworks that integrate block-level and atomic details. However, these models often depend on predefined clusters or geometric constraints. CheapNet addresses these limitations with a data-driven approach, utilizing soft clustering of atoms based on learned embeddings and cross-attention to dynamically model diverse protein-ligand interactions.

## 3 METHODS

In this section, we present the architecture of CheapNet, a model designed for protein-ligand binding affinity prediction. CheapNet first employs a graph encoder to learn atom-level embeddings of the protein-ligand complex (Figure 2(a)). Subsequently, a differentiable pooling mechanism is used to aggregate atom-level embeddings into cluster-level representation (Figure 2(b)). Next, a cross-attention mechanism is introduced between the protein and ligand clusters, allowing the model to focus on the most relevant interactions for binding affinity prediction (Figure 2(c)).

### 3.1 PROBLEM DEFINITION

In this study, we aim to predict the binding affinity of protein-ligand complexes. Each complex is represented as a graph $\mathcal{G} = (V, E) = (V_l \cup V_p, E_l \cup E_p \cup E_{lp})$, where $V_l$ and $V_p$ denote the set of nodes corresponding to atoms in the ligand and protein, respectively. Each node $v_i \in V$ is associated with a feature vector $\boldsymbol{x}_i \in \mathbb{R}^d$, representing atomic properties (which may vary across datasets), and a 3D coordinate $\boldsymbol{r}_i \in \mathbb{R}^3$. The edge sets $E_l$ and $E_p$ represent intra-molecular covalent bonds within the ligand and protein, while $E_{lp}$ denotes inter-molecular, non-covalent interactions between ligand and protein atoms within a distance of 5Å. The target variable, $y \in \mathbb{R}$, represents the binding affinity of the complex, expressed as $-\log(K_d)$ or $-\log(K_i)$, where $K_d$ and $K_i$ are the dissociation and inhibition constants, respectively. The objective is to train a predictive model $f$ that estimates the binding affinity $\hat{y} = f(\mathcal{G})$ by minimizing the error between $\hat{y}$ and the true affinity $y$.

### 3.2 ATOM-LEVEL EMBEDDING VIA GRAPH ENCODING

Before clustering the atoms, we first update the embeddings of the protein and ligand nodes to capture both local atomic properties and interactions within the protein-ligand complex. For each node $v_i \in V$, representing an atom in either the protein or ligand, we employ an interaction-based graph neural network with geometric information (GIGN) (Yang et al., 2023). This model updates

node embeddings by aggregating information from neighboring nodes, incorporating both structural and geometric data while ensuring translation and rotation invariance in the 3D coordinate space. $\boldsymbol{h}_i = \text{GNN}(\boldsymbol{x}_i, \boldsymbol{r}_i, \mathcal{N}(v_i))$ where $\boldsymbol{h}_i \in \mathbb{R}^d$ is the updated embedding for the node $v_i$, $\mathcal{N}(v_i)$ denotes the set of neighboring nodes, and $d$ is the embedding dimension. While this work utilizes GIGN, other graph neural networks, such as GCN (Kipf & Welling, 2016), or SE(3)-equivariant encoders like EGNN (Satorras et al., 2021) and SE(3)-Transformer (Fuchs et al., 2020), can also be applied. The embeddings $\boldsymbol{h}_i$ produced in this step serve as inputs for the subsequent cluster-level representations, ensuring that both local and global interaction patterns are effectively captured.

### 3.3 CLUSTER-ATTENTION FOR PROTEIN-LIGAND COMPLEX

Traditional models often focus on atom-level interactions, which can lead to excessive computational complexity. To address this, we propose a novel **cluster-attention** mechanism that clusters atoms using a differentiable pooling mechanism and applies cross-attention at the cluster level.

#### 3.3.1 CLUSTER-LEVEL PROTEIN-LIGAND INTERACTION

In protein-ligand complexes, it is often unclear which atoms interact most significantly. To address this, we employ a differentiable pooling method (Ying et al., 2018) to group atoms into clusters, capturing interaction patterns at a higher level of abstraction. By learning soft cluster assignment matrices and generating cluster embeddings, we reduce the complexity of the graph while preserving critical structural and interaction information.

First, a soft cluster assignment is performed separately for the ligand and protein atoms. This allows the model to aggregate atoms with similar representations. The soft cluster assignment matrices for the ligand and protein, $\boldsymbol{S}_l \in \mathbb{R}^{|V_l| \times c_l}$ and $\boldsymbol{S}_p \in \mathbb{R}^{|V_p| \times c_p}$, are computed as:

$$\boldsymbol{S}_l = \text{softmax}(\text{GNN}_{\theta_l})(\boldsymbol{H}_l, E_l) \quad \boldsymbol{S}_p = \text{softmax}(\text{GNN}_{\theta_p})(\boldsymbol{H}_p, E_p) \tag{1}$$

where $\boldsymbol{H}_l \in \mathbb{R}^{|V_l| \times d}$ and $\boldsymbol{H}_p \in \mathbb{R}^{|V_p| \times d}$ are the atom-level embeddings for the ligand and protein, respectively, with $E_l$ and $E_p$ as their intra-molecular edges. $c_l$ and $c_p$ denote the numbers of clusters for the ligand and protein.

Once the cluster assignments are obtained, we compute the high-level cluster representations by aggregating the atom embeddings within each cluster. The cluster-level embeddings for the ligand and protein are given by:

$$\boldsymbol{Z}_l = \boldsymbol{S}_l^T \boldsymbol{H}_l, \quad \boldsymbol{Z}_p = \boldsymbol{S}_p^T \boldsymbol{H}_p \tag{2}$$

where $\boldsymbol{Z}_l \in \mathbb{R}^{c_l \times d}$ and $\boldsymbol{Z}_p \in \mathbb{R}^{c_p \times d}$ represent the cluster embeddings for the ligand and protein, respectively. The cluster adjacency matrices, which capture the interactions between clusters, are updated based on the original graph structure as follows:

$$\tilde{\mathbf{A}}_l = \mathbf{S}_l^T \mathbf{A}_l \mathbf{S}_l, \quad \tilde{\mathbf{A}}_p = \mathbf{S}_p^T \mathbf{A}_p \mathbf{S}_p \tag{3}$$

where $\boldsymbol{A}_l$ and $\boldsymbol{A}_p$ denote the adjacency matrix of $E_l$ and $E_p$, respectively. $\tilde{\boldsymbol{A}}_l \in \mathbb{R}^{c_l \times c_l}$ and $\tilde{\boldsymbol{A}}_p \in \mathbb{R}^{c_p \times c_p}$ represent the cluster-level adjacency matrices for the ligand and protein, respectively.

Next, we finally update the cluster-level embeddings based on the cluster representations and cluster adjacency matrices. Formally, the final cluster representations for the ligand and protein are computed as follows:

$$\boldsymbol{Z}_l^{\text{final}} = \text{GNN}_{\psi_l}(\boldsymbol{Z}_l, \tilde{\mathbf{A}}_l), \quad \boldsymbol{Z}_p^{\text{final}} = \text{GNN}_{\psi_p}(\boldsymbol{Z}_p, \tilde{\mathbf{A}}_p) \tag{4}$$

where $\boldsymbol{Z}_l^{\text{final}} \in \mathbb{R}^{c_l \times d}$ and $\boldsymbol{Z}_p^{\text{final}} \in \mathbb{R}^{c_p \times d}$ denote the final cluster-level feature representations for the ligand and protein, respectively.

#### 3.3.2 CROSS-ATTENTION MECHANISMS ON CLUSTERS

After obtaining cluster-level representations, we apply a cross-attention mechanism (Vaswani, 2017; Chen et al., 2021; Lin et al., 2022) between the protein and ligand clusters to capture the critical intermolecular interactions. This mechanism serves not only to capture key interactions between clusters but also to filter out irrelevant or noisy clusters, allowing the model to focus on the most biologically

meaningful binding interactions. By dynamically adjusting the attention weights, CheapNet effectively selects the clusters that are most predictive of the binding affinity, thereby enhancing both efficiency and accuracy.

We first compute the query, key, and value matrices for the ligand-to-protein (L2P) and protein-to-ligand (P2L) attention mechanisms. For the L2P attention, the query, key, and value matrices are given by (for simplicity, we omit the superscript $final$).:

$$\boldsymbol{Q}_{l2p} = \boldsymbol{W}_{Q_{l2p}}\boldsymbol{Z}_l, \quad \boldsymbol{K}_{l2p} = \boldsymbol{W}_{K_{l2p}}\boldsymbol{Z}_p, \quad \boldsymbol{V}_{l2p} = \boldsymbol{W}_{V_{l2p}}\boldsymbol{Z}_p \tag{5}$$

where $\boldsymbol{W}_{Q_{l2p}}$, $\boldsymbol{W}_{K_{l2p}}$, and $\boldsymbol{W}_{V_{l2p}}$ are learnable weight matrices. Similarly, for the P2L attention,

$$\boldsymbol{Q}_{p2l} = \boldsymbol{W}_{Q_{p2l}}\boldsymbol{Z}_p, \quad \boldsymbol{K}_{p2l} = \boldsymbol{W}_{K_{p2l}}\boldsymbol{Z}_l, \quad \boldsymbol{V}_{p2l} = \boldsymbol{W}_{V_{p2l}}\boldsymbol{Z}_l \tag{6}$$

The attention weights and representations for both directions are computed using the scaled dot-product attention:

$$\boldsymbol{Z}_{l2p} = \text{softmax}(\frac{\boldsymbol{Q}_{l2p}\boldsymbol{K}_{l2p}^T}{\sqrt{d}})\boldsymbol{V}_{l2p}, \quad \boldsymbol{Z}_{p2l} = \text{softmax}(\frac{\boldsymbol{Q}_{p2l}\boldsymbol{K}_{p2l}^T}{\sqrt{d}})\boldsymbol{V}_{p2l} \tag{7}$$

The representations of ligand-to-protein $\boldsymbol{Z}_{l2p}$ and protein-to-ligand $\boldsymbol{Z}_{p2l}$ are combined to form the final representation $\boldsymbol{Z}_{complex}$ of the complex with multi-layer perceptron (MLP) and residual connection:

$$\boldsymbol{Z}_{complex} = MLP(\sum_{i=1}^{c_l} \boldsymbol{Z}_{l2p}^{(i,:)} + \sum_{j=1}^{c_p} \boldsymbol{Z}_{p2l}^{(j,:)}) + \sum_{i=1}^{c_l} \boldsymbol{Z}_l^{(i,:)} + \sum_{j=1}^{c_p} \boldsymbol{Z}_p^{(j,:)} \tag{8}$$

Finally, this combined representation is passed through a MLP-based classifier $f_{clf}$ to predict the binding affinity: $\hat{y} = f_{clf}(\boldsymbol{Z}_{complex})$.

### 3.4 PERMUTATION INVARIANCE OF CLUSTER ORDER FOR CROSS ATTENTION

An essential property of the proposed cluster-level cross-attention mechanism is its permutation invariance with respect to cluster ordering, ensuring consistent model outputs regardless of the order of ligand and protein clusters. This property enhances the robustness and reliability in processing cluster-level representations. The detailed proof is provided in Appendix A.3. Additionally, CheapNet's modularity allows for the integration of (S)E(3)-equivariant encoders, enabling CheapNet to address a broader range of symmetries in protein-ligand interactions.

### 3.5 LOSS FUNCTION FOR OPTIMIZATION

To optimize our model, we employ the mean squared error (MSE) loss function, which quantifies the L2 distance between the predicted binding affinity and the actual value. The MSE loss is defined as: $\mathcal{L}_{\text{MSE}} = \frac{1}{n}\sum_{i=1}^{n}(\hat{y}_i - y_i)^2$ where $\hat{y}_i$ represents the predicted binding affinity for the $i$-th protein-ligand complex, $y_i$ is the corresponding ground-truth value and $n$ is the total number of samples.

We explore incorporating additional loss functions, such as link prediction and entropy regularization losses proposed by Ying et al. (2018), to guide clustering based on geometric proximity. However, ablation studies in Appendix A.11 show no significant performance improvements. This suggests that clustering atoms based on geometric proximity does not align with our goal of dynamically identifying biologically meaningful clusters. Thus, we retain the MSE loss for its simplicity and effectiveness in optimizing binding affinity predictions.

## 4 EXPERIMENTS

In this section, we evaluate our CheapNet on various protein-ligand affinity tasks including ligand binding affinity (LBA) prediction, ligand efficacy prediction (LEP). Detailed hyperparameter setting and experimental setup are provided in Appendix A1. We provide comprehensive comparisons with state-of-the-art methods, as well as ablation studies to assess the contributions of individual components. The code is available at https://github.com/hyukjunlim/CheapNet.

Table 1: Performance comparison of CheapNet and baselines on the cross-dataset evaluation with parameter counts. The top results are shown in **bold**, and the second-best are underlined, respectively. The complete results, including all baselines with standard deviations are at Appendix A.6.

| Model | Params # | v2013 core set | | v2016 core set | | v2019 holdout set | |
|---|---|---|---|---|---|---|---|
| | | RMSE ↓ | Pearson ↑ | RMSE ↓ | Pearson ↑ | RMSE ↓ | Pearson ↑ |
| **Interaction-free** | | | | | | | |
| DeepDTA (Öztürk et al., 2018) | 1.93M | 1.639 | 0.718 | 1.357 | 0.785 | 1.485 | 0.586 |
| GraphDTA-GAT-GCN (Nguyen et al., 2021) | 4.75M | 1.645 | 0.711 | 1.434 | 0.754 | 1.705 | 0.474 |
| MGraphDTA (Yang et al., 2022) | 3.05M | 1.680 | 0.696 | 1.439 | 0.753 | 1.553 | 0.538 |
| **Interaction-based** | | | | | | | |
| PotentialNet (Feinberg et al., 2018) | 0.08M | 1.607 | 0.773 | 1.503 | 0.772 | 1.514 | 0.564 |
| SchNet (Schütt et al., 2017) | 0.28M | 1.570 | 0.754 | 1.390 | 0.787 | 1.522 | 0.560 |
| GNN-DTI (Lim et al., 2019) | 0.22M | 1.533 | 0.767 | 1.384 | 0.779 | 1.446 | 0.614 |
| IGN (Jiang et al., 2021) | 1.66M | 1.428 | 0.807 | 1.269 | 0.821 | 1.410 | 0.630 |
| EGNN (Satorras et al., 2021) | 1.59M | 1.498 | 0.782 | 1.289 | 0.816 | 1.399 | 0.628 |
| GIGN (Yang et al., 2023) | 0.62M | 1.380 | 0.821 | 1.190 | 0.840 | 1.393 | 0.641 |
| **Interaction-based (attention mechanism)** | | | | | | | |
| AttentionSiteDTI (Yazdani-Jahromi et al., 2022) | 42.66M | 1.444 | 0.792 | 1.352 | 0.784 | 1.539 | 0.563 |
| CAPLA (Jin et al., 2023) | 0.31M | 1.409 | 0.816 | 1.206 | 0.841 | - | - |
| GAABind (Tan et al., 2024) | 17.95M | 1.488 | 0.772 | 1.297 | 0.803 | - | - |
| DEAttentionDTA (Chen et al., 2024) | 2.32M | 1.470 | 0.800 | 1.266 | 0.827 | - | - |
| **Interaction-based (cluster-attention mechanism)** | | | | | | | |
| CheapNet (ours) | 1.33M | **1.262** | **0.857** | **1.107** | **0.870** | **1.343** | **0.665** |

## 4.1 LIGAND BINDING AFFINITY

**Task.** The Ligand Binding Affinity (LBA) task aims to predict the strength of interaction between a protein and a ligand. This regression task estimates the binding affinity of a protein-ligand complex.

**Dataset & Evaluation.** We evaluate CheapNet on the widely-used PDBbind dataset (Liu et al., 2017b), which contains 3D structures of protein-ligand complexes with experimentally measured binding affinities. For a fair comparison, we follow the experimental settings, including data splits, used in existing works. CheapNet is evaluated in two settings:

- *Cross-dataset evaluation*: Following the protocol in GIGN (Yang et al., 2023), we train and validate CheapNet on the PDBbind v2016 general set and test it on three independent datasets: PDBbind v2013 core set, v2016 core set, and v2019 holdout set. This configuration assesses CheapNet's generalization across different dataset versions.

- *Diverse Protein evaluation*: As in Atom3D (Townshend et al., 2020), the PDBbind v2019 refined set (Liu et al., 2017b) is divided based on protein sequence identity thresholds of 30% (LBA 30%) and 60% (LBA 60%), ensuring that the test proteins have reduced similarity to those in the training set. This setup is designed to assess the model's robustness to structurally diverse proteins.

The number of clusters for both protein and ligand is predefined as a constant through hyperparameter tuning, with the median number of nodes in the training set chosen to balance overfitting and generalizability (see Appendix A.10). We report RMSE and Pearson correlation coefficient for both settings, with the addition of Spearman correlation in the diverse protein evaluation. Each experiment is conducted three times with different random seeds for reliability.

**Baselines.** We compare CheapNet against a diverse range of baselines of interaction-free methods and interaction-based methods. We also include pre-training models that are trained on large-scale data with significantly larger model parameters; interaction-free models (e.g., DeepDTA (Öztürk et al., 2018), GraphDTA (Nguyen et al., 2021)), interaction-based models (e.g., IGN (Jiang et al., 2021), GIGN (Yang et al., 2023)), cluster-level models (e.g., LEFTNet (Du et al., 2024), GET (Kong et al., 2024)), and pre-training models (e.g., BindNet (Feng et al., 2024)).

**Performances.** Tables 1 and 2 summarize the results for the LBA task. CheapNet demonstrates significant improvements across both evaluation settings, outperforming interaction-free, interaction-based, and even pre-trained models. Notably, CheapNet achieves the best results in terms of RMSE and Pearson correlation coefficient for the cross-dataset evaluation, showcasing its

Table 2: Performance comparison of CheapNet and baselines on the diverse protein evaluation with parameter counts. The top results are shown in **bold**, and the second-best are underlined, respectively. The complete results, including all baselines with standard deviations are at Appendix A.6.

| Model | Params # | LBA 30% | | | LBA 60% | | |
|---|---|---|---|---|---|---|---|
| | | RMSE ↓ | Pearson ↑ | Spearman ↑ | RMSE ↓ | Pearson ↑ | Spearman ↑ |
| **Interaction-free** | | | | | | | |
| DeepDTA (Öztürk et al., 2018) | 1.93M | 1.866 | 0.472 | 0.471 | 1.762 | 0.666 | 0.663 |
| SSA (Bepler & Berger, 2019) | 48.8M | 1.985 | 0.165 | 0.152 | 1.891 | 0.249 | 0.275 |
| TAPE (Rao et al., 2019) | 93.0M | 1.890 | 0.338 | 0.286 | 1.633 | 0.568 | 0.571 |
| **Interaction-based** | | | | | | | |
| Atom3D-GNN (Townshend et al., 2021) | 0.38M | 1.601 | 0.545 | 0.533 | 1.408 | 0.743 | 0.743 |
| IEConv (Hermosilla et al., 2021) | 5.80M | 1.554 | 0.414 | 0.428 | 1.473 | 0.667 | 0.675 |
| ProNet (Wang et al., 2023) | 1.39M | 1.463 | 0.551 | 0.551 | 1.343 | 0.765 | 0.761 |
| **Cluster-level** | | | | | | | |
| GemNet (Gasteiger et al., 2021)[a] | 1.37M | | OOM | | - | - | - |
| Equiformer (Liao & Smidt, 2022)[a] | 1.10M | | OOM | | - | - | - |
| LEFTNet (Du et al., 2024)[a] | 0.85M | 1.366 | 0.592 | 0.580 | - | - | - |
| GET (Kong et al., 2024) | 0.69M | 1.327 | 0.620 | 0.611 | - | - | - |
| **Pre-training** | | | | | | | |
| EGNN-PLM (Wu et al., 2023) | 650M | 1.403 | 0.565 | 0.544 | 1.559 | 0.644 | 0.646 |
| Uni-Mol (Zhou et al., 2023) | 47.61M | 1.434 | 0.565 | 0.540 | 1.357 | 0.753 | 0.750 |
| ProFSA (Gao et al., 2023) | >47.61M[b] | 1.377 | 0.628 | 0.620 | 1.377 | 0.764 | 0.762 |
| BindNet (Feng et al., 2024) | >47.61M[b] | 1.340 | 0.632 | 0.620 | **1.230** | 0.793 | 0.788 |
| **Interaction-based (cluster-attention)** | | | | | | | |
| CheapNet (ours) | 1.39M | **1.311** | **0.642** | **0.639** | 1.238 | **0.794** | **0.789** |

[a] Adapted from GET (Kong et al., 2024), which used hierarchical approaches from atom-level to block-level.
[b] Accurate parameter estimation for ProFSA and BindNet is not possible due to the unavailability of the pre-training model checkpoints. However, their parameter count is likely higher than that of Uni-Mol, as both models are based on it.

ability to capture complex protein-ligand interactions. For the diverse protein evaluation, although CheapNet achieved a comparable result on the LBA 60% dataset, slightly trailing BindNet, it is particularly noteworthy that it demonstrates exceptional first-place performance on the more challenging LBA 30% dataset, where there is lower similarity between the training and test sets. Despite using far fewer parameters and requiring no pre-training, CheapNet consistently outperforms more complex models, highlighting its efficiency and robustness.

## 4.2 LIGAND EFFICACY PREDICTION

**Task.** Ligand Efficacy Prediction (LEP) is a binary classification task that predicts whether a ligand activates or inactivates a target protein. This task is crucial in drug discovery, as it helps identify potential drug candidates that either enhance or inhibit protein activity.

**Dataset & Evaluation.** For a fair comparison, we evaluate CheapNet using the LEP dataset and experimental setting derived from the Atom3D benchmark (Townshend et al., 2020). The dataset contains protein-ligand complexes labeled for activation or inactivation. For evaluation, we report Area Under the Receiver Operating Characteristic Curve (AUROC) and Area Under the Precision-Recall Curve (AUPRC). Each experiment is run independently with different random seeds.

**Baselines.** We compare CheapNet against a range of models, including interaction-free methods such as DeepDTA (Öztürk et al., 2018); interaction-based methods such as ProNet (Wang et al., 2023); cluster-level approach such as GET (Kong et al., 2024); pre-treining methods such as BindNet (Feng et al., 2024).

Table 3: Comparison results of CheapNet and baselines on LEP datasets. The top results are shown in **bold**, and the second-best are underlined, respectively. The complete results, including all baselines with standard deviations are at Appendix A.7.

| Model | AUROC ↑ | AUPRC ↑ |
|---|---|---|
| **Interaction-free** | | |
| DeepDTA (Öztürk et al., 2018) | 0.696 | - |
| **Interaction-based** | | |
| Atom3D-GNN (Townshend et al., 2021) | 0.681 | 0.598 |
| GVP-GNN (Jing et al., 2021) | 0.628 | - |
| ProNet-All-Atom (Wang et al., 2023) | 0.692 | - |
| **Cluster-level** | | |
| SchNet (Schütt et al., 2017)[a] | 0.736 | 0.731 |
| EGNN (Satorras et al., 2021)[a] | 0.724 | 0.720 |
| GET (Kong et al., 2024)[a] | 0.761 | 0.751 |
| **Pre-training** | | |
| GeoSSL (Liu et al., 2022) | 0.776 | 0.694 |
| Uni-Mol (Zhou et al., 2023) | 0.823 | 0.787 |
| ProFSA (Gao et al., 2023) | 0.840 | 0.806 |
| BindNet (Feng et al., 2024) | 0.882 | 0.870 |
| **Interaction-based (cluster-attention)** | | |
| CheapNet (ours) | **0.935** | **0.924** |

[a] Adapted from GET (Kong et al., 2024), which used hierarchical approaches from atom-level to block-level.

Table 4: Ablation study results showing RMSE, Pearson correlation coefficient, and performance improvement ($\Delta$) for different graph encoders on the PDBbind v2013 core set, v2016 core set, and v2019 holdout set. The top results are shown in **bold**, and the second-best are underlined, respectively. Standard deviations are at Appendix A.8.

| Model | v2013 core set | | v2016 core set | | v2019 holdout set | |
|---|---|---|---|---|---|---|
| | RMSE ↓ | Pearson ↑ | RMSE ↓ | Pearson ↑ | RMSE ↓ | Pearson ↑ |
| GCN (Kipf & Welling, 2016) | 1.395 | 0.819 | 1.295 | 0.809 | 1.460 | 0.593 |
| CheapNet-GCN | 1.368 | 0.820 | 1.246 | 0.823 | 1.391 | 0.635 |
| $\Delta(\%)$ | +1.935 | +0.122 | +3.784 | +1.731 | +4.726 | +7.083 |
| EGNN (Satorras et al., 2021) | 1.498 | 0.782 | 1.289 | 0.811 | 1.399 | 0.628 |
| CheapNet-EGNN | 1.321 | 0.843 | 1.161 | 0.856 | 1.343 | 0.664 |
| $\Delta(\%)$ | +11.816 | +7.801 | +9.930 | +5.549 | +4.003 | +5.732 |
| GIGN (Yang et al., 2023) | 1.380 | 0.821 | 1.190 | 0.840 | 1.393 | 0.641 |
| CheapNet-GIGN | 1.262 | 0.857 | 1.107 | 0.870 | 1.343 | 0.665 |
| $\Delta(\%)$ | +8.551 | +4.385 | +6.975 | +3.571 | +3.589 | +3.744 |

**Performances.** As shown in Table 3, CheapNet achieves state-of-the-art performance on the LEP task, significantly outperforming all baselines, including larger pre-training models. For AUROC, CheapNet achieves a score of 0.935, surpassing the previous best by BindNet (0.882), as well as other models like Uni-Mol (0.823) and GeoSSL (0.776). For AUPRC, CheapNet also achieves the best score of 0.924, outperforming BindNet (0.870). This performance is attributed to CheapNet's cluster-attention mechanism, which effectively captures complex protein-ligand relationships.

## 4.3 ABLATION STUDIES

To demonstrate the effectiveness of CheapNet components, we conduct ablation studies: (1) adaptability of cluster-attention, (2) hierarchical representations and attention mechanism, (3) number of clusters, (4) auxiliary loss function, (5) atom selection & grouping approaches. Due to the limited space, experimental results of (3)-(5) are reported in Appendix A.10, A.11, and A.12, respectively.

### 4.3.1 ADAPTABILITY OF CHEAPNET WITH DIFFERENT GRAPH ENCODER

In this section, we demonstrate the adaptability of CheapNet by evaluating its performance when combined with different graph encoders. Specifically, we assess how adding CheapNet's cluster-attention mechanisms impacts models using GCN, EGNN, and GIGN as base encoders.

Table 4 shows that CheapNet consistently improves performance across all encoders, regardless of the underlying architecture. While GIGN achieves the highest overall performance, both EGNN and GCN also benefit from notable improvements when paired with CheapNet's hierarchical representation and cluster-level attention mechanisms. Notably, GCN, which does not use 3D structural information, achieves substantial gains, demonstrating CheapNet's flexibility in enhancing atom-level encoders and improving accuracy across datasets.

### 4.3.2 HIERARCHICAL REPRESENTATIONS AND ATTENTION MECHANISMS

We perform an ablation study to evaluate the impact of hierarchical representations (Cluster) and attention mechanisms (Attention) on CheapNet's performance. Table 5 presents the RMSE and Pearson correlation coefficient across the PDBbind v2013 core set, v2016 core set, and v2019 holdout set.

The results show that both hierarchical representations and cross-attention work together to improve CheapNet's performance. Cluster-level representations are particularly effective on the v2019 holdout set, where larger protein-ligand complexes benefit from reduced computational complexity and better interaction modeling at higher scales (see Appendix A.5 for details). Otherwise, Cross-attention mechanisms enable CheapNet to focus on biologically meaningful interactions by filtering out irrelevant clusters, which is reflected in the sharp performance drop when this component is removed.

Table 5: Ablation study results for the effect of using hierarchical representations (Hierarchical), and type of attention mechanism (Attention) on the PDBbind v2013 core set, v2016 core set, and v2019 holdout set. The top results are shown in **bold**, and the second-best are underlined, respectively. Standard deviations are at Appendix A.9.

| Cluster | Attention | v2013 core set | | v2016 core set | | v2019 holdout set | |
|---|---|---|---|---|---|---|---|
| | | RMSE ↓ | Pearson ↑ | RMSE ↓ | Pearson ↑ | RMSE ↓ | Pearson ↑ |
| ✗ | ✗ | 1.345 | 0.844 | 1.189 | 0.851 | 1.360 | 0.652 |
| ✗ | Self | 1.305 | 0.850 | 1.166 | 0.854 | 1.367 | 0.650 |
| ✗ | Cross | 1.293 | 0.853 | 1.151 | 0.857 | 1.362 | 0.653 |
| ✓ | ✗ | 1.330 | 0.840 | 1.161 | 0.853 | 1.348 | 0.662 |
| ✓ | Self | 1.327 | 0.841 | 1.168 | 0.853 | 1.348 | 0.662 |
| ✓ | Cross | **1.262** | **0.857** | **1.107** | **0.870** | **1.343** | **0.665** |

## 4.4 EVALUATION ON EXTERNAL BENCHMARKS

To demonstrate CheapNet's robustness and generalization, we evaluate its performance on three external benchmarks on protein-ligand related tasks: the CSAR NRC-HiQ dataset (Dunbar Jr et al., 2013), the CASF-2016 dataset (Su et al., 2018), virtual screening on DUD-E dataset (Mysinger et al., 2012). These benchmarks assess the model's predictive power on unseen, structurally diverse protein-ligand complexes, testing its real-world applicability. Furthermore, we extend the scope of CheapNet to Protein-Protein Affinity (PPA) prediction on Protein-Protein Affinity Benchmark Version2 (Vreven et al., 2015) to evaluate its generalizability. Due to the limited space, the evaluations of the CASF-2016 dataset , the DUD-E dataset, and the PPA prediction are presented in the Appendix A.14 , A.15 and A.20.

### 4.4.1 CSAR NRC-HiQ DATASET

We evaluate CheapNet on the CSAR NRC-HiQ dataset (Dunbar Jr et al., 2013), an external benchmark for protein-ligand binding affinity prediction. After removing complexes that RDKit could not process and overlaps with the training data, 14 samples remained for evaluation. Table 6 compares CheapNet with other interaction-based methods.

CheapNet achieve the best performance, with an RMSE of 1.381 and a Pearson correlation coefficient of 0.901, outperforming all other models. These results highlight CheapNet's ability to handle complex protein-ligand interactions, especially in the external dataset.

Table 6: Performance comparison of CheapNet on the CSAR NRC-HiQ dataset. The top results are shown in **bold**, and the second-best are underlined, respectively. Standard deviations are at Appendix A.13.

| Model | RMSE ↓ | Pearson ↑ |
|---|---|---|
| **Interaction-based** | | |
| PotentialNet (Feinberg et al., 2018) | 1.730 | 0.718 |
| GNN-DTI (Lim et al., 2019) | 1.675 | 0.855 |
| IGN (Jiang et al., 2021) | 1.647 | 0.846 |
| EGNN (Satorras et al., 2021) | 1.640 | 0.866 |
| GIGN (Yang et al., 2023) | 1.827 | 0.766 |
| **Interaction-based (cluster-attention mechanism)** | | |
| CheapNet (ours) | **1.381** | **0.901** |

### 4.5 MEMORY FOOTPRINT ANALYSIS

Figure 3 compares the memory usage of CheapNet with other attention-based models, GAABind (Tan et al., 2024) and DEAttentionDTA (Chen et al., 2024), across different batch sizes and complex sizes.

GAABind, which uses atom-level all-pairwise attention, consumes substantial memory and can only handle small batch sizes for small complexes. In contrast, DEAttentionDTA is more memory-efficient with residue-level protein representations but still requires significant memory for larger complexes due to residue-to-atom attention calculations.

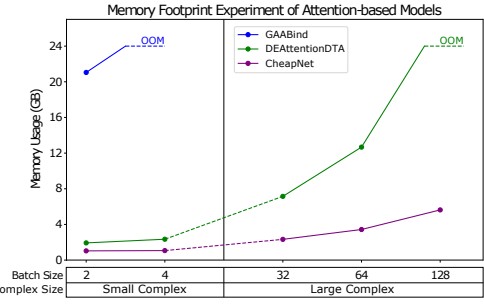

Figure 3: **Memory footprint analysis.** Comparison of CheapNet, GAABind, and DEAttentionDTA across different batch sizes for small (50–100 atoms) and large (400–450 atoms) complexes. 'OOM' indicates out-of-memory.

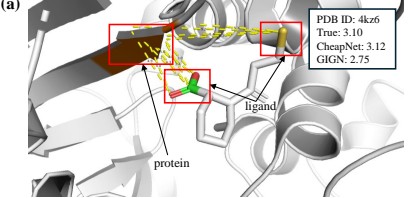 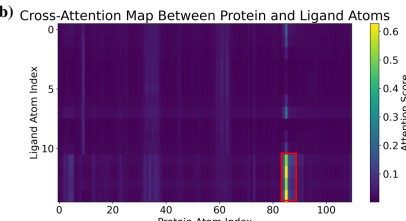

Figure 4: **Visualization of CheapNet interpretation and cross-attention map of protein-ligand complex (PDB ID: 4kz6).** (a) The high-attended pairs of ligand and protein atoms are highlighted in the red box, with yellow dashed lines representing interactions. (b) Cross-attention map between ligand and protein atoms. The high-attention regions marked in yellow within the red box.

In comparison, CheapNet maintains consistently low memory usage across varying batch and complex sizes, even handling large protein-ligand interactions efficiently without OOM issues. This efficiency underscores the advantage of CheapNet's cluster-level attention mechanism, which captures the essential binding interactions without the memory overhead typical of atom-level approaches. These results highlight CheapNet's scalability and suitability for handling large, complex interactions. Detailed experimental setups can be found in Appendix A.16.

## 4.6    INTERPRETABILITY OF CHEAPNET

This section analyzes the interpretability of CheapNet using the protein-ligand complex in PDBbind v2016 core set (PDB ID: 4kz6). Figure 4 provides key insights into how CheapNet focuses attention on critical binding regions. Further visualizations and details on how the attention scores are computed are provided in Appendix A.17.

CheapNet's cluster-attention mechanism enables the identification of significant interactions between ligand and protein, by focusing on the most relevant clusters involved in binding. As seen in Figure 4, CheapNet assigns higher attention to clusters that are known to be critical for binding, while assigning low attention weights to less relevant clusters, thereby demonstrating its ability to filter out noise.

Therefore, by accurately capturing key atomic interactions through its cluster-attention mechanism, CheapNet not only achieves near-perfect binding affinity predictions but also offers clear visual evidence of the interactions driving these affinities. This interpretability makes CheapNet a powerful tool for understanding the molecular mechanisms behind protein-ligand binding, which is crucial for drug discovery applications.

## 5    CONCLUSION & DISCUSSION

In this paper, we propose CheapNet, a novel interaction-based model that captures protein-ligand binding affinity by integrating hierarchical cluster-level representations with cross-attention mechanisms. By leveraging a differentiable pooling approach, CheapNet effectively balances capturing intricate inter-molecular interactions with computational efficiency. Extensive evaluations demonstrate state-of-the-art performance across diverse datasets, suggesting that hierarchical modeling of molecular interactions is a promising direction for enhancing binding affinity prediction.

Although CheapNet achieves strong results, several directions remain for further exploration. Its performance benefits from strong protein and ligand encoders, and integrating SE(3)-equivariant encoders could enhance its ability to handle global and local 3D symmetries. While CheapNet can operate without 3D structural data (as shown in Table 4), optimal performance relies on 3D information. Advances like AlphaFold3 (Abramson et al., 2024) now provide access to predicted structures, and CheapNet could be further developed to remain robust even with noise in these predictions (see Appendix A.18). Finally, extending the use of 3D information to cluster-level attention or adopting dual-awareness framework that combine atom- and cluster-level features offers exciting potential for future work (see Appendix A.19).

ACKNOWLEDGMENTS

This paper was partly supported by the Basic Science Research Program through the National Research Foundation of Korea (NRF) funded by the Ministry of Education (RS-2023-00246586), the Bio & Medical Technology Development Program of NRF funded by the Ministry of Science & ICT(NRF-2022M3E5F3085677), Institute of Information and communications Technology Planning and Evaluation (IITP) grant funded by the Korea government (MSIT) [NO. 2021-0-01343, Artificial Intelligence Graduate School Program (Seoul National University), No.RS-2022-00155915, Artificial Intelligence Convergence Innovation Human Resources Development (Inha University)], the ICT at Seoul National University, and AIGENDRUG Co. Ltd., by Korea Institute for Advancement of Technology (KIAT) grant funded by the Korea Government (Ministry of Education) (P0025681-G02P22450002201-10054408, Semiconductor-Specialized University).

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

# A  APPENDIX

## A.1  ADDITIONAL EXPLANATIONS OF RELATED WORKS.

In this section, we briefly introduce several representative approaches including interaction-free, interaction-based, cluster-level, and pre-training models for protein-ligand binding affinity prediction.

### A.1.1  INTERACTION-FREE MODELS

- DeepDTA (Öztürk et al., 2018) uses convolutional neural networks (CNNs) to analyze SMILES representations of molecules. This approach demonstrates the potential of deep learning to capture molecular features effectively. Instead, the focus is primarily on learning molecular representations independently for proteins and ligands.

- GraphDTA (Nguyen et al., 2021) and MGraphDTA (Yang et al., 2022) extend DeepDTA by representing molecules as graphs, using various graph neural network (GNN) architectures such as Graph Convolutional Networks (GCN), Graph Isomorphism Networks (GIN), and Graph Attention Networks (GAT). These approaches capture the structural information of molecules more effectively compared to SMILES-based representations.

### A.1.2  INTERACTION-BASED MODELS

- IGN (Jiang et al., 2021) introduce a significant advancement by employing both intra-graph and inter-graph convolutions to capture pairwise atomic interactions. This allow the model to account for both local and global interactions within the protein-ligand complex, leading to more accurate binding affinity predictions.

- Equivariant Graph Neural Networks (EGNN) (Satorras et al., 2021) addresses a critical challenge in molecular modeling, ensuring that predictions are invariant to changes in the orientation or position of the protein-ligand complex. This geometric invariance ensures that predictions remain physically meaningful, regardless of how the complex is rotated or translated.

- Geometric Interaction Graph Neural Networks (GIGN) (Yang et al., 2023) further advances this concept by incorporating both intra- and inter-molecular geometric information, allowing the model to capture complex 3D spatial relationships within the protein-ligand complex.

### A.1.3  CLUSTER-LEVEL MODELS

- GemNet (Gasteiger et al., 2021) employs directional message passing to capture both local and global molecular interactions using geometric features such as distances and angles. It focuses on fine-grained spatial relationships, achieving high accuracy in molecular property prediction tasks.

- Equiformer (Liao & Smidt, 2022) combines Transformer architecture with SE(3)/E(3)-equivariant features to handle 3D molecular graphs. It integrates spherical harmonics and tensor products to represent complex interactions while preserving rotational and translational symmetry.

- LEFTNet (Du et al., 2024) introduces hierarchical representations for 3D molecular graphs, utilizing predefined clusters (e.g., residues or motifs) to encode higher-order interactions. It emphasizes computational efficiency while maintaining expressiveness.

- GET (Kong et al., 2024) models molecular complexes as geometric graphs of sets using a bilevel design. It captures block-level sparsity and atom-level density through bilevel attention mechanisms, ensuring adaptability across diverse molecular domains.

### A.1.4  PRE-TRAINING MODELS

- GeoSSL (Liu et al., 2022) introduces a 3D coordinate denoising pretraining framework designed to model the dynamic behavior of 3D molecules, where their continuous movement

within the 3D Euclidean space generates a smooth potential energy surface. Extensive experiments, including quantum mechanics and force prediction as well as binding affinity prediction, validate the effectiveness and robustness of this proposed method.

- ProFSA (Gao et al., 2023) introduces a novel pocket pretraining approach that harnesses knowledge from high-resolution atomic protein structures, supported by effective pretrained small molecule representations. By segmenting protein structures into drug-like fragments and corresponding pockets, ProFSA simulates ligand-receptor interactions, generating over 5 million complexes. The pocket encoder is then trained contrastively to align with pseudo-ligand representations from pretrained small molecule encoders.

- BindNet (Feng et al., 2024) emphasizes discerning intricate binding patterns from fine-grained interactions. This self-supervised learning problem is formulated as predicting the final binding complex structure given a pocket and ligand through a Transformer-based interaction module, which naturally emulates the binding process. To ensure the representation of rich binding information, two pretraining tasks are introduced: atomic pairwise distance map prediction and masked ligand reconstruction, comprehensively modeling fine-grained interactions in both structural and feature spaces.

## A.2 PSEUDOCODE OF CHEAPNET

In this section, we provide the detailed pseudocode for CheapNet, which outlines the step-by-step process for predicting protein-ligand binding affinity (Algorithm 1). The algorithm starts by initializing atom-level embeddings for both the protein and ligand components.

1. **Initialization**: The initialization process of the atom representation follows that of GIGN (Yang et al., 2023) for Cross-Dataset Evaluation, and Atom3D (Townshend et al., 2020) for Diverse Protein Evaluation and Ligand Efficacy Prediction. Both methods initialize each node's features using one-hot encoding based on atom types (e.g., elements like C, H, O, etc.). In addition, for GIGN, the degree of an atom, hybridization, and number of valence electrons are considered. For Atom3D, co-crystallized metals are considered (e.g., elements like Zn, Na, Fe, etc.) for proteins. Finally, linear layers are applied to obtain the initial embedding, refining the atom representation.

2. **Atom-Level Embedding**: Each atom's embeddings are updated using a Graph Neural Network (GNN), capturing intricate local interactions within the protein and ligand structures.

3. **Cluster-Level Representation**: Soft cluster assignment matrices are computed using the GNN outputs, enabling the model to form hierarchical, cluster-level representations. This step allows CheapNet to capture more abstract, functionally relevant features beyond the atom level. Finally, using the soft cluster assignments, cluster-level representations and adjacency matrices are derived. These cluster embeddings are further refined via a GNN, incorporating higher-level structural information.

4. **Cross-Attention Mechanism**: The core of CheapNet involves a cross-attention mechanism that models interactions between the protein and ligand clusters. The model computes query, key, and value matrices to perform scaled dot-product attention, ensuring that critical inter-molecular interactions are accurately captured. This step also filters out irrelevant clusters by focusing on those with higher attention weights.

5. **Final Representation**: The output of the cross-attention mechanism is combined and pooled to form a comprehensive representation of the protein-ligand complex, which is then passed through an MLP to predict binding affinity.

## A.3 PERMUTATION INVARIANCE OF CLUSTERS ORDER FOR CROSS ATTENTION

An important aspect of the proposed cross-attention mechanism on cluster-level representations is the permutation invariance of cluster ordering. This property ensures that the model's output remains consistent regardless of the order of ligand and protein clusters. Maintaining this invariance is crucial for the robustness of the model, as it prevents the network from being sensitive to arbitrary orderings of atoms or clusters, which should not influence the physical properties of the complex. By ensuring that the model's predictions are unaffected by cluster permutations, we preserve the reliability of our cluster-attention mechanism.

---

**Algorithm 1** CheapNet for Protein-Ligand Binding Affinity Prediction

---

**Require:** Protein-ligand complex graph $G = (V_l \cup V_p, E_l \cup E_p \cup E_{lp})$
**Ensure:** Predicted binding affinity $\hat{y}$

1: **Initialization:** Atom embeddings $\boldsymbol{H}_l \in \mathbb{R}^{|V_l| \times d}$, $\boldsymbol{H}_p \in \mathbb{R}^{|V_p| \times d}$, obtained by one-hot encoding followed by a linear transformation.

2: **Atom-Level Embedding:**
3: **for** each node $v_i \in V_l \cup V_p$ **do**
4: $\quad h_i = \text{GNN}(x_i, r_i, \mathcal{N}(v_i))$
5: **end for**

6: **Cluster-Level Representation:**
7: Compute soft cluster assignment matrices
$\quad \triangleright$ dynamically group together atoms by their embeddings
$$\boldsymbol{S}_l = \text{softmax}(\text{GNN}_{\theta_l})(\boldsymbol{H}_l, E_l), \quad \boldsymbol{S}_p = \text{softmax}(\text{GNN}_{\theta_p})(\boldsymbol{H}_p, E_p)$$

8: Obtain cluster-level representations:
$$\boldsymbol{Z}_l = \boldsymbol{S}_l^\top \boldsymbol{H}_l, \quad \boldsymbol{Z}_p = \boldsymbol{S}_p^\top \boldsymbol{H}_p$$

9: Obtain cluster-level adjacency matrices:
$$\tilde{\mathbf{A}}_l = \mathbf{S}_l^T \mathbf{A}_l \mathbf{S}_l, \quad \tilde{\mathbf{A}}_p = \mathbf{S}_p^T \mathbf{A}_p \mathbf{S}_p$$

10: Final update the cluster-level embeddings:
$\quad \triangleright$ learn cluster-level interactions for each ligand and protein before cluster-attention
$$\boldsymbol{Z}_l^{\text{final}} = \text{GNN}_{\psi_l}(\boldsymbol{Z}_l, \tilde{\mathbf{A}}_l), \quad \boldsymbol{Z}_p^{\text{final}} = \text{GNN}_{\psi_p}(\boldsymbol{Z}_p, \tilde{\mathbf{A}}_p)$$

11: **Cross-Attention Mechanism:**
12: For ligand-to-protein attention, compute query, key, value matrices:
$$\boldsymbol{Q}_{l2p} = \boldsymbol{W}_Q \boldsymbol{Z}_l^{\text{final}}, \quad \boldsymbol{K}_{l2p} = \boldsymbol{W}_K \boldsymbol{Z}_p^{\text{final}}, \quad \boldsymbol{V}_{l2p} = \boldsymbol{W}_V \boldsymbol{Z}_p^{\text{final}}$$

13: Apply scaled dot-product attention:
$\quad \triangleright$ filters out irrelevant clusters by focusing on those with higher attention weights
$$\boldsymbol{Z}_{l2p} = \text{softmax}\left(\frac{\boldsymbol{Q}_{l2p}\boldsymbol{K}_{l2p}^\top}{\sqrt{d}}\right)\boldsymbol{V}_{l2p}$$

14: For protein-to-ligand attention, perform similar computations L12-13:
$$\boldsymbol{Z}_{p2l} = \text{softmax}\left(\frac{\boldsymbol{Q}_{p2l}\boldsymbol{K}_{p2l}^\top}{\sqrt{d}}\right)\boldsymbol{V}_{p2l}$$

15: **Final Representation:**
16: Combine outputs:
$$\boldsymbol{Z}_{complex} = \text{MLP}(\sum_{i=1}^{c_l} \boldsymbol{Z}_{l2p}^{(i,:)} + \sum_{j=1}^{c_p} \boldsymbol{Z}_{p2l}^{(j,:)}) + \sum_{i=1}^{c_l} \boldsymbol{Z}_l^{(i,:)} + \sum_{j=1}^{c_p} \boldsymbol{Z}_p^{(j,:)}$$

17: **Prediction:**
$$\hat{y} = \text{MLP}(Z_{complex})$$

---

Consider the ligand-to-protein attention mechanism (for simplicity, we omit the subscript $l2p$). Assume permutation matrices $\boldsymbol{P}_\phi$ and $\boldsymbol{P}_\rho$ for the ligand and protein, respectively. The permuted cluster-level representations of the ligand and protein are given by:

$$\boldsymbol{Z}_l^\phi = \boldsymbol{P}_\phi \boldsymbol{Z}_l, \quad \boldsymbol{Z}_p^\rho = \boldsymbol{P}_\rho \boldsymbol{Z}_p \tag{9}$$

The corresponding permuted query, key, and value matrices are then:

$$\boldsymbol{Q}^{\phi} = \boldsymbol{W}_Q \boldsymbol{Z}_l^{\phi} = \boldsymbol{W}_Q \boldsymbol{P}_{\phi} \boldsymbol{Z}_l = \boldsymbol{P}_{\phi} \boldsymbol{W}_Q \boldsymbol{Z}_l = \boldsymbol{P}_{\phi} \boldsymbol{Q} \tag{10}$$

$$\boldsymbol{K}^{\rho} = \boldsymbol{W}_K \boldsymbol{Z}_p^{\rho} = \boldsymbol{W}_K \boldsymbol{P}_{\rho} \boldsymbol{Z}_p = \boldsymbol{P}_{\rho} \boldsymbol{W}_K \boldsymbol{Z}_p = \boldsymbol{P}_{\rho} \boldsymbol{K} \tag{11}$$

$$\boldsymbol{V}^{\rho} = \boldsymbol{W}_V \boldsymbol{Z}_p^{\rho} = \boldsymbol{W}_V \boldsymbol{P}_{\rho} \boldsymbol{Z}_p = \boldsymbol{P}_{\rho} \boldsymbol{W}_V \boldsymbol{Z}_p = \boldsymbol{P}_{\rho} \boldsymbol{V} \tag{12}$$

The attention weights for the permuted representations, denoted by $\alpha^{\phi,\rho}$, are computed as:

$$\alpha^{\phi,\rho} = \text{softmax}(\frac{\boldsymbol{Q}^{\phi}(\boldsymbol{K}^{\rho})^T}{\sqrt{d}}) = \text{softmax}(\frac{\boldsymbol{P}_{\phi} \boldsymbol{Q} (\boldsymbol{P}_{\rho} \boldsymbol{K})^T}{\sqrt{d}})$$
$$= \text{softmax}(\boldsymbol{P}_{\phi} \frac{\boldsymbol{Q}\boldsymbol{K}^T}{\sqrt{d}} \boldsymbol{P}_{\rho}^T) \tag{13}$$

Since $\boldsymbol{P}_{\phi}$ and $\boldsymbol{P}_{\rho}$ are permutation matrices, they simply reorder the rows and columns of the attention matrix. The softmax function is applied row-wise and is invariant to row permutations. Therefore:

$$\alpha^{\phi,\rho} = \boldsymbol{P}_{\phi} \, \text{softmax}(\frac{\boldsymbol{Q}\boldsymbol{K}^T}{\sqrt{d}}) \boldsymbol{P}_{\rho}^T \tag{14}$$

Next, the attention output is computed as:

$$\boldsymbol{Z}^{\phi,\rho} = \alpha^{\phi,\rho} \boldsymbol{V}^{\rho} = \boldsymbol{P}_{\phi} \, \text{softmax}(\frac{\boldsymbol{Q}\boldsymbol{K}^T}{\sqrt{d}}) \boldsymbol{P}_{\rho}^T \boldsymbol{P}_{\rho} \boldsymbol{V}$$
$$= \boldsymbol{P}_{\phi} \, \text{softmax}(\frac{\boldsymbol{Q}\boldsymbol{K}^T}{\sqrt{d}}) \boldsymbol{V} = \boldsymbol{P}_{\phi} \boldsymbol{Z} \tag{15}$$

Finally, we apply sum pooling over the cluster dimension $c_l$. Since the summation is invariant to the order of the elements, the sum pooling of the permuted attention output is:

$$\sum_{i=1}^{c_l} \boldsymbol{Z}^{\phi,\rho,(i,:)} = \sum_{i=1}^{c_l} \boldsymbol{Z}^{(i,:)} \tag{16}$$

Thus, the output of the sum pooling for the ligand-to-protein attention is permutation-invariant with respect to the ligand clusters.

The same reasoning applies to the protein-to-ligand attention mechanism. Therefore, since both the ligand-to-protein and protein-to-ligand outputs are pooled in a permutation-invariant manner, the final representation $\boldsymbol{Z}_{complex}$ will remain unchanged regardless of the order in which the ligand or protein clusters are arranged.

**Geometric Symmetries in Protein-Ligand Complexes and Their Treatment in CheapNet:** CheapNet's cluster-attention mechanism is designed to be permutation invariant with respect to the ordering of clusters in the graph representation, ensuring consistent outputs regardless of how ligand and protein clusters are indexed. However, this invariance pertains to the graph-level discrete ordering of clusters and should be distinguished from geometric symmetries (translation, rotation, and permutation of 3D coordinates) inherent to protein-ligand complexes.

When addressing symmetries in 3D coordinates, CheapNet's ability to handle translation, rotation, and permutation invariance relies on the properties of the atom-level encoder used to compute embeddings for proteins and ligands. In this study, we employed GIGN (Yang et al., 2023), which enforces translation and rotation invariance and is permutation equivariant by the nature of GNNs. These symmetry-preserving properties propagate to the cluster-level representations and outputs of CheapNet, ensuring global translation, rotation, and permutation invariance. However, CheapNet's cluster-attention mechanism itself operates on graph representations derived from these embeddings and does not directly enforce additional symmetries.

In local coordinates (e.g., within a specific cluster), rotation and permutation invariance depend entirely on the encoder's properties. For global coordinates (e.g., protein-ligand complex as a whole), the translation and rotation invariance of GIGN ensures that CheapNet can handle these symmetries

effectively. To further enhance its ability to capture symmetry-aware features, integrating (S)E(3)-equivariant encoders (Satorras et al., 2021; Fuchs et al., 2020) into CheapNet's modular framework is a promising direction for future improvement.

This modularity allows CheapNet to flexibly adapt to tasks and datasets with varying symmetry requirements. However, as the cluster-attention mechanism itself does not enforce additional symmetries, its performance might depend on the quality of the embeddings provided by the encoder. Future work could explore extending the use of 3D information within the cluster-attention mechanism itself to improve its ability to handle local symmetries dynamically.

## A.4  DETAILS OF HYPERPARAMETERS & EXPERIMENT SETTINGS

In table A1, we present the hyperparameter search space used to optimize CheapNet's performance across cross-dataset evaluation, diverse protein evaluation (LBA 30%, LBA 60%), and LEP. For LEP task, to combine the results of CheapNet for active and inactive complexes, we applied a multi-layer perceptron (MLP) and trained models using binary cross-entropy (BCE) loss. All experiments were conducted on two separate NVIDIA RTX 3090 GPUs (24GB each), with each model running on a single GPU. Each model was trained with early stopping based on validation RMSE.

Table A1: The search space of hyperparameters for cross-dataset, LBA 30%, LBA 60%, LEP task. The optimal hyperparameters are shown in **bold**.

| Hyperparameters | Cross-dataset | LBA 30% | LBA 60% | LEP |
|---|---|---|---|---|
| Activation function | SiLU, GELU, **Mish** | **Mish** | **Mish** | **Mish** |
| Batch size | 64, **128** | 8, **16** | 8, **16** | **4**, 8 |
| Cutoff-intra | - | **3Å** | **3Å** | **3Å** |
| Cutoff-inter | **5Å** | **5Å** | **5Å** | **5Å** |
| Dropout rate | 0, **0.1**, 0.2, 0.3 | 0, **0.1** | 0, **0.1** | 0, **0.1** |
| Epoch | **800** | 10, **15** | **500**, 600 | **10**, 15 |
| Hidden dim | 64, **256** | 64, **256** | 64, **256** | 64, **256** |
| Learning rate | 5e-3, **1e-4**, 1.5e-4 | 5e-3, 1e-4, **1.5e-4** | 5e-3, **1e-4**, 1.5e-4 | 5e-3, 1e-4, **1.5e-4** |
| LR scheduler | **ReduceLROnPlateau** | - | **ReduceLROnPlateau** | - |
| Optimizer | **Adam**, AdamW | **Adam**, AdamW | **Adam**, AdamW | **Adam**, AdamW |
| Weight decay | 1e-7, **1e-6**, 1e-5, 1e-4 | **1e-6** | **1e-6** | **1e-6** |
| **Number of clusters** | | | | |
| Protein | 156 | 372 | 362 | 312 |
| Ligand | 28 | 25 | 24 | 49 |
| **Number of layers** | | | | |
| Message Passing | 1, 2, **3** | 1, 2, **3** | 1, 2, **3** | 1, 2, **3** |
| Diffentiable pooling | **1**, 2 | **1**, 2 | **1**, 2 | **1**, 2 |
| Prediction MLP | 1, **2**, 3 | 1, **2**, 3 | 1, **2**, 3 | 1, **2**, 3 |

## A.5  STATISTICS OF THE DATASETS AND EVALUDATION SCHEMES

Table A2 provides detailed statistics of the datasets used for cross-dataset evaluation, diverse protein evaluation (LBA 30%, LBA 60%), and ligand efficacy prediction (LEP). The table summarizes the total number of complexes, as well as the quartiles (Q1-Q4), averages, and standard deviations for the number of atoms in proteins, ligands, and overall complexes across these datasets.

For cross-dataset evaluation, the PDBbind v2016 general set is used as the training and validation dataset, while the v2013 core set, v2016 core set, and v2019 holdout set serve as test datasets. Among these test sets, the v2019 holdout set contains the largest and most diverse complexes, with an average of 191.36 atoms per complex and a standard deviation of 48.31, indicating a wide variety in protein-ligand sizes.

For diverse protein evaluation, the PDBbind v2019 refined set is used, with the LBA 30% and LBA 60% datasets split based on protein sequence identity thresholds of 30% and 60%, respectively. These datasets, along with LEP dataset consist of larger and more diverse ligands and proteins compared to the cross-dataset evaluation sets. For example, the average number of atoms in protein structures in the LBA 30% and LBA 60% dataset is 371.63, and in the LEP dataset, it's 327.96, reflecting

the complex and varied nature of these datasets. These characteristics highlight the challenging and comprehensive nature of the evaluation benchmarks used to assess CheapNet's performance.

**Ligand Binding Affinity / Cross-data Evaluation - Test Dataset**   To ensure fair comparisons across all 19 models, including CheapNet, we adopted the same test datasets as described in GIGN (Yang et al., 2023):

- **PDB v2013 core set** (N=107)
- **PDB v2016 core set** (N=285)
- **PDB v2019 holdout set** (N=,4366)

These datasets were consistently used to evaluate all models' generalization capabilities, with differences in training and validation datasets depending on the baseline model's protocol.

**Ligand Binding Affinity / Cross-data Evaluation -Training and Validation Details**

1. **GIGN (Yang et al., 2023) Protocol (16 Models, including CheapNet)**
    - **Training Set:** 11,904 samples from the PDBbind v2016 general set.
    - **Validation Set:** 1,000 samples from the PDBbind v2016 general set.

    All 16 models followed the protocol established in GIGN for training, validation, and testing.

2. **CAPLA (Jin et al., 2023), GAABind (Tan et al., 2024), DEAttentionDTA (Chen et al., 2024)** These models provided pre-trained checkpoints based on training and validation datasets derived from the PDBbind v2020 general set (CAPLA: v2016 general + refined sets). Their test evaluations included only the PDB v2013 and PDB v2016 core sets, as their respective original papers limited comparisons to these benchmarks.

**Ligand Binding Affinity / Diverse Protein Evaluation**   We carefully reviewed the original papers and datasets to ensure consistency in evaluation protocols, as the following two steps:

**1) Dependency Mapping.** Previous studies are mentioned while citing two main references: HoloProt (Somnath et al., 2021) and Atom3D Townshend et al. (2021).

- *ProNet* (Wang et al., 2023) references the dataset protocol from *HoloProt* (Somnath et al., 2021).
- *ProFSA* (Gao et al., 2023), *BindNet* (Feng et al., 2024), and *GET* (Kong et al., 2024) follow the dataset splits established by *Atom3D* (Townshend et al., 2021)

**2) Dataset Consistency.** To verify dataset consistency, we compared datasets using public repositories provided by HoloProt and Atom3D, and we found that the datasets are identical:

- Sequence identity 30%:
    - **Training Set:** 3,507 samples
    - **Validation Set:** 466 samples
    - **Test Set:** 490 samples
- Sequence identity 60%:
    - **Training Set:** 3,563 samples
    - **Validation Set:** 448 samples
    - **Test Set:** 452 samples

**Baseline Results**   The baseline results were directly adopted from the following sources:

- **HoloProt** (Somnath et al., 2021): DeepDTA, SSA, TAPE, IEConv, MaSIF, Holoprot-Full Surface, Holoprot-Superpixel, ProtTrans
- **Atom3D** (Townshend et al., 2021): Atom3D-3DCNN, Atom3D-ENN, Atom3D-GNN

- **ProNet** (Wang et al., 2023): ProNet-Amino Acid, ProNet-Backbone, ProNet-All-Atom
- **ProFSA** (Gao et al., 2023): EGNN-PLM, ProFSA
- **BindNet** (Feng et al., 2024): DeepAffinity, SMT-DTA, GeoSSL, Uni-Mol, BindNet
- **GET** (Kong et al., 2024) : SchNet, GemNet, Equiformer, TorchMD-Net, MACE, LEFT-Net, GET

**Ligand Efficacy Prediction**    All 17 models were evaluated using the same training, validation, and test splits defined in the Atom3D benchmark (Townshend et al., 2021)

**Baseline Results.** The baseline results were directly adopted from the following sources:

- **Atom3D** (Townshend et al., 2021): Atom3D-3DCNN, Atom3D-ENN, Atom3D-GNN
- **ProNet** (Wang et al., 2023): GVP-GNN, ProNet-Amino Acid, ProNet-Backbone, ProNet-All-Atom
- **ProFSA** (Gao et al., 2023): ProFSA
- **BindNet** (Feng et al., 2024): DeepDTA, GeoSSL, Uni-Mol, BindNet
- **GET** (Kong et al., 2024): SchNet, EGNN, TorchMD-Net, GET

Table A2: Dataset statistics for cross-dataset evaluation, diverse protein evaluation (LBA 30%, LBA 60%), and LEP. The table summarizes total number of complexes, as well as Q1-Q4 quantiles, averages, and standard deviations for the number of atoms in proteins, ligands, and complexes across the datasets.

| Statistics | | Cross-dataset | | | | Diverse protein | | LEP |
|---|---|---|---|---|---|---|---|---|
| | | v2016 general set | v2013 core set | v2016 core set | v2019 holdout set | LBA 30% | LBA 30% | |
| **# of complex** | | 12904 | 107 | 285 | 4366 | 4463 | | 518 |
| **Protein Atom #** | Q1 | 130 | 129 | 126 | 127 | 260 | | 282 |
| | Q2 | 156 | 159 | 152 | 153.5 | 360 | | 325 |
| | Q3 | 186 | 186.5 | 178 | 183 | 462 | | 372 |
| | Q4 | 500 | 280 | 280 | 454 | 1021 | | 650 |
| | Avg | 160.84 | 160.87 | 153.53 | 157.77 | 371.63 | | 327.96 |
| | Std | 47.78 | 41.27 | 38.20 | 48.34 | 139.48 | | 71.25 |
| **Ligand Atom #** | Q1 | 20 | 16 | 17 | 21 | 17 | | 42 |
| | Q2 | 28 | 24 | 23 | 28 | 24 | | 51 |
| | Q3 | 37 | 31 | 30 | 37 | 32 | | 59 |
| | Q4 | 177 | 67 | 67 | 161 | 71 | | 147 |
| | Avg | 32.65 | 25.41 | 24.55 | 33.59 | 25.43 | | 51.47 |
| | Std | 21.61 | 11.20 | 9.81 | 21.97 | 11.24 | | 15.24 |
| **Complex Atom #** | Q1 | 154 | 149.5 | 147 | 153 | 286 | | 324 |
| | Q2 | 186 | 186 | 173 | 184 | 383 | | 378 |
| | Q3 | 224 | 218 | 205 | 220 | 488 | | 430 |
| | Q4 | 595 | 332 | 332 | 533 | 1085 | | 796 |
| | Avg | 193.50 | 186.28 | 178.08 | 191.36 | 397.06 | | 379.43 |
| | Std | 60.51 | 49.34 | 45.10 | 61.29 | 145.22 | | 83.21 |

## A.6    PERFORMANCE OF CHEAPNET ON LBA TASKS WITH PARAMETER COUNTS AND STANDARD DEVIATIONS

Tables A3 and A4 provide a detailed breakdown of the parameter counts and standard deviations for all models evaluated on the LBA tasks in both the cross-dataset and diverse protein evaluations. These tables reinforce the efficiency of CheapNet, as it consistently delivers superior performance with a significantly smaller parameter count compared to other models, especially pre-trained models like BindNet, which utilize orders of magnitude more parameters.

Moreover, CheapNet demonstrates smaller standard deviations in its predictions across all datasets, indicating greater stability and reliability. This consistency is particularly noteworthy given that CheapNet does not rely on large-scale pre-training, further emphasizing its robustness in handling diverse protein-ligand interactions. These findings affirm that CheapNet achieves state-of-the-art performance with a reasonable computational footprint, making it a highly practical and effective solution for protein-ligand binding affinity prediction tasks.

To measure the statistical significance of performance differences between models, we used Z-tests, as paired t-tests were not feasible due to relying on reported results from previous studies. With the available means, standard deviations, and sample sizes, Z-tests provided a suitable alternative.

We compared CheapNet's performance against the second-best model in terms of RMSE, Pearson correlation coefficient, and Spearman correlation coefficient on the LBA task. The p-values corresponding to the Z-statistics are indicated at the end of the table. The results show that CheapNet's improvements over the second-best model are statistically significant (p-value $< 0.001$) or comparable.

Table A3: Performance comparison of CheapNet and baselines with parameter counts and standard deviations on the cross-dataset evaluation. The top results are shown in **bold**, and the second-best are underlined, respectively.

| Model | Params # | v2013 core set | | v2016 core set | | v2019 holdout set | |
|---|---|---|---|---|---|---|---|
| | | RMSE ↓ | Pearson ↑ | RMSE ↓ | Pearson ↑ | RMSE ↓ | Pearson ↑ |
| **Interaction-free** | | | | | | | |
| DeepDTA (Öztürk et al., 2018) | 1.93M | $1.639 \pm 0.026$ | $0.718 \pm 0.014$ | $1.357 \pm 0.015$ | $0.785 \pm 0.007$ | $1.485 \pm 0.023$ | $0.586 \pm 0.012$ |
| GraphDTA-GCN (Nguyen et al., 2021) | 2.06M | $1.749 \pm 0.062$ | $0.662 \pm 0.032$ | $1.513 \pm 0.048$ | $0.719 \pm 0.023$ | $1.763 \pm 0.039$ | $0.439 \pm 0.021$ |
| GraphDTA-GAT (Nguyen et al., 2021) | 1.46M | $2.043 \pm 0.029$ | $0.476 \pm 0.022$ | $1.748 \pm 0.019$ | $0.594 \pm 0.010$ | $1.663 \pm 0.027$ | $0.432 \pm 0.016$ |
| GraphDTA-GIN (Nguyen et al., 2021) | 1.30M | $1.691 \pm 0.124$ | $0.694 \pm 0.059$ | $1.470 \pm 0.065$ | $0.743 \pm 0.027$ | $1.676 \pm 0.032$ | $0.472 \pm 0.021$ |
| GraphDTA-GAT-GCN (Nguyen et al., 2021) | 4.75M | $1.645 \pm 0.085$ | $0.711 \pm 0.036$ | $1.434 \pm 0.064$ | $0.754 \pm 0.025$ | $1.705 \pm 0.075$ | $0.474 \pm 0.028$ |
| MGraphDTA (Yang et al., 2022) | 3.05M | $1.680 \pm 0.093$ | $0.696 \pm 0.046$ | $1.439 \pm 0.047$ | $0.753 \pm 0.022$ | $1.553 \pm 0.028$ | $0.538 \pm 0.013$ |
| **Interaction-based** | | | | | | | |
| Pafnucy (Stepniewska-Dziubinska et al., 2018) | - | $1.517 \pm 0.014$ | $0.783 \pm 0.005$ | $1.450 \pm 0.047$ | $0.769 \pm 0.019$ | $1.438 \pm 0.016$ | $0.612 \pm 0.014$ |
| OnionNet (Zheng et al., 2019) | 1.80M | $1.583 \pm 0.079$ | $0.741 \pm 0.037$ | $1.399 \pm 0.076$ | $0.770 \pm 0.027$ | $1.510 \pm 0.034$ | $0.573 \pm 0.014$ |
| PotentialNet (Feinberg et al., 2018) | 0.08M | $1.607 \pm 0.027$ | $0.773 \pm 0.010$ | $1.503 \pm 0.033$ | $0.772 \pm 0.007$ | $1.514 \pm 0.028$ | $0.564 \pm 0.014$ |
| SchNet (Schütt et al., 2017) | 0.28M | $1.570 \pm 0.029$ | $0.754 \pm 0.030$ | $1.390 \pm 0.023$ | $0.787 \pm 0.016$ | $1.522 \pm 0.071$ | $0.560 \pm 0.028$ |
| GNN-DTI (Lim et al., 2019) | 0.22M | $1.533 \pm 0.084$ | $0.767 \pm 0.040$ | $1.384 \pm 0.013$ | $0.779 \pm 0.008$ | $1.446 \pm 0.006$ | $0.614 \pm 0.007$ |
| IGN (Jiang et al., 2021) | 1.66M | $1.428 \pm 0.020$ | $0.807 \pm 0.001$ | $1.269 \pm 0.030$ | $0.821 \pm 0.013$ | $1.410 \pm 0.015$ | $0.630 \pm 0.008$ |
| EGNN (Satorras et al., 2021) | 1.59M | $1.498 \pm 0.025$ | $0.782 \pm 0.015$ | $1.289 \pm 0.021$ | $0.816 \pm 0.011$ | $1.399 \pm 0.013$ | $0.628 \pm 0.010$ |
| GIGN (Yang et al., 2023) | 0.62M | $\underline{1.380 \pm 0.009}$ | $\underline{0.821 \pm 0.003}$ | $\underline{1.190 \pm 0.017}$ | $\underline{0.840 \pm 0.007}$ | $\underline{1.393 \pm 0.007}$ | $\underline{0.641 \pm 0.006}$ |
| **Interaction-based (attention mechanism)** | | | | | | | |
| AttentionSiteDTI (Yazdani-Jahromi et al., 2022) | 42.66M | $1.444 \pm 0.037$ | $0.792 \pm 0.014$ | $1.352 \pm 0.022$ | $0.784 \pm 0.008$ | $1.539 \pm 0.015$ | $0.563 \pm 0.004$ |
| CAPLA (Jin et al., 2023) | 0.31M | $1.409$ | $0.816$ | $1.206$ | $\underline{0.841}$ | - | - |
| GAABind (Tan et al., 2024) | 17.95M | $1.488$ | $0.772$ | $1.297$ | $0.803$ | - | - |
| DEAttentionDTA (Chen et al., 2024) | 2.32M | $1.470$ | $0.800$ | $1.266$ | $0.827$ | - | - |
| **Interaction-based (cluster-attention mechanism)** | | | | | | | |
| CheapNet (ours) | 1.33M | $\mathbf{1.262 \pm 0.017}$ | $\mathbf{0.857 \pm 0.004}$ | $\mathbf{1.107 \pm 0.011}$ | $\mathbf{0.870 \pm 0.002}$ | $\mathbf{1.343 \pm 0.007}$ | $\mathbf{0.665 \pm 0.003}$ |
| Statistical Significance (p-value) | | *** | *** | *** | ***$^a$ | *** | *** |

$^a$ Statistical test was performed assuming zero standard due to the unavailability of the standard deviation.
*** : p-value $< 0.001$

## A.7 PERFORMANCE OF CHEAPNET ON LEP TASK WITH PARAMETER COUNTS AND STANDARD DEVIATIONS

Table A5 provides a comparison of CheapNet and baseline models on the LEP task, including standard deviations for AUROC and AUPRC metrics. Notably, only GeoSSL, Uni-Mol, and ProFSA report standard deviations based on repeated experiments. Consistent with its performance on the LBA tasks, CheapNet demonstrates smaller standard deviations compared to other methods, reflecting its stable and reliable performance across multiple runs. These results further emphasize CheapNet's efficiency and robustness in modeling ligand efficacy, making it an effective solution for capturing complex protein-ligand interactions. Z-tests were also performed as in Appendix A.6 to compare CheapNet's performance against the second-best performing model in terms of AUROC and AURPC on the LEP task. The results demonstrate that CheapNet's improvement over the second-best model is statistically significant (p-value $< 0.001$).

## A.8 ABLATION STUDIES (1): ADAPTABILITY OF CHEAPNET WITH PARAMETER COUNTS AND STANDARD DEVIATIONS

Table A6 presents the detailed results of the ablation study with standard deviations for RMSE and Pearson correlation coefficients across the PDBbind v2013 core set, v2016 core set, and v2019 holdout set. Despite only a modest increase in the number of parameters, CheapNet combined with various graph encoders (GCN, EGNN, and GIGN) demonstrates consistent and substantial performance improvements across all datasets. This highlights CheapNet's ability to enhance predictive accuracy effectively while maintaining parameter efficiency, making it a scalable choice for protein-ligand binding affinity tasks.

Table A4: Performance comparison of CheapNet and baselines with parameter counts and standard deviations on the diverse protein evaluation. The top results are shown in **bold**, and the second-best are underlined, respectively.

| Model | Params # | LBA 30% | | | LBA 60% | | |
|---|---|---|---|---|---|---|---|
| | | RMSE ↓ | Pearson ↑ | Spearman ↑ | RMSE ↓ | Pearson ↑ | Spearman ↑ |
| **Interaction-free** | | | | | | | |
| DeepDTA (Öztürk et al., 2018) | 1.93M | $1.866 \pm 0.080$ | $0.472 \pm 0.022$ | $0.471 \pm 0.024$ | $1.762 \pm 0.261$ | $0.666 \pm 0.012$ | $0.663 \pm 0.015$ |
| SSA (Bepler & Berger, 2019) | 48.8M | $1.985 \pm 0.006$ | $0.165 \pm 0.006$ | $0.152 \pm 0.024$ | $1.891 \pm 0.004$ | $0.249 \pm 0.006$ | $0.275 \pm 0.008$ |
| TAPE (Rao et al., 2019) | 93.0M | $1.890 \pm 0.035$ | $0.338 \pm 0.044$ | $0.286 \pm 0.124$ | $1.633 \pm 0.016$ | $0.568 \pm 0.033$ | $0.571 \pm 0.021$ |
| **Interaction-based** | | | | | | | |
| Atom3D-3DCNN (Townshend et al., 2020) | 2.22M | $1.416 \pm 0.021$ | $0.550 \pm 0.021$ | $0.553 \pm 0.009$ | $1.621 \pm 0.025$ | $0.608 \pm 0.020$ | $0.615 \pm 0.028$ |
| Atom3D-ENN (Townshend et al., 2020) | 0.06M | $1.568 \pm 0.012$ | $0.389 \pm 0.024$ | $0.408 \pm 0.021$ | $1.620 \pm 0.049$ | $0.623 \pm 0.015$ | $0.633 \pm 0.021$ |
| Atom3D-GNN (Townshend et al., 2020) | 0.38M | $1.601 \pm 0.048$ | $0.545 \pm 0.027$ | $0.533 \pm 0.033$ | $1.408 \pm 0.069$ | $0.743 \pm 0.022$ | $0.743 \pm 0.027$ |
| IEConv (Hermosilla et al., 2021) | 5.80M | $1.554 \pm 0.016$ | $0.414 \pm 0.053$ | $0.428 \pm 0.032$ | $1.473 \pm 0.024$ | $0.667 \pm 0.011$ | $0.675 \pm 0.019$ |
| MaSIF (Gainza et al., 2020) | 0.62M | $1.484 \pm 0.018$ | $0.467 \pm 0.020$ | $0.455 \pm 0.014$ | $1.426 \pm 0.017$ | $0.709 \pm 0.001$ | $0.701 \pm 0.001$ |
| Holoprot-Full Surface (Somnath et al., 2021) | 1.44M | $1.464 \pm 0.006$ | $0.509 \pm 0.002$ | $0.500 \pm 0.005$ | $1.365 \pm 0.038$ | $0.749 \pm 0.014$ | $0.742 \pm 0.011$ |
| Holoprot-Superpixel (Somnath et al., 2021) | 1.76M | $1.491 \pm 0.004$ | $0.491 \pm 0.014$ | $0.482 \pm 0.032$ | $1.416 \pm 0.022$ | $0.724 \pm 0.011$ | $0.715 \pm 0.006$ |
| ProNet-Amino Acid (Wang et al., 2023) | 1.38M | $1.455 \pm 0.009$ | $0.536 \pm 0.012$ | $0.526 \pm 0.012$ | $1.397 \pm 0.018$ | $0.741 \pm 0.008$ | $0.734 \pm 0.009$ |
| ProNet-Backbone (Wang et al., 2023) | 1.39M | $1.458 \pm 0.003$ | $0.546 \pm 0.007$ | $0.550 \pm 0.008$ | $1.349 \pm 0.019$ | $0.764 \pm 0.006$ | $0.759 \pm 0.001$ |
| ProNet-All-Atom (Wang et al., 2023) | 1.39M | $1.463 \pm 0.001$ | $0.551 \pm 0.005$ | $0.551 \pm 0.008$ | $1.343 \pm 0.025$ | $0.765 \pm 0.009$ | $0.761 \pm 0.003$ |
| SchNet (Schütt et al., 2017)[a] | 0.21M | $1.370 \pm 0.028$ | $0.590 \pm 0.017$ | $0.571 \pm 0.028$ | - | - | - |
| GemNet (Gasteiger et al., 2021)[a] | 1.37M | | **OOM** | | - | - | - |
| Equiformer (Liao & Smidt, 2022)[a] | 1.10M | | **OOM** | | - | - | - |
| TorchMD-Net (Thölke & De Fabritiis, 2022)[a] | 0.30M | $1.383 \pm 0.009$ | $0.580 \pm 0.008$ | $0.564 \pm 0.004$ | - | - | - |
| MACE (Batatia et al., 2022)[a] | 3.91M | $1.372 \pm 0.021$ | $0.612 \pm 0.010$ | $0.592 \pm 0.010$ | - | - | - |
| LEFTNet (Du et al., 2024)[a] | 0.85M | $1.366 \pm 0.016$ | $0.592 \pm 0.014$ | $0.580 \pm 0.011$ | - | - | - |
| GET (Kong et al., 2024) | 0.69M | $\underline{1.327 \pm 0.005}$ | $0.620 \pm 0.004$ | $0.611 \pm 0.003$ | - | - | - |
| **Pre-training** | | | | | | | |
| DeepAffinity (Karimi et al., 2019) | - | $1.893 \pm 0.650$ | 0.415 | 0.426 | - | - | - |
| SMT-DTA (Pei et al., 2022) | - | 1.574 | 0.458 | 0.447 | 1.347 | 0.758 | 0.754 |
| GeoSSL (Liu et al., 2022) | - | $1.451 \pm 0.030$ | $0.577 \pm 0.020$ | $0.572 \pm 0.010$ | - | - | - |
| ProtTrans (Elnaggar et al., 2021) | 2.4M | $1.544 \pm 0.015$ | $0.438 \pm 0.053$ | $0.434 \pm 0.058$ | $1.641 \pm 0.016$ | $0.595 \pm 0.014$ | $0.588 \pm 0.009$ |
| EGNN-PLM (Wu et al., 2023) | 650M | $1.403 \pm 0.010$ | $0.565 \pm 0.020$ | $0.544 \pm 0.010$ | $1.559 \pm 0.020$ | $0.644 \pm 0.020$ | $0.646 \pm 0.020$ |
| Uni-Mol (Zhou et al., 2023) | 47.61M | 1.434 | 0.565 | 0.540 | 1.357 | 0.753 | 0.750 |
| ProFSA (Gao et al., 2023) | >47.61M[b] | $1.377 \pm 0.010$ | $0.628 \pm 0.010$ | $0.620 \pm 0.010$ | $1.377 \pm 0.010$ | $0.764 \pm 0.000$ | $0.762 \pm 0.010$ |
| BindNet (Feng et al., 2024) | >47.61M[b] | 1.340 | $\underline{0.632}$ | $\underline{0.620}$ | **1.230** | $\underline{0.793}$ | $\underline{0.788}$ |
| **Interaction-based (cluster-attention mechanism)** | | | | | | | |
| CheapNet (ours) | 1.39M | $\mathbf{1.311 \pm 0.003}$ | $\mathbf{0.642 \pm 0.001}$ | $\mathbf{0.639 \pm 0.010}$ | $\underline{1.238 \pm 0.005}$ | $\mathbf{0.794 \pm 0.002}$ | $\mathbf{0.789 \pm 0.001}$ |
| Statistical Significance (p-value) | | *** | ***[c] | ***[c] | ns | ns | * |

[a] Adapted from GET (Kong et al., 2024), which used hierarchical approaches from atom-level to block-level.
[b] Accurate parameter estimation for BindNet is not possible due to the unavailability of the pre-training model checkpoint, but it is likely higher than Uni-Mol since it is based on Uni-Mol.
[c] Statistical test was performed assuming zero standard due to the unavailability of the standard deviation.
[*] : p-value $< 0.05$
[**] : p-value $< 0.01$
[***] : p-value $< 0.001$
[ns] : non-significant

## A.9  ABLATION STUDIES (2): HIERARCHICAL REPRESENTATIONS AND ATTENTION MECHANISMS WITH PARAMETER COUNTS AND STANDARD DEVIATIONS

Table A7 shows CheapNet's performance under various configurations for RMSE and Pearson correlation coefficients across the PDBbind v2013 core set, v2016 core set, and v2019 holdout set, showing the effects of hierarchical representations and cross-attention mechanisms. The results show that the combination of cluster-level representations and cross-attention yields the best performance, highlighting the significant improvements achieved by integrating both components.

## A.10  ABLATION STUDIES (3): EFFECTS OF NUMBER OF CLUSTERS

We investigate how varying the number of clusters, defined by quantiles of the number of complex nodes (Q1–Q4), affects CheapNet's performance. The model uses a differentiable pooling mechanism to cluster atoms based on functional similarity, and the number of clusters can influence both accuracy and efficiency. To evaluate this, we experimented with four different cluster sizes (Q1, Q2, Q3, Q4) and summarized the results in Table A8.

The results show that cluster size q2 consistently provides the best balance between RMSE and Pearson correlation coefficient across the datasets. While q4 shows slightly better results on the v2019 holdout set, q2 performs optimally on the v2013 and v2016 core sets. This suggests that q2 strikes the best balance between computational efficiency and capturing key molecular interactions, making it the most suitable cluster size for most applications in CheapNet.

Table A5: Comparison results of CheapNet and baselines on LEP datasets. The top results are shown in **bold**, and the second-best are underlined, respectively.

| Model | Params # | AUROC ↑ | AUPRC ↑ |
|---|---|---|---|
| **Interaction-free** | | | |
| DeepDTA (Öztürk et al., 2018) | - | 0.696 | - |
| **Interaction-based** | | | |
| Atom3D-3DCNN (Townshend et al., 2020) | 97.51M | 0.589 | 0.483 |
| Atom3D-ENN (Townshend et al., 2020) | - | 0.663 | 0.551 |
| Atom3D-GNN (Townshend et al., 2020) | 1.21M | 0.681 | 0.598 |
| GVP-GNN (Jing et al., 2021) | - | 0.628 | - |
| ProNet-Amino Acid (Wang et al., 2023) | - | 0.646 | - |
| ProNet-Backbone (Wang et al., 2023) | - | 0.687 | - |
| ProNet-All-Atom (Wang et al., 2023) | - | 0.692 | - |
| SchNet (Schütt et al., 2017)[a] | 0.20M | 0.736 ± 0.020 | 0.731 ± 0.048 |
| EGNN (Satorras et al., 2021)[a] | 0.17M | 0.724 ± 0.027 | 0.720 ± 0.056 |
| TorchMD-NET (Thölke & De Fabritiis, 2022)[a] | 0.29M | 0.717 ± 0.033 | 0.724 ± 0.055 |
| GET (Kong et al., 2024) | 1.60M | 0.761 ± 0.012 | 0.751 ± 0.012 |
| **Pre-training** | | | |
| GeoSSL (Liu et al., 2022) | - | 0.776 ± 0.030 | 0.694 ± 0.060 |
| Uni-Mol (Zhou et al., 2023) | 47.61M | 0.782 ± 0.020 | 0.695 ± 0.070 |
| ProFSA (Gao et al., 2023) | >47.61M[b] | 0.840 ± 0.040 | 0.806 ± 0.040 |
| BindNet (Feng et al., 2024) | >47.61M[b] | 0.882 | 0.870 |
| **Interaction-based (cluster-attention)** | | | |
| CheapNet (ours) | 1.45M | **0.935 ± 0.002** | **0.924 ± 0.000** |
| Statistical Significance (p-value) | | ***[c] | ***[c] |

[a] Adapted from GET (Kong et al., 2024), which used hierarchical approaches from atom-level to block-level.
[b] Accurate parameter estimation for BindNet is not possible due to the unavailability of the pre-training model checkpoint, but it is likely higher than Uni-Mol since it is based on Uni-Mol.
[c] Statistical test was performed assuming zero standard due to the unavailability of the standard deviation.
*** : p-value < 0.001

Table A6: Ablation study results showing RMSE, Pearson correlation coefficient, and performance improvement ($\Delta$) for different graph encoders with parameter counts and standard deviations on the PDBbind v2013 core set, v2016 core set, and v2019 holdout set. The top results are shown in **bold**, and the second-best are underlined, respectively.

| Model | Params # | v2013 core set | | v2016 core set | | v2019 holdout set | |
|---|---|---|---|---|---|---|---|
| | | RMSE ↓ | Pearson ↑ | RMSE ↓ | Pearson ↑ | RMSE ↓ | Pearson ↑ |
| GCN (Kipf & Welling, 2016) | 0.25M | 1.395 ± 0.033 | 0.819 ± 0.013 | 1.295 ± 0.014 | 0.809 ± 0.004 | 1.460 ± 0.009 | 0.593 ± 0.002 |
| CheapNet-GCN | 1.09M | 1.368 ± 0.039 | 0.820 ± 0.012 | 1.246 ± 0.034 | 0.823 ± 0.014 | 1.391 ± 0.018 | 0.635 ± 0.010 |
| $\Delta$(%) | | +1.935 | +0.122 | +3.784 | +1.731 | +4.726 | +7.083 |
| EGNN (Satorras et al., 2021) | 1.59M | 1.498 ± 0.025 | 0.782 ± 0.015 | 1.289 ± 0.021 | 0.816 ± 0.011 | 1.399 ± 0.013 | 0.628 ± 0.010 |
| CheapNet-EGNN | 2.43M | 1.321 ± 0.027 | 0.843 ± 0.012 | 1.161 ± 0.010 | 0.856 ± 0.000 | 1.343 ± 0.009 | 0.664 ± 0.004 |
| $\Delta$(%) | | +11.816 | +7.801 | +9.930 | +5.549 | +4.003 | +5.732 |
| GIGN (Yang et al., 2023) | 0.62M | 1.380 ± 0.009 | 0.821 ± 0.003 | 1.190 ± 0.017 | 0.840 ± 0.007 | 1.393 ± 0.007 | 0.641 ± 0.006 |
| CheapNet-GIGN | 1.33M | 1.262 ± 0.017 | 0.857 ± 0.004 | 1.107 ± 0.011 | 0.870 ± 0.002 | 1.343 ± 0.007 | 0.665 ± 0.003 |
| $\Delta$(%) | | +8.551 | +4.385 | +6.975 | +3.571 | +3.589 | +3.744 |

Table A7: Ablation study results for the effect of using hierarchical representations (Hierarchical), and type of attention mechanism (Attention) with parameter counts and standard deviations on the PDBbind v2013 core set, v2016 core set, and v2019 holdout set. The top results are shown in **bold**, and the second-best are underlined, respectively.

| Hierarchical | Attention | Params # | v2013 core set | | v2016 core set | | v2019 holdout set | |
|---|---|---|---|---|---|---|---|---|
| | | | RMSE ↓ | Pearson ↑ | RMSE ↓ | Pearson ↑ | RMSE ↓ | Pearson ↑ |
| ✗ | ✗ | 0.49M | 1.345 ± 0.017 | 0.844 ± 0.003 | 1.189 ± 0.005 | 0.851 ± 0.001 | 1.360 ± 0.001 | 0.652 ± 0.001 |
| ✗ | Self | 1.02M | 1.305 ± 0.030 | 0.850 ± 0.004 | 1.166 ± 0.003 | 0.854 ± 0.002 | 1.367 ± 0.003 | 0.650 ± 0.002 |
| ✗ | Cross | 1.02M | 1.293 ± 0.022 | 0.853 ± 0.002 | 1.151 ± 0.003 | 0.857 ± 0.001 | 1.362 ± 0.011 | 0.653 ± 0.004 |
| ✓ | ✗ | 0.81M | 1.330 ± 0.017 | 0.840 ± 0.006 | 1.161 ± 0.012 | 0.853 ± 0.002 | 1.348 ± 0.005 | 0.662 ± 0.005 |
| ✓ | Self | 1.33M | 1.327 ± 0.044 | 0.841 ± 0.015 | 1.168 ± 0.003 | 0.853 ± 0.001 | 1.348 ± 0.004 | 0.662 ± 0.002 |
| ✓ | Cross | 1.33M | **1.262 ± 0.017** | **0.857 ± 0.004** | **1.107 ± 0.011** | **0.870 ± 0.002** | **1.343 ± 0.007** | **0.665 ± 0.003** |

## A.11 ABLATION STUDIES (4): EFFECTS OF ADDITIONAL AUXILIARY LOSS

We considered incorporating auxiliary losses, including a link prediction loss and an entropy regularization loss, as proposed by Ying et al. (2018). The link prediction loss is defined as $L_{LP} = \|A - SS^T\|_F$, where $\| \cdot \|$ means Frobenius norm. The entropy regularization loss is

Table A8: Ablation study results of CheapNet for different number of clusters with parameter counts and standard deviations on the PDBbind v2013 core set, v2016 core set, and v2019 holdout set. The top results are shown in **bold**, and the second-best are underlined, respectively.

| Cluster size | Params # | v2013 core set | | v2016 core set | | v2019 holdout set | |
|---|---|---|---|---|---|---|---|
| | | RMSE ↓ | Pearson ↑ | RMSE ↓ | Pearson ↑ | RMSE ↓ | Pearson ↑ |
| Q1 | 1.32M | $1.303 \pm 0.016$ | $0.850 \pm 0.005$ | $1.153 \pm 0.015$ | $0.856 \pm 0.004$ | $1.356 \pm 0.009$ | $0.657 \pm 0.006$ |
| **Q2** | 1.33M | **$1.262 \pm 0.017$** | $\underline{0.857 \pm 0.004}$ | **$1.107 \pm 0.011$** | **$0.870 \pm 0.002$** | $1.343 \pm 0.007$ | $0.665 \pm 0.003$ |
| Q3 | 1.34M | $\underline{1.274 \pm 0.029}$ | **$0.862 \pm 0.011$** | $\underline{1.142 \pm 0.025}$ | $0.863 \pm 0.007$ | $\underline{1.340 \pm 0.008}$ | $0.665 \pm 0.004$ |
| Q4 | 1.46M | $1.314 \pm 0.032$ | $0.847 \pm 0.011$ | $1.147 \pm 0.002$ | $0.859 \pm 0.002$ | **$1.334 \pm 0.009$** | **$0.669 \pm 0.005$** |

Table A9: Ablation study results of CheapNet for auxiliary loss with standard deviations on the PDBbind v2013 core set, v2016 core set, and v2019 holdout set. The top results are shown in **bold**, and the second-best are underlined, respectively.

| auxilliary loss | v2013 core set | | v2016 core set | | v2019 holdout set | |
|---|---|---|---|---|---|---|
| | RMSE ↓ | Pearson ↑ | RMSE ↓ | Pearson ↑ | RMSE ↓ | Pearson ↑ |
| ✗ | **$1.262 \pm 0.017$** | $\underline{0.857 \pm 0.004}$ | **$1.107 \pm 0.011$** | **$0.870 \pm 0.002$** | **$1.343 \pm 0.007$** | **$0.665 \pm 0.003$** |
| ✓ | $\underline{1.288 \pm 0.017}$ | **$0.858 \pm 0.003$** | $\underline{1.143 \pm 0.007}$ | $\underline{0.861 \pm 0.004}$ | $\underline{1.348 \pm 0.002}$ | $\underline{0.661 \pm 0.001}$ |

Table A10: Ablation study results for the effect of using various pooling methods compared to the differential pooling of CheapNet with parameter counts and standard deviations on the LBA 30% dataset of Diverse Protein Evaluation.

| Model | Params # | RMSE ↓ | Pearson ↑ | Spearman ↑ |
|---|---|---|---|---|
| TopKPooling (Gao & Ji, 2019) | 1.03M | $1.478 \pm 0.048$ | $0.578 \pm 0.013$ | $0.574 \pm 0.030$ |
| ASAPooling (Ranjan et al., 2020) | 1.16M | $1.419 \pm 0.040$ | $0.592 \pm 0.017$ | $0.594 \pm 0.020$ |
| SAGPooling (Gao et al., 2023) | 1.03M | $1.514 \pm 0.020$ | $0.582 \pm 0.013$ | $0.590 \pm 0.007$ |
| CheapNet (ours) | 1.39M | $1.311 \pm 0.003$ | $0.642 \pm 0.001$ | $0.639 \pm 0.010$ |

given by $L_E = \frac{1}{n} \sum_{i=1}^{n} H(S_i)$, where $H$ is the entropy function, and $S_i$ represents the $i$-th row of $S$.

As shown in Table A9, the use of auxiliary loss does not consistently improve performance. While it provides a marginal boost in Pearson correlation coefficient on a smaller dataset such ac v2013 core set, it tends to degrade results on larger datasets. This indicates that clustering atoms based on geometric positions is less effective compared to clustering based purely on features, particularly when used with cross-attention mechanisms.

### A.12 ABLATION STUDIES (5): COMPARISON OF ATOM SELECTION & GROUPING APPROACHES AND CLUSTER-ATTENTION MECHANISM OF CHEAPNET

To further analyze the impact of soft-clustering approaches in CheapNet, we evaluate different atom selection or grouping approaches by replacing CheapNet's differential pooling method with alternative strategies. Specifically, we tested hard node selection methods such as TopKPooling (Gao & Ji, 2019) and structure-based pooling methods like ASAPooling (Ranjan et al., 2020) and SAGPooling (Lee et al., 2019). As presented in Table A10, CheapNet consistently demonstrated superior performance in the LBA 30% dataset on the Diverse Protein Evaluation. Unlike hard selection or structure-based clustering approaches, CheapNet's cluster-attention mechanism prioritizes clustering by atom embeddings, rather than relying on geometric properties, offering a complementary perspective on clustering.

### A.13 PERFORMANCE OF CHEAPNET ON CSAR NRC-HiQ DATASET WITH PARAMETER COUNTS AND STANDARD DEVIATIONS

Table A11 summarizes the detailed performance of CheapNet and various interaction-based models on the CSAR NRC-HiQ dataset, including parameter counts and standard deviations. CheapNet not only achieves the best RMSE and Pearson correlation coefficient but also exhibits the smallest standard deviations across both metrics (except RMSE of EGNN), indicating its consistent and reliable

Table A11: Performance comparison of CheapNet and various interaction-based models with parameter counts and standard deviations on the CSAR NRC-HiQ dataset. The top results are shown in **bold**, and the second-best are underlined, respectively, with standard deviations.

| Model | Params # | RMSE ↓ | Pearson ↑ |
|---|---|---|---|
| **Interaction-based** | | | |
| PotentialNet (Feinberg et al., 2018) | 0.08M | $1.730 \pm 0.119$ | $0.718 \pm 0.056$ |
| GNN-DTI (Lim et al., 2019) | 0.22M | $1.675 \pm 0.256$ | $0.855 \pm 0.123$ |
| IGN (Jiang et al., 2021) | 1.66M | $1.647 \pm 0.265$ | $0.846 \pm 0.052$ |
| EGNN (Satorras et al., 2021) | 1.59M | $1.640 \pm 0.068$ | $0.866 \pm 0.031$ |
| GIGN (Yang et al., 2023) | 0.62M | $1.827 \pm 0.166$ | $0.766 \pm 0.086$ |
| **Interaction-based (cluster-attention mechanism)** | | | |
| CheapNet | 1.33M | **$1.381 \pm 0.089$** | **$0.901 \pm 0.016$** |

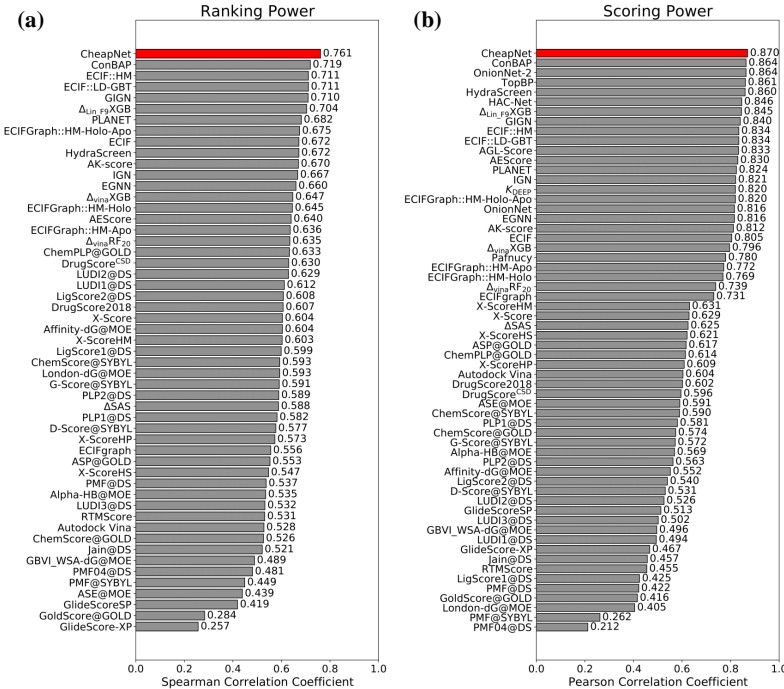

Figure A1: **Comparison of (a) ranking power and (b) scoring power of CheapNet and various other models based on Spearman correlation coefficient and Pearson correlation coefficient.**

performance. Despite using more parameters than some models, CheapNet's cluster-attention mechanism offers significant improvements, demonstrating its robustness in handling complex protein-ligand interactions.

## A.14 PERFORMANCE OF CHEAPNET ON CASF-2016 DATASET

We assess the ranking power and scoring power of CheapNet using the CASF-2016 dataset (Su et al., 2018). To evaluate ranking power, we calculate the Spearman correlation coefficient for 5 ligands, averaged over 57 complexes. As shown in Figure A1(a), CheapNet ranked first with a correlation of 0.761, outperforming all other models. For the scoring test, we assess the Pearson correlation coefficient for 285 complexes. As illustrated in Figure A1(b), CheapNet is the best-performing model among those evaluated. These excellent results demonstrate CheapNet's potential to significantly advance protein-ligand ranking in pharmaceutical research.

Table A12: Performance comparison of models with and without the cluster-attention mechanism on the DUD-E dataset. The integration of the cluster-attention mechanism significantly improves performance with fewer parameters. The top results are shown in **bold**, and the second-best are underlined, respectively.

| Model | Params # | AUROC ↑ | $EF_{0.5\%}$ ↑ | $EF_{1\%}$ ↑ | $EF_{2\%}$ ↑ | $EF_{5\%}$ ↑ |
|---|---|---|---|---|---|---|
| GCN (Kipf & Welling, 2016) | 0.08M | $0.677 \pm 0.030$ | $9.951 \pm 0.694$ | $5.062 \pm 0.353$ | $4.148 \pm 0.865$ | $3.269 \pm 0.626$ |
| EGNN (Satorras et al., 2021) | 0.03M | $0.771 \pm 0.017$ | $11.368 \pm 5.423$ | $9.202 \pm 4.079$ | $7.393 \pm 2.714$ | $5.534 \pm 1.418$ |
| GIGN (Yang et al., 2023) | 0.01M | $0.780 \pm 0.017$ | $11.079 \pm 5.019$ | $8.659 \pm 5.039$ | $7.492 \pm 3.757$ | $5.693 \pm 1.538$ |
| AttentionSiteDTI (Yazdani-Jahromi et al., 2022) | 42.59M | $\underline{0.820 \pm 0.012}$ | $\underline{13.985 \pm 7.580}$ | $\underline{11.694 \pm 7.418}$ | $\underline{9.447 \pm 5.861}$ | $\underline{6.846 \pm 2.635}$ |
| CheapNet (ours) | 0.03M | $\mathbf{0.826 \pm 0.011}$ | $\mathbf{24.646 \pm 10.922}$ | $\mathbf{16.249 \pm 7.617}$ | $\mathbf{12.549 \pm 5.032}$ | $\mathbf{8.109 \pm 2.144}$ |

## A.15 APPLICATION TO REAL-WORLD SCENARIO: EXAMPLE OF VIRTUAL SCREENING TASK

In drug discovery, accurately predicting whether a ligand will bind to a receptor protein—a process known as virtual screening—is essential. To demonstrate the effectiveness of CheapNet in capturing the relationship between a protein and a ligand, we curated a dataset from the well-established DUD-E dataset (Mysinger et al., 2012) and processed its 3D structures using RDKit. For each active ligand, pockets were extracted, and corresponding graphs were constructed. Undersampled decoys were then generated in equal numbers, resulting in 11,109 actives and 10,987 decoys. The number of target proteins is 52. With CheapNet, we compared GCN (Kipf & Welling, 2016), EGNN (Satorras et al., 2021), GIGN (Yang et al., 2023), and AttentionSiteDTI (Yazdani-Jahromi et al., 2022) as baselines. A 3-fold cross-validation was applied, and the averaged results are reported. For fair comparison, a hidden dimension of 35 is adopted.

Table A12 clearly shows that CheapNet notably enhances performances of the virtual screening task. Importantly, this improvement is achieved with greater parameter efficiency, as the cluster-attention mechanism boosts performance while reducing the parameter count.

Furthermore, to demonstrate the interpretability of CheapNet, we performed a case study using the Tyrosine Protein Kinase SRC protein (SRC) with one of its active ligands from the DUD-E dataset. SRC is a disease-causing protein that promotes the growth of cancer cells, making it an important target for cancer treatments (Luo et al., 2022).

As shown in Figure A2, CheapNet showed high confidence in the activity of the ligand with SRC, with a predicted score of 0.9998. Notably, the interpretability of CheapNet in this scenario was impressive, as it successfully pinpointed the regions of critical interaction between the ligand and the protein. Using cluster-level cross attention, CheapNet effectively identified key molecular interactions, demonstrating its potential to enhance AI-based drug development by improving both accuracy and interpretability.

## A.16 DETAILS OF MEMORY FOOTPRINT ANALYSIS

The memory usage in attention-based models is significantly influenced by the number of atoms in a protein-ligand complex, due to the quadratic complexity of attention matrices. To assess CheapNet's memory efficiency, we compared it against GAABind and DEAttentionDTA using varying numbers of atoms and batch sizes.

We divided the PDBbind v2016 core set based on the number of atoms in the complexes, ranging from 50 to 450 in increments of 50. For each interval, 128 complexes were randomly sampled to ensure fair comparison. For the smaller atom interval (50–100 atoms, "small complex"), experiments were conducted with batch sizes of 2 and 4. For the larger interval (400–450 atoms, "large complex"), batch sizes of 32, 64, and 128 were used. Although we tested a wider range of intervals and batch sizes, Figure 3 in the main text focuses on representative settings to highlight the differences in memory efficiency.

Each model was trained for 20 epochs, with memory usage monitored via nvidia-smi. CheapNet consistently required less GPU memory compared to GAABind and DEAttentionDTA, owing to its cluster-attention mechanism, which reduces the burden of large attention matrices by leveraging hierarchical representations. Notably, CheapNet's memory usage remained stable even with larger batch sizes, demonstrating its scalability for protein-ligand binding predictions.

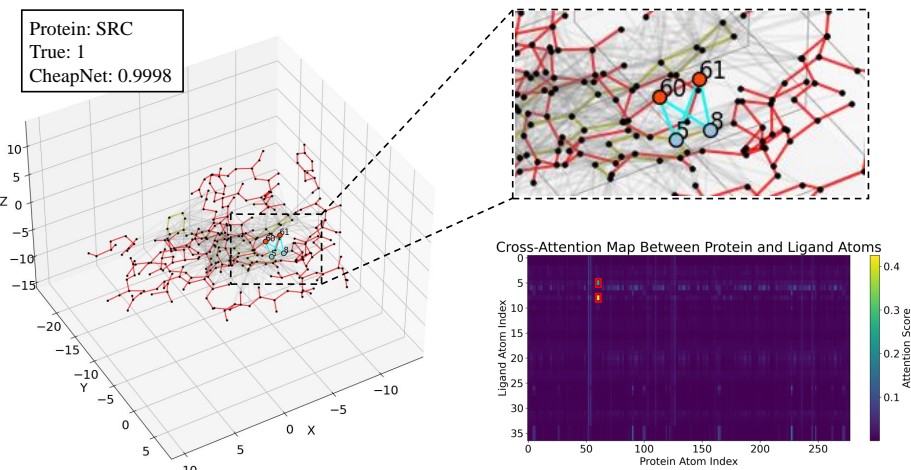

Figure A2: **Visualization of CheapNet's applicability in a active ligand-SRC complex in DUD-E dataset.** The most attended pairs of protein and ligand atoms are highlighted, with cyan lines representing interactions. An object connected with yellow lines is ligand, the other with red is protein, This figure demonstrates CheapNet's effectiveness in capturing key binding regions which correspond to cross-attention maps.

## A.17 FURTHER VISUALIZATION OF CHEAPNET'S INTERPRETABILITY

We further explored other cases that show CheapNet's ability of interpretability at Figure A3. To fully utilize CheapNet's soft assignment and cross-attention mechanism, we summarize the attention scores across both ligand and protein clusters. The attention score between ligand and protein atoms is computed by considering the cross-attention between ligand-to-protein ($Q_{l2p}$, $K_{l2p}$) and protein-to-ligand ($Q_{p2l}$, $K_{p2l}$) attention scores, along with the assignment matrices for ligand ($S_l$) and protein ($S_p$) atoms. The overall attention score can be expressed as:

$$\mathbf{A} = \mathbf{S}_l \left( \text{softmax} \left( \frac{\boldsymbol{Q}_{l2p} \boldsymbol{K}_{l2p}^\top}{\sqrt{d}} \right) + \text{softmax} \left( \frac{\boldsymbol{Q}_{p2l} \boldsymbol{K}_{p2l}^\top}{\sqrt{d}} \right) \right) \mathbf{S}_p^T \tag{17}$$

The visualization result on 4 samples of PDBbind v2016 core set in Figure A3 further illustrate CheapNet's ability to capture meaningful interactions between ligand and protein atoms. Across the cases, the higher attention score regions of cross-attention map indicate key binding regions. CheapNet closely predicts the true binding affinity and performs comparably to GIGN, CheapNet's baseline model. Leveraging the cluster-attention mechanism to identify critical interactions, Cheap-Net achieves higher accuracy to predict protein-ligand binding affinity. These findings demonstrate CheapNet's strength in providing interpretable insights into protein-ligand interactions through its cluster-attention mechanism.

## A.18 SIMULATION ON LACK OF THREE-DIMENSIONAL HIGH-QUALITY DATA

In real-world applications, high-quality data such as three-dimensional crystallized protein structures, as used in this study, may not always be available. In such cases, predicted structures generated by tools like AlphaFold 3 (Abramson et al., 2024) provide a viable alternative. However, these predicted structures often contain noise, which can negatively impact model performance.

To evaluate the robustness of CheapNet's clustering mechanism in handling noisy data, we conducted an ablation study by adding Gaussian noise $\eta \sim \mathcal{N}(0, 1)$, to the initialized atom embeddings in Algorithm 1. This experiment aimed to assess how clustering helps group meaningful atoms while filtering out irrelevant information, mitigating the effects of noise. To isolate the contribution of clustering, we disabled the cross-attention mechanism in CheapNet for this evaluation, focusing solely on hierarchical representations.

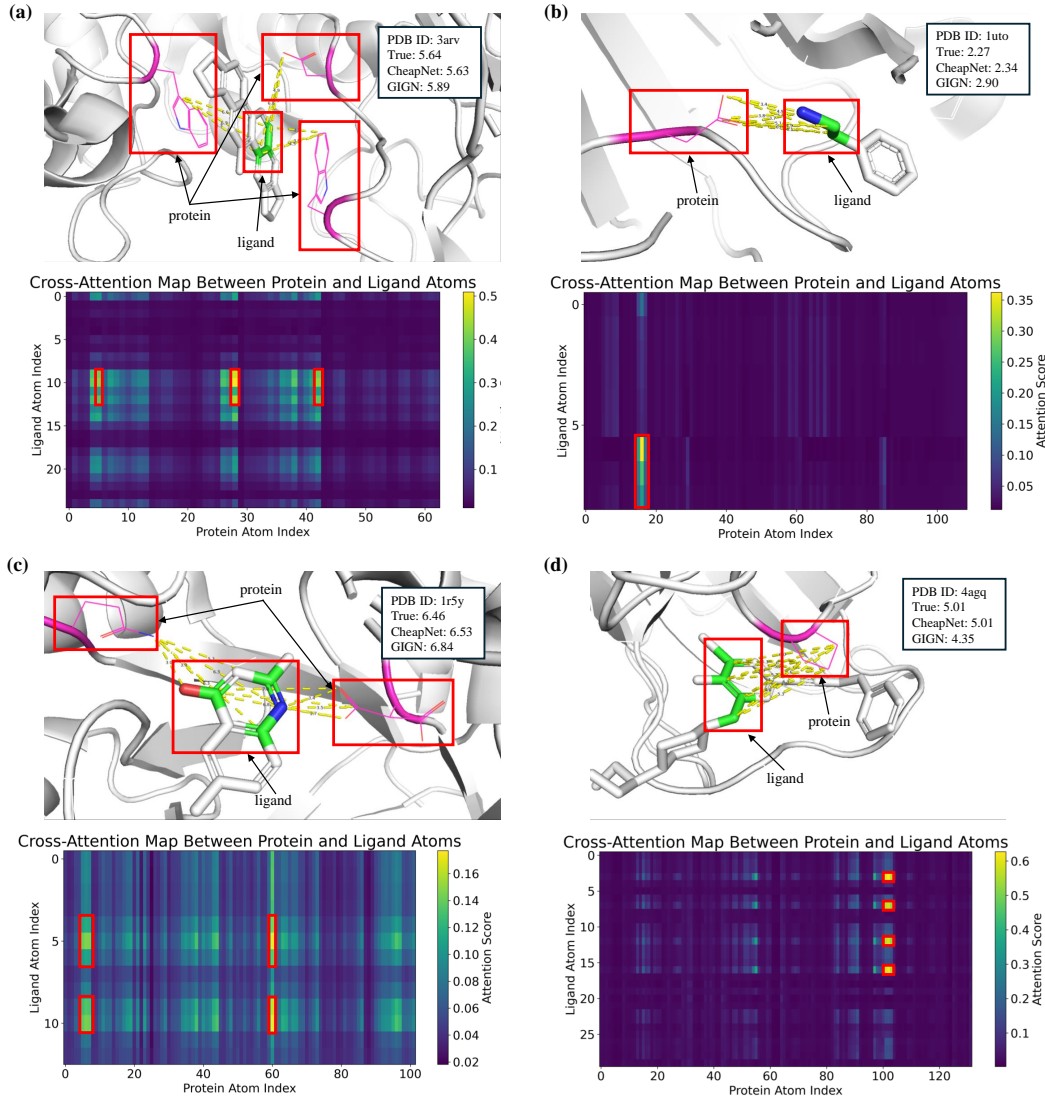

Figure A3: **Visualization of CheapNet's interpretability in various protein-ligand complexes (PDB ID: (a) 3arv, (b) 1uto, (c) 1r5y, (d) 4agq).** The most attended pairs of protein and ligand atoms are highlighted in red boxes, with yellow dashed lines representing interactions. This figure demonstrates CheapNet's effectiveness in capturing key binding regions which correspond to cross-attention maps, offering valuable insights into protein-ligand interactions.

Table A13 summarizes the results of the study. CheapNet, when using hierarchical representations, exhibited less performance decline compared to other models, demonstrating its robustness in noisy environments. While SE(3)-equivariant EGNN exhibited even smaller performance declines under noisy conditions, CheapNet maintained superior overall performance across all datasets. This highlights the value of CheapNet's clustering mechanism for noise reduction and flexible feature extraction, which is especially beneficial for handling noisy predicted structures in practical applications.

Furthermore, inspired by GET (Kong et al., 2024), we evaluated CheapNet's performance under varying levels of coordinate noise to simulate imperfect real-world conditions. As shown in Table A14, CheapNet maintains stable performance under low noise levels but experiences a gradual decline as noise increases, likely due to the GIGN encoder (Yang et al., 2023), which is only

Table A13: Ablation study results for the effect of using hierarchical representations to control additional noise with performance decline ($\Delta$), parameter counts, and standard deviations on the PDBbind v2013 core set, v2016 core set, and v2019 holdout set.

| Model | Noise | Params # | v2013 core set | | v2016 core set | | v2019 holdout set | |
|---|---|---|---|---|---|---|---|---|
| | | | RMSE ↓ | Pearson ↑ | RMSE ↓ | Pearson ↑ | RMSE ↓ | Pearson ↑ |
| GCN (Kipf & Welling, 2016) | ✗ | 0.25M | 1.395 ± 0.033 | 0.819 ± 0.013 | 1.295 ± 0.014 | 0.809 ± 0.004 | 1.460 ± 0.009 | 0.593 ± 0.002 |
| | ✓ | | 1.502 ± 0.039 | 0.773 ± 0.010 | 1.337 ± 0.036 | 0.793 ± 0.014 | 1.510 ± 0.018 | 0.570 ± 0.004 |
| $\Delta$(%) | | | -7.670 | -5.617 | -3.243 | -1.978 | -3.425 | -3.879 |
| IGN (Jiang et al., 2021) | ✗ | 1.66M | 1.428 ± 0.020 | 0.807 ± 0.001 | 1.269 ± 0.030 | 0.821 ± 0.013 | 1.410 ± 0.015 | 0.630 ± 0.008 |
| | ✓ | | 1.474 ± 0.014 | 0.786 ± 0.003 | 1.332 ± 0.020 | 0.795 ± 0.009 | 1.437 ± 0.023 | 0.620 ± 0.008 |
| $\Delta$(%) | | | -3.221 | -2.602 | -4.964 | -3.167 | -1.915 | -1.587 |
| EGNN (Satorras et al., 2021) | ✗ | 1.59M | 1.498 ± 0.025 | 0.782 ± 0.015 | 1.289 ± 0.021 | 0.816 ± 0.011 | 1.399 ± 0.013 | 0.628 ± 0.010 |
| | ✓ | | 1.507 ± 0.006 | 0.780 ± 0.009 | 1.300 ± 0.021 | 0.812 ± 0.015 | 1.402 ± 0.019 | 0.629 ± 0.007 |
| $\Delta$(%) | | | -0.601 | -0.256 | -0.853 | -0.490 | -0.214 | +0.159 |
| CheapNet (w/o cross-attention) | ✗ | 0.81M | 1.330 ± 0.017 | 0.840 ± 0.006 | 1.161 ± 0.012 | 0.853 ± 0.002 | 1.348 ± 0.005 | 0.662 ± 0.005 |
| | ✓ | | 1.352 ± 0.041 | 0.827 ± 0.013 | 1.190 ± 0.025 | 0.842 ± 0.007 | 1.386 ± 0.014 | 0.649 ± 0.008 |
| $\Delta$(%) | | | -1.654 | -1.548 | -2.498 | -1.290 | -2.819 | -1.964 |

Table A14: Ablation study results for the effect of varying error levels to noise coordinate with standard deviations on the LBA 30% dataset of Diverse Protein Evaluation.

| Model | Error Level (Å) | RMSE ↓ | Pearson ↑ | Spearman ↑ |
|---|---|---|---|---|
| CheapNet | 0.0 | 1.311 ± 0.003 | 0.642 ± 0.001 | 0.639 ± 0.010 |
| | 0.1 | 1.325 ± 0.011 | 0.631 ± 0.005 | 0.625 ± 0.003 |
| | 0.5 | 1.340 ± 0.010 | 0.616 ± 0.005 | 0.601 ± 0.005 |
| | 1.0 | 1.348 ± 0.003 | 0.608 ± 0.007 | 0.600 ± 0.007 |
| | 2.0 | 1.364 ± 0.014 | 0.603 ± 0.010 | 0.585 ± 0.009 |
| CheapNet-EGNN | 0.0 | 1.416 ± 0.006 | 0.548 ± 0.007 | 0.532 ± 0.013 |
| | 0.1 | 1.420 ± 0.011 | 0.545 ± 0.011 | 0.542 ± 0.016 |
| | 0.5 | 1.418 ± 0.019 | 0.548 ± 0.020 | 0.544 ± 0.013 |
| | 1.0 | 1.418 ± 0.011 | 0.545 ± 0.013 | 0.530 ± 0.014 |
| | 2.0 | 1.435 ± 0.002 | 0.532 ± 0.004 | 0.522 ± 0.021 |

translation and rotation invariant. To address this, we replaced GIGN with EGNN (Satorras et al., 2021), which is translation-, rotation-, and permutation-equivariant. The resulting model, CheapNet-EGNN, demonstrated more robust performance under higher noise levels. This highlights the modularity of CheapNet's architecture, allowing the GNN encoder to be easily replaced to better suit data quality requirements.

## A.19 EXTENDING 3D INFORMATION TO CLUSTER-LEVEL ATTENTION AND DUAL-AWARENESS FRAMEWORK

While CheapNet currently learns interactions during the graph encoding stage through atom embedding computation, these interactions could be explicitly incorporated in later stages, such as the cross-attention mechanism.

As outlined in Algorithm 2, one potential approach involves pre-computing atom-level edges based on distances between ligand and protein atoms within a threshold (e.g., 5 Å). These edges can then be aggregated into cluster-level weights using the soft clustering assignments of atoms to clusters. The resulting cluster-level weights, representing interaction likelihoods based on atom-level proximity, could serve as biases in the cross-attention mechanism to guide attention scores. This preserves CheapNet's end-to-end differentiability while embedding biologically meaningful priors into interaction modeling.

To explore a dual-awareness framework that utilizes both atom- and cluster-level representations, we conducted experiments with atom selectors such as TopKPooling (Gao & Ji, 2019), which considers individual node embeddings, and ASAPooling (Ranjan et al., 2020), which aggregates local cluster representations. Additionally, we combined TopKPooing or ASAPooling with CheapNet to implement the dual-awareness framework.

Table A15 shows that integrating TopKPooling or ASAPooling with CheapNet (Dual-) improves performance compared to using atom selectors alone. Notably, CheapNet alone achieves the best

---

**Algorithm 2** Pseudo Code for Distance-Driven Cluster Interaction Weighting

**Require:** Ligand atoms $\{la_i\}$, Protein atoms $\{pa_j\}$, Atom positions $\{r_{la_i}, r_{pa_j}\}$, Distance threshold $d_{\text{threshold}}$, Soft cluster assignments $M_{cl,la}$ and $M_{cp,pa}$

**Ensure:** Cluster-level weights $E_{\text{cluster}}^{\text{dist}}(cl, cp)$

1: **Step 1: Compute Atom-Level Distance Edges**
2: **for** each ligand atom $la_i$ and protein atom $pa_j$ **do**
3:    Compute $E_{\text{atom}}^{\text{dist}}(la_i, pa_j)$:

$$E_{\text{atom}}^{\text{dist}}(la_i, pa_j) = \begin{cases} 1 & \text{if } \|r_{la_i} - r_{pa_j}\| \leq d_{\text{threshold}} \\ 0 & \text{otherwise.} \end{cases}$$

4: **end for**
5: **Step 2: Aggregate Atom-Level Edges to Cluster-Level Weights**
6: **for** each ligand cluster $cl$ and protein cluster $cp$ **do**
7:    Compute $E_{\text{cluster}}^{\text{dist}}(cl, cp)$:

$$E_{\text{cluster}}^{\text{dist}}(cl, cp) = \sum_{la_i} \sum_{pa_j} M_{cl,la_i} \cdot M_{cp,pa_j} \cdot E_{\text{atom}}^{\text{dist}}(la_i, pa_j)$$

8: **end for**
9: **Step 3: Integrate into Cross-Attention Mechanism**
10: Modify cross-attention computation as:

$$\text{Attention}(Q, K, V) = \text{softmax}\left(\frac{QK^T + \alpha E_{\text{cluster}}^{\text{dist}}}{\sqrt{d}}\right) V$$

---

Table A15: Results for the dual-awareness framework of CheapNet with parameter counts and standard deviations on the LBA 30% dataset of Diverse Protein Evaluation.

| Model | Params # | RMSE ↓ | Pearson ↑ | Spearman ↑ |
|---|---|---|---|---|
| TopKPooling (Gao & Ji, 2019) | 1.03M | $1.478 \pm 0.048$ | $0.578 \pm 0.013$ | $0.574 \pm 0.030$ |
| ASAPooling (Ranjan et al., 2020) | 1.16M | $1.419 \pm 0.040$ | $0.592 \pm 0.017$ | $0.594 \pm 0.020$ |
| Dual-TopKPooling (Gao & Ji, 2019) | 1.46M | $1.417 \pm 0.007$ | $0.589 \pm 0.012$ | $0.587 \pm 0.010$ |
| Dual-ASAPooling (Ranjan et al., 2020) | 1.59M | $1.394 \pm 0.032$ | $0.618 \pm 0.013$ | $0.619 \pm 0.017$ |
| CheapNet (ours) | 1.39M | $1.311 \pm 0.003$ | $0.642 \pm 0.001$ | $0.639 \pm 0.010$ |

Table A16: Pearson Correlation Results with parameter counts and standard deviations on the Protein-Protein Affinity Prediction. The top results are shown in **bold**, and the second-best are underlined, respectively.

| Model | Params # | Rigid ↑ | Medium ↑ | Flexible ↑ | All ↑ |
|---|---|---|---|---|---|
| SchNet (Gasteiger et al., 2021) | 0.37M | $0.542 \pm 0.028$ | $0.507 \pm 0.020$ | $0.098 \pm 0.011$ | $0.438 \pm 0.017$ |
| GemNet (Gasteiger et al., 2021) | 2.64M | | **OOM** | | |
| TorchMD-NET (Thölke & De Fabritiis, 2022) | 1.00M | $0.572 \pm 0.051$ | $0.498 \pm 0.025$ | $0.109 \pm 0.093$ | $0.438 \pm 0.026$ |
| MACE (Batatia et al., 2022) | 25.7M | $0.616 \pm 0.069$ | $0.461 \pm 0.050$ | $0.275 \pm 0.032$ | $0.466 \pm 0.020$ |
| Equiformer (Liao & Smidt, 2022) | 1.10M | | **OOM** | | |
| LEFTNet (Du et al., 2024) | 3.10M | $0.533 \pm 0.059$ | $0.494 \pm 0.026$ | $0.165 \pm 0.031$ | $0.445 \pm 0.024$ |
| GET (Kong et al., 2024) | 2.50M | $\underline{0.670 \pm 0.017}$ | $\underline{0.512 \pm 0.010}$ | $\underline{0.381 \pm 0.014}$ | $\underline{0.514 \pm 0.011}$ |
| CheapNet (ours) | 2.72M | $\mathbf{0.680 \pm 0.016}$ | $\mathbf{0.518 \pm 0.008}$ | $\mathbf{0.392 \pm 0.004}$ | $\mathbf{0.529 \pm 0.002}$ |

overall results, but the dual-awareness framework demonstrates promising potential for future work.

Table A17: Spearman Correlation Results with parameter counts and standard deviations on the Protein-Protein Affinity Prediction. The top results are shown in **bold**, and the second-best are underlined, respectively.

| Model | Params # | Rigid ↑ | Medium ↑ | Flexible ↑ | All ↑ |
|---|---|---|---|---|---|
| SchNet (Gasteiger et al., 2021) | 0.37M | $0.476 \pm 0.017$ | $0.523 \pm 0.014$ | $0.072 \pm 0.021$ | $0.424 \pm 0.016$ |
| GemNet (Gasteiger et al., 2021) | 2.64M | | OOM | | |
| TorchMD-NET (Thölke & De Fabritiis, 2022) | 1.00M | $0.547 \pm 0.045$ | $0.516 \pm 0.019$ | $0.100 \pm 0.111$ | $0.438 \pm 0.029$ |
| MACE (Batatia et al., 2022) | 25.7M | $0.580 \pm 0.075$ | $0.476 \pm 0.048$ | $0.282 \pm 0.036$ | $0.470 \pm 0.016$ |
| Equiformer (Liao & Smidt, 2022) | 1.10M | | OOM | | |
| LEFTNet (Du et al., 2024) | 3.10M | $0.476 \pm 0.082$ | $0.494 \pm 0.037$ | $0.151 \pm 0.019$ | $0.446 \pm 0.029$ |
| GET (Kong et al., 2024) | 2.50M | $\underline{0.622 \pm 0.030}$ | $\underline{0.533 \pm 0.014}$ | $\underline{0.363 \pm 0.017}$ | $\underline{0.533 \pm 0.011}$ |
| CheapNet (ours) | 2.72M | $\mathbf{0.640 \pm 0.005}$ | $\mathbf{0.535 \pm 0.008}$ | $\mathbf{0.387 \pm 0.017}$ | $\mathbf{0.542 \pm 0.002}$ |

## A.20 BROADENING THE SCOPE OF CHEAPNET: APPLICATION TO PROTEIN-PROTEIN AFFINITY PREDICTION

Protein-Protein affinity (PPA) prediction is essential for understanding the strength of interactions between proteins, which is important in applications such as drug design, signaling pathway analysis, and disease mechanism studies. To demonstrate CheapNet's generability and flexibility beyond protein-ligand tasks, we extend it to PPA prediction, highlighting its capability to hand more complex interfaces.

**Task.** PPA prediction involves modeling the binding strength between two proteins, which often requires analyzing large and intricate interfaces. This task presents unique challenges compared to protein-ligand interactions due to the complexity and variability of protein-protein binding sites.

**Dataset & Evaluation.** For a fair comparison, we follow the protocol from GET (Kong et al., 2024), using 2,500 training complexes from PDBBind (Wang et al., 2004), split by 30% sequence identity. For evaluation, we use the Protein-Protein Affinity Benchmark Version 2 (Vreven et al., 2015), which categorizes 176 protein-protein complexes into three difficulty levels: Rigid, Medium, and Flexible settings. The Flexible category, in particular, involves substantial conformational changes, making it the most challenging. Evaluation metrics include Pearson and Spearman correlation coefficients, with experiments repeated three times using different random seeds.

**Baselines.** We compare CheapNet against state-of-the-art models, including TorchMD-NET (Thölke & De Fabritiis, 2022), LEFTNet (Du et al., 2024), and GET (Kong et al., 2024), following identical experimental settings for a fair comparison. The results were adopted from GET (Kong et al., 2024), which used the hierarchical methods that integrate atom-level and block-level information.

**Performances.** Table A16 and Table A17 demonstrate that CheapNet consistently outperforms all baselines across all difficulty levels, particularly performing exceptionally well in the most challenging Flexible setting. By integrating the cluster-attention mechanism, CheapNet effectively models meaningful interactions between protein clusters, enabling it to handle the complexity of protein-protein interactions. These results demonstrate CheapNet's ability to deliver competitive performance without extensive hyperparameter tuning, reflecting its robustness and adaptability.

These findings highlight CheapNet's adaptability, demonstrating its ability to address not only protein-ligand tasks but also the more complex challenges of protein-protein affinity prediction. This flexibility suggests that CheapNet can serve as a robust framework for a wide range of interaction-related tasks in computational biology.

## A.21 LIMITATIONS

While CheapNet demonstrates strong performance and efficiency in protein-ligand binding affinity prediction, there are some limitations. First, the cluster-level attention mechanism may not capture all nuances of atom-level interactions, especially for complexes where fine-grained atomic interactions are crucial. Second, although our model achieves lower memory usage, its performance is

dependent on the quality of the differentiable pooling and cross-attention mechanisms, which may require fine-tuning for optimal results across diverse datasets. Lastly, CheapNet's efficiency and scalability have not been extensively tested on extremely large protein-ligand complexes, which could impact its applicability in some real-world scenarios. Future work will aim to address these challenges, potentially by integrating more sophisticated clustering techniques or exploring multi-scale representations.

