# OpenReview forum: "CheapNet: Cross-attention on Hierarchical representations for Efficient protein-ligand binding Affinity Prediction"
_ICLR.cc/2025/Conference — ICLR 2025 Poster_

### Official Review · Reviewer_NPNb · 2024-11-01

**Soundness:** 2
**Presentation:** 3
**Contribution:** 2
**Rating:** 6
**Confidence:** 3

**Summary:**

Summary:  The paper introduces CheapNet, a novel model designed for protein-ligand binding affinity prediction. Focusing on efficiency, CheapNet employs a cross-attention mechanism on hierarchical representations to address limitations of traditional atom-level methods, which often capture noise by treating all atom interactions equally. By integrating differentiable pooling, CheapNet selectively forms clusters of atoms that are relevant to binding interactions, reducing computational complexity and improving accuracy. Experimental results demonstrate CheapNet’s competitive, state-of-the-art performance across multiple datasets.

**Strengths:**

1.The paper is clearly written, the authors focus on a critical issue in drug discovery for which they proposed a solution and demonstrated by well designed experiments, that their proposed solution works well.
2.The main idea of using cluster-based, attention-guided binding predictions is well-motivated and aligned with current needs in efficient, scalable drug discovery models.
3.CheapNet achieves state-of-the-art performance across multiple datasets, showcasing its ability to balance accuracy and computational efficiency effectively.

**Weaknesses:**

1.The application of clustering and cross-attention is not novel for this field, as clustering is used in models like GemNet, Equiformer, and LEFTNet. Although CheapNet integrates these methods, it does not introduce substantial methodological innovations.

2.The paper lacks a discussion of relevant clustering methods and does not provide sufficient analysis of different clustering approaches. This omission makes it difficult to assess the comparative advantages of CheapNet’s differentiable pooling mechanism.

3.The comparison lacks depth with state-of-the-art methods, including GemNet, Equiformer, and LEFTNet, all of which employ unique strategies for interaction prediction that CheapNet could be benchmarked against more thoroughly.

4.Equation 2’s category count (number of clusters) is unclear. It would be beneficial to specify whether it is a predefined constant or dynamically determined based on molecular size or complexity. This information is crucial for assessing how different molecular structures might impact CheapNet’s clustering performance.

5.The authors should review and discuss the relevance of clustering methods used by existing models like LEFTNet, which employs a layered approach to handle structural hierarchies. Additionally, the work should compare or at least mention methods from “Generalist Equivariant Transformer Towards 3D Molecular Interaction Learning” to position CheapNet’s approach among recent advancements.

**Questions:**

See in weakness.

---

> ### Author Response · Authors · 2024-11-21
> **Response to Reviewer NPNb (Part 1/2)**
>
> Thanks for your efforts in reviewing our paper. Here are the responses that we hope to resolve your concerns.
>
> > **Q1**:  The application of clustering and cross-attention is not novel for this field, as clustering is used in models like GemNet, Equiformer, and LEFTNet. Although CheapNet integrates these methods, it does not introduce substantial methodological innovations.
> >
> > **Q3**: The comparison lacks depth with state-of-the-art methods, including GemNet, Equiformer, and LEFTNet, all of which employ unique strategies for interaction prediction that CheapNet could be benchmarked against more thoroughly.
>
> **A1+3**:  We appreciate the reviewer’s observation regarding the use of clustering and cross-attention mechanisms in the field of protein-ligand binding affinity prediction, and acknowledge the contributions of prior works such as GemNet [1], Equiformer [2], LEFTNet [3], and GET [4]. While CheapNet indeed employs clustering and cross-attention, the novelty of our approach lies in **soft clustering of atoms with cross-attention to capture meaningful interactions dynamically**. Specifically, our approach assigns soft clusters to atoms based on their embeddings, which are **not (and should not be) limited by geometric constraints.** Unlike previous methods that rely on domain knowledge or geometric properties for clustering, our method ensures flexible and meaningful atomic grouping.
>
> Additionally, we perform an additional experimental result in the LBA 30% task. As shown in the table below, CheapNet achieves the better performances, which distinguish it from existing approaches.
>
> | Model           | Params # | RMSE ↓             | Pearson ↑          | Spearman ↑         |
> |---------------|--------|-----------------|-----------------|-----------------|
> | GemNet          | 1.37M    |                   | OOM               |                   |
> | Equiformer      | 1.10M    |                   | OOM               |                   |
> | LEFTNet         | 0.85M    | 1.366 ± 0.016 | 0.592 ± 0.014 | 0.580 ± 0.011 |
> | GET         | 0.69M    | 1.327 ± 0.005 | 0.620 ± 0.004 | 0.611 ± 0.003 |
> | CheapNet (ours) | 1.39M    | **1.311 ± 0.003** | **0.642 ± 0.001** | **0.639 ± 0.010** |
>
>
> To further clarify this distinction, we have added a detailed comparison with GemNet, Equiformer, LEFTNet, and GET in Section "Related Works" in the main text and "Additional Explanations of Related Works" in the Appendix A.1 to highlight the unique contributions and effectiveness of our method.
>
> ---
>
> > **Q2**: The paper lacks a discussion of relevant clustering methods and does not provide sufficient analysis of different clustering approaches. This omission makes it difficult to assess the comparative advantages of CheapNet’s differentiable pooling mechanism.
>
> **A2**: We acknowledge that the initial discussion on clustering approaches did not adequately highlight the advantages of CheapNet's cluster-attention mechanism. To address this, we have conducted an additional ablation study evaluating the impact of various clustering methods on model performance. This study compares hard node selection methods, such as TopKPooling [5], and structure-based pooling methods, such as ASAPooling [6] and SAGPooling [7]. On the LBA 30% performance of the Diverse Protein Evaluation benchmark, CheapNet consistently outperforms these models. Unlike hard node selection or structure-based clustering techniques, CheapNet's cluster-attention mechanism emphasizes dynamically clustering by atom embeddings rather than relying solely on geometric proximity or pre-defined substructures. This approach provides a complementary perspective on clustering and contributes to its superior performance.
>
> | Model           | Params # | RMSE ↓             | Pearson ↑          | Spearman  ↑        |
> |---------------|--------|-----------------|-----------------|-----------------|
> | TopKPooling          | 1.03M    |  1.478 ± 0.048   | 0.578 ± 0.013               |  0.574 ± 0.030  |
> | ASAPooling      | 1.16M    |  1.419 ± 0.040  |0.592 ± 0.017              |   0.594 ± 0.020   |
> | SAGPooling         | 1.03M    | 1.514 ± 0.020 | 0.582 ± 0.013 | 0.590 ± 0.007 |
> | CheapNet (ours) | 1.39M    | **1.311 ± 0.003** | **0.642 ± 0.001** | **0.639 ± 0.010** |

---

> > ### Author Response · Authors · 2024-11-21
> > **Response to Reviewer NPNb (Part 2/2)**
> >
> > > **Q4**: Equation 2’s category count (number of clusters) is unclear. It would be beneficial to specify whether it is a predefined constant or dynamically determined based on molecular size or complexity. This information is crucial for assessing how different molecular structures might impact CheapNet’s clustering performance.
> >
> > **A4**:  We appreciate the reviewer’s observation. The category count (number of clusters) in Equation 2 is predefined as a constant, determined through hyperparameter tuning. An ablation study, summarized in Appendix A.10, demonstrates that using the median value of the training set for the number of nodes in each protein and ligand achieves the best balance between overfitting and generalizability in our case. Furthermore, as noted in DiffPool [8], while the category count is a predefined parameter, the soft clustering approach dynamically learns to utilize the appropriate number of clusters through end-to-end training, with some clusters potentially remaining unused based on the assignment matrix. Additionally, we apply a cross-attention mechanism to further refine the clustering process, selectively emphasizing protein-ligand interactions that are most relevant, which enhances the model’s interpretability and performance.
> >
> > ---
> >
> > > **Q5**:  The authors should review and discuss the relevance of clustering methods used by existing models like LEFTNet, which employs a layered approach to handle structural hierarchies. Additionally, the work should compare or at least mention methods from “Generalist Equivariant Transformer Towards 3D Molecular Interaction Learning” to position CheapNet’s approach among recent advancements.
> >
> > **A5**: We thank the reviewer for this valuable suggestion. In the revised manuscript, we have included a discussion of clustering methods used by existing models, such as LEFTNet [3] and “Generalist Equivariant Transformer Towards 3D Molecular Interaction Learning” (GET) [4]. Unlike these models, which rely on predefined building blocks or clusters determined by geometric information, CheapNet employs a soft-clustering mechanism where atom embeddings are dynamically grouped through end-to-end training. This structure allows CheapNet to adapt flexibly to diverse molecular structures, enhancing its applicability across varying datasets. Additionally, we have expanded our related work section to include an in-depth discussion of LEFTNet and GET.
> >
> > ---
> >
> > [1] Gasteiger, J., Becker, F., & Günnemann, S. (2021). Gemnet: Universal directional graph neural networks for molecules. _Advances in Neural Information Processing Systems_, _34_, 6790-6802.
> >
> > [2] Liao, Y. L., & Smidt, T. (2022). Equiformer: Equivariant graph attention transformer for 3d atomistic graphs. _arXiv preprint arXiv:2206.11990_.
> >
> > [3] Du, Y., Wang, L., Feng, D., Wang, G., Ji, S., Gomes, C. P., & Ma, Z. M. (2024). A new perspective on building efficient and expressive 3D equivariant graph neural networks. _Advances in Neural Information Processing Systems_, _36_.
> >
> > [4] Kong, X., Huang, W., & Liu, Y. (2023). Generalist equivariant transformer towards 3d molecular interaction learning. _arXiv preprint arXiv:2306.01474_.
> >
> > [5] Gao, H., & Ji, S. (2019, May). Graph u-nets. In _international conference on machine learning_ (pp. 2083-2092). PMLR.
> >
> > [6] Ranjan, E., Sanyal, S., & Talukdar, P. (2020, April). Asap: Adaptive structure aware pooling for learning hierarchical graph representations. In _Proceedings of the AAAI conference on artificial intelligence_ (Vol. 34, No. 04, pp. 5470-5477).
> >
> > [7] Lee, J., Lee, I., & Kang, J. (2019, May). Self-attention graph pooling. In _International conference on machine learning_ (pp. 3734-3743). pmlr.
> >
> > [8] Ying, Z., You, J., Morris, C., Ren, X., Hamilton, W., & Leskovec, J. (2018). Hierarchical graph representation learning with differentiable pooling. _Advances in neural information processing systems_, _31_.

---

> > > ### Comment · Reviewer_NPNb · 2024-11-23
> > >
> > > Thank you very much for your detailed response. The additional experiments you have conducted have made the paper more comprehensive and further validated the effectiveness of your proposed method, which combines 'soft clustering' with cross-attention for molecular representation. While I still believe that the methodological novelty is somewhat limited, I appreciate its effectiveness and the solid experimental results. Based on this, I am willing to raise my score to a 6. Good luck!

---

> > > > ### Author Response · Authors · 2024-11-23
> > > >
> > > > Dear Reviewer NPNb,
> > > >
> > > > Thank you very much for taking the time to provide such thoughtful and constructive feedback. We are truly grateful for your kind words and are delighted to hear that the additional experiments and revisions have made the paper more comprehensive and validated the effectiveness of our approach.
> > > >
> > > > We understand and appreciate your perspective regarding the methodological novelty and will carefully consider this for future work to further improve our contributions. Your acknowledgment of the solid experimental results and the increase in your score are both deeply encouraging to us.
> > > >
> > > > Thank you once again for your support and best wishes.
> > > >
> > > > Sincerely,
> > > > The authors

---

### Official Review · Reviewer_vcDV · 2024-11-02

**Soundness:** 3
**Presentation:** 2
**Contribution:** 3
**Rating:** 6
**Confidence:** 3

**Summary:**

The manuscript proposes a cross-attention method based on atom clustering for protein-ligand affinity prediction. This approach employs a soft-assignment method to separately cluster atoms in the protein and ligand, followed by a cluster-level attention mechanism to facilitate information exchange between the two. The model demonstrates significantly improved performance over baseline methods, and ablation studies indicate that integrating both clustering and cross-attention mechanisms into existing methods enhances prediction accuracy.

**Strengths:**

The method is innovative. By clustering atoms separately within the protein and ligand, it effectively improves affinity prediction performance.

**Weaknesses:**

1. The method’s details are not clearly explained. For example, how are the numbers of clusters for protein and ligand selected? Additionally, the process for initializing the representations of the protein and ligand is unclear.

2. While the method is interesting, the reasoning behind the clustering and cross-attention mechanisms' positive impact on model performance is not fully explained. Further analysis would provide valuable insights. For instance, although a soft clustering method is applied, this component is not explicitly supervised in the loss function. Why does such a soft clustering approach improve the model's performance?

**Questions:**

1. How the numbers of clusters for protein and ligand selected?

2. Is there any experimental evidence explaining why the clustering method enhances model performance?

---

> ### Author Response · Authors · 2024-11-21
> **Response to Reviewer vcDV (Part 1/2)**
>
> We thank the reviewer very much for the careful reading and comments regarding our work. Please, see below our response to the raised comments/questions.
>
> > **Q1**: The method’s details are not clearly explained. For example, how are the numbers of clusters for protein and ligand selected? Additionally, the process for initializing the representations of the protein and ligand is unclear.
> >
> > **Q3**:  How the numbers of clusters for protein and ligand selected?
>
> **A1+3**: We appreciate the reviewer’s insightful comment and have revised the manuscript to provide a clearer explanation of the method’s details. The number of clusters for both protein and ligand is predefined as a constant, determined through hyperparameter tuning, as detailed in Appendix A.10. Specifically, the median value of the training set for the number of nodes in each protein and ligand was selected, as this achieves a balance between overfitting and generalizability.
>
> Regarding the initialization of protein and ligand representations, we have clarified in the revised manuscript that atom representations are initialized following the methods of GIGN [1] for Cross-Dataset Evaluation (Section 4.1) and Atom3D [2] for Diverse Protein Evaluation (Section 4.1) and Ligand Efficacy Prediction (Section 4.2). Both approaches use one-hot encoding based on atom types (e.g., elements like C, H, O, etc.) to initialize each node's features. Additionally, GIGN considers atomic properties such as the degree of an atom, hybridization, and number of valence electrons, while Atom3D incorporates co-crystallized metals (e.g., Zn, Na, Fe, etc.) for protein representation. Due to limited space, a detailed explanation of the initialization process has been added to Appendix A.2.

---

> > ### Author Response · Authors · 2024-11-21
> > **Response to Reviewer vcDV (Part 2/2)**
> >
> > > **Q2**:  While the method is interesting, the reasoning behind the clustering and cross-attention mechanisms' positive impact on model performance is not fully explained. Further analysis would provide valuable insights. For instance, although a soft clustering method is applied, this component is not explicitly supervised in the loss function. Why does such a soft clustering approach improve the model's performance?
> > >
> > > **Q4**: Is there any experimental evidence explaining why the clustering method enhances model performance?
> >
> > **A2+4**: We thank the reviewer for their thoughtful comment. To address the **concern regarding the reasoning behind the positive impact of the clustering and cross-attention mechanisms**, we have clarified and simplified the relevant experiments presented in Table 5 of the main text. This ablation study demonstrates that both the clustering and cross-attention mechanisms contribute significantly to improving model performance across multiple datasets.
> >
> >
> > | Cluster | Cross-Attention | v2013 Core Set (RMSE ↓ / Pearson ↑) | v2016 Core Set (RMSE ↓ / Pearson ↑) | v2019 Holdout Set (RMSE ↓ / Pearson ↑) |
> > |--------------|-----------|-------------------------------------|-------------------------------------|---------------------------------------|
> > | ✗            | ✗         | 1.345 / 0.844                      | 1.189 / 0.851                      | 1.360 / 0.652                        |
> > | ✗            | ✓      | 1.293 / 0.853                      | 1.151 / 0.857                      | 1.362 / 0.653                        |
> > | ✓            | ✗         | 1.330 / 0.840                      | 1.161 / 0.853                      | 1.348 / 0.662                        |
> > | ✓            | ✓      | **1.262** / **0.857**              | **1.107** / **0.870**              | **1.343** / **0.665**                |
> >
> > The clustering mechanism enables the model to group atoms based on their embeddings, rather than relying solely on geometric proximity or pre-defined substructures. This facilitates the representation of higher-order interactions that are critical for protein-ligand binding. Similarly, the cross-attention mechanism focuses on identifying and refining key interactions between proteins and ligands, improving the representation of their binding dynamics. These mechanisms work synergistically, as shown by the performance improvements when both are applied, as compared to their individual contributions or their absence.
> >
> > We apologize for not clearly **conveying the relevant experiments regarding the supervision of the soft clustering component in the loss function.** As outlined in Appendix A.11, we have already experimented with incorporating auxiliary losses, such as link prediction and entropy regularization, as suggested by DiffPool [3]. The results of these experiments indicate that auxiliary losses do not improve performance.
> >
> > This result likely arises from the fact that the link prediction and entropy regularization losses guide the model to cluster atoms based primarily on geometric proximity, which does not fully align with our goal of dynamically identifying biologically meaningful clusters.
> >
> > We have clarified these results in the revised manuscript to ensure this key aspect of our work is better communicated.
> >
> > ---
> >
> > [1] Yang, Z., Zhong, W., Lv, Q., Dong, T., & Yu-Chian Chen, C. (2023). Geometric interaction graph neural network for predicting protein–ligand binding affinities from 3d structures (gign). _The journal of physical chemistry letters_, _14_(8), 2020-2033.
> >
> > [2] Townshend, R. J., Vögele, M., Suriana, P., Derry, A., Powers, A., Laloudakis, Y., ... & Dror, R. O. (2020). Atom3d: Tasks on molecules in three dimensions. _arXiv preprint arXiv:2012.04035_.
> >
> > [3] Ying, Z., You, J., Morris, C., Ren, X., Hamilton, W., & Leskovec, J. (2018). Hierarchical graph representation learning with differentiable pooling. _Advances in neural information processing systems_, _31_.

---

> > > ### Author Response · Authors · 2024-11-25
> > >
> > > Dear Reviewer vcDV,
> > >
> > > Thank you very much for your insightful comments and feedback. We have uploaded our response to your comments and hope it adequately addresses your concerns.
> > >
> > > If you have any further questions or feedback regarding our response, we would be delighted to discuss them. We are committed to improving our manuscript based on your input and will do our best to respond promptly within the remaining 45 hours of the discussion period ends.
> > >
> > > Best regards,
> > >
> > > Authors

---

> > > > ### Comment · Reviewer_vcDV · 2024-11-26
> > > > **Response**
> > > >
> > > > The response address my concerns. I'll raise my score to 6.

---

> ### Author Response · Authors · 2024-11-26
>
> Dear Reviewer vcDV,
>
> We sincerely thank you for your thoughtful feedback and for acknowledging that our responses addressed your concerns. We truly appreciate your decision to raise the score and are grateful for the opportunity to improve our work based on your valuable insights.
>
>
> Best regards,
>
> The authors

---

### Official Review · Reviewer_2uzU · 2024-11-02

**Soundness:** 3
**Presentation:** 2
**Contribution:** 3
**Rating:** 5
**Confidence:** 4

**Summary:**

The authors developed a novel interaction-based model (called CheapNet) that combines atom-level representations with hierarchical cluster-level interactions using a cross-attention mechanism to predict binding affinity tasks. The authors showed that CheapNet can effectively capture key higher-order molecular representations necessary for accurate binding predictions. They also performed extensive evaluations to show that CheapNet can deliver state-of-the-art performance in various binding affinity prediction tasks while maintaining efficiency in computation.

**Strengths:**

-Used both local and global information to predict the binding affinity. The idea is relatively novel.
-Compared the performance of the proposed approach to that of many baseline approaches.
-Demonstrated the model interpretability.

**Weaknesses:**

-The model performance representation can be further improved, such as using p-values to evaluate whether the proposed approach is significantly better than the baselines? the authors claimed "significantly outperforming all baselines", but there are no metrics to support the conclusion.
-It is not clear whether all the comparisons in the results tables are fair comparisons. For example, all these baselines are based on the same data evaluation strategy as the proposed approach (or the same set of training, validation and test sets) ? If the baseline results are from the original papers, how can we make sure the performance evaluations are fair?

**Questions:**

I am curious whether the author can apply the trained models to predict real-world cases involving known disease-causing proteins. Can the models identify compounds from a large library, such as ZINC250K, that are likely to bind to these proteins? Since the authors have emphasized the model's strong interpretability, I would appreciate seeing how the model’s functions are used to interpret the prediction results in this case.

**Details Of Ethics Concerns:**

Not applicable.

---

> ### Author Response · Authors · 2024-11-21
> **Response to Reviewer 2uzU (Part 1/2)**
>
> We thank the reviewer very much for the careful reading and comments regarding our work. Please, see below our answer to the raised comments/questions.
>
> > **Q1**: The model performance representation can be further improved, such as using p-values to evaluate whether the proposed approach is significantly better than the baselines? the authors claimed "significantly outperforming all baselines", but there are no metrics to support the conclusion.
>
> **A1**: We thank the reviewer for pointing out the need for more rigorous statistical evaluation to support our claim of "significantly outperforming all baselines." We acknowledge that directly calculating p-values using a paired t-test is not feasible because the performance metrics for the baseline models were obtained from previously published papers, where only the average and standard deviation were reported. However, to address this limitation, we have performed a statistical significance analysis using a Z-test. This method uses the average performance, standard deviations, and the number of repetitions for each model to calculate p-values and determine whether the differences between the models are statistically significant.
>
> To address this concern, we performed Z-tests for baseline models and evaluation tasks where the average, standard deviation, and number of repetitions were available. Statistical significance was observed in all tasks where CheapNet outperformed the baselines (please see Table A3, A4, and A5 in Appendix A.6 and A.7). For instance, in the LBA 30% evaluation task (metric: RMSE), the comparison between CheapNet and GET (the second-best model) yielded a p-value of approximately ( $2.01 \times 10^{-6}$ ), which is much smaller than 0.05. Furthermore, the 95% confidence intervals for the RMSE scores—CheapNet: [1.308, 1.314], GET: [1.321, 1.333]—demonstrate that the performance difference is statistically significant. These statistical analyses have been incorporated into the revised manuscript to strengthen our claims.
>
> ---
>
> > **Q2**: It is not clear whether all the comparisons in the results tables are fair comparisons. For example, all these baselines are based on the same data evaluation strategy as the proposed approach (or the same set of training, validation and test sets) ? If the baseline results are from the original papers, how can we make sure the performance evaluations are fair?
>
> **A2**: We thank the reviewer for raising this important concern. In the revised manuscript, we have clarified that all comparisons in the results tables are based on the same data evaluation strategy and adhere to the data splits and evaluation settings originally proposed for each task. Specifically, for each evaluation task, we followed the standard data splits (training, validation, and test sets) and metrics defined in the corresponding literature to ensure consistency.
>
> As noted, the baseline results in the results tables are taken from the original papers. While this approach ensures that the baseline models were evaluated under their intended conditions, it is possible that minor differences in implementation or experimental setups may exist. However, as our experiments strictly follow the same data evaluation strategies and settings proposed for each task, we believe the performance evaluations are fair and comparable.
>
> We have added these clarifications to the revised manuscript to address potential concerns about fairness in the comparisons.

---

> > ### Author Response · Authors · 2024-11-21
> > **Response to Reviewer 2uzU (Part 2/2)**
> >
> > > **Q3**: I am curious whether the author can apply the trained models to predict real-world cases involving known disease-causing proteins. Can the models identify compounds from a large library, such as ZINC250K, that are likely to bind to these proteins? Since the authors have emphasized the model's strong interpretability, I would appreciate seeing how the model’s functions are used to interpret the prediction results in this case.
> >
> > **A3**: We thank the reviewer for their insightful question. It is indeed feasible to apply our trained models to real-world cases involving known disease-causing proteins. To demonstrate this, we conducted a virtual screening task using the well-established DUD-E dataset [1], which includes 11,109 active molecules, 10,987 decoys, and 52 target proteins. We curated the dataset by processing its 3D structures with RDkit, extracting protein pockets for active ligands, and constructing corresponding graphs. Undersampled decoys were generated in equal numbers, resulting in a balanced dataset for evaluation.
> >
> > As shown in the below table, our model, CheapNet, achieved superior performance compared to baselines such as GCN [2], EGNN [3], GIGN [4], and AttentionSiteDTI[5] across metrics including AUROC and EF 0.5%. Details of this experiment are provided in Appendix A.15.
> >
> > | Model              | AUROC ↑          | EF0.5% ↑         |
> > |--------------------|------------------|------------------|
> > | GCN           | 0.677 ± 0.030   | 9.951 ± 0.694   |
> > | EGNN          | 0.770 ± 0.017   | 11.368 ± 5.423  |
> > | GIGN              | 0.780 ± 0.017   | 11.079 ± 5.019  |
> > | AttentionSiteDTI  | 0.820 ± 0.012   | 13.985 ± 7.580  |
> > | CheapNet (ours)                      | **0.826 ± 0.011** | **24.646 ± 10.922** |
> >
> >
> > Based on these results, we believe CheapNet can effectively screen large libraries like ZINC250K for compounds likely to bind to specific disease-causing proteins. Furthermore, with the recent availability of AlphaFold3’s code, it is now possible to generate putative 3D complexes for proteins and ligands lacking structural information, which could further enhance the accuracy of virtual screening in real-world applications.
> >
> > Additionally, the interpretability of CheapNet enables insights into why certain compounds are predicted to bind effectively. For example, the attention maps generated by the model can highlight key interaction regions between the protein and ligand, aiding in the understanding of binding mechanisms.
> >
> > ---
> >
> > [1] Mysinger, M. M., Carchia, M., Irwin, J. J., & Shoichet, B. K. (2012). Directory of useful decoys, enhanced (DUD-E): better ligands and decoys for better benchmarking. _Journal of medicinal chemistry_, _55_(14), 6582-6594.
> >
> > [2] Kipf, T. N., & Welling, M. (2016). Semi-supervised classification with graph convolutional networks. _arXiv preprint arXiv:1609.02907_.
> >
> > [3] Satorras, V. G., Hoogeboom, E., & Welling, M. (2021, July). E (n) equivariant graph neural networks. In _International conference on machine learning_ (pp. 9323-9332). PMLR.
> >
> > [4] Yang, Z., Zhong, W., Lv, Q., Dong, T., & Yu-Chian Chen, C. (2023). Geometric interaction graph neural network for predicting protein–ligand binding affinities from 3d structures (gign). _The journal of physical chemistry letters_, _14_(8), 2020-2033.
> >
> > [5] Yazdani-Jahromi, M., Yousefi, N., Tayebi, A., Kolanthai, E., Neal, C. J., Seal, S., & Garibay, O. O. (2022). AttentionSiteDTI: an interpretable graph-based model for drug-target interaction prediction using NLP sentence-level relation classification. _Briefings in Bioinformatics_, _23_(4), bbac272.

---

> > > ### Comment · Reviewer_2uzU · 2024-11-25
> > >
> > > I appreciate the authors' feedback. However, while the authors claim that 'all comparisons in the results tables are based on the same data evaluation strategy and adhere to the data splits and evaluation settings originally proposed for each task. Specifically, for each evaluation task, we followed the standard data splits (training, validation, and test sets) and metrics defined in the corresponding literature to ensure consistency,' I remain skeptical about whether the proposed study and the baselines actually used the same training, validation, and test sets. I reviewed some of the baselines but I did not find that they provided the specific training, validation, and test sets. Therefore, I am not entirely convinced that the performance comparison is fair. As a result, I stand by my original score.

---

> > > > ### Author Response · Authors · 2024-11-25
> > > > **Response to Performance Comparison Fairness (Part 1/4)**
> > > >
> > > > We sincerely thank the reviewer for their follow-up comment and for highlighting concerns about the fairness of our performance comparisons. We deeply value this feedback and have taken additional steps to address these points thoroughly.
> > > >
> > > > ---
> > > > ## Section 4.1 Ligand Binding Affinity / Cross-data Evaluation (Table 1 & Table A.3)
> > > >
> > > > To ensure fair comparisons across all 19 models, including CheapNet, we adopted the same test datasets as described in GIGN [1]:
> > > > - **PDB v2013 core set** (N=107)
> > > > - **PDB v2016 core set** (N=285)
> > > > - **PDB v2019 holdout set** (N=4366)
> > > >
> > > > As stated in GIGN’s **"Data Set Preparation"** section:
> > > > > "... Three independent external test sets, the PDBbind 2013 core set (N = 107), the 2016 core set (N = 285), and the 2019 holdout set (N = 4366), are used to test the generalization capability of GIGN."
> > > > > "The 2013 and 2016 core sets are two commonly used benchmarks to evaluate the performance of binding affinity prediction. (3,5,10,14,35)"
> > > > > "However, their small sample sizes tend to result in overly optimistic results. (4) Therefore, we collect 4366 samples from PDBbind ver. 2019 that are unavailable in the other four sets as a new external holdout set, mimicking a real temporal split scenario in which binding affinities for newly released structures are predicted by a model trained on past structural data. ..."
> > > >
> > > > ### Training and Validation Details
> > > > 1. **GIGN Protocol (16 Models) (except CAPLA, GAABind, DEAttention DTA)**
> > > >    Following the experimental protocol established in GIGN, 16 of the models were trained and validated using identical data splits:
> > > >    - **Training Set:** 11,904 samples from the PDBbind v2016 general set.
> > > >    - **Validation Set:** 1,000 samples from the PDBbind v2016 general set.
> > > >
> > > >    Among these 16 models:
> > > >    - 14 models (excluding AttentionSiteDTI and CheapNet) were directly reported in GIGN.
> > > >    - GIGN explicitly states in its **"Baselines"** section that:
> > > >      > "All the baselines are implemented using the source code provided by the original papers."
> > > >
> > > > 2. **AttentionSiteDTI**
> > > >    The results for AttentionSiteDTI were reproduced in this study using the provided source code, following the same GIGN protocol for training, validation, and testing.
> > > >
> > > > 3. **CheapNet (Ours)**
> > > >    CheapNet also adheres to the GIGN protocol for training, validation, and testing, ensuring a consistent experimental setup.
> > > >
> > > >
> > > > 4. **CAPLA, GAABind, DEAttentionDTA**
> > > > CAPLA, GAABind, and DEAttentionDTA provided pre-trained model checkpoints, which we used to evaluate their performance on the PDB v2013 and PDB v2016 datasets, following the evaluation protocol employed in GIGN. Each model checkpoint was trained as follows:
> > > >
> > > > 5. **[Check Point] CAPLA [2]**
> > > >     CAPLA utilized the PDBbind v2016 general set and the refined set to make train and validation datasets, respectively:
> > > >    - **CAPLA:** 11,906 training samples (from PDB v2016 general set) / 1,000 validation samples (from PDB v2016 refined set).
> > > >    - CAPLA evaluated their performances on the PDB v2016 core set and the CASF-2013 set (=PDB v2013 core set).
> > > >    - CAPLA explicitly states in its **"2.1 Datasets"** section that:
> > > >    > "The commonly used dataset of protein–ligand binding affinity was derived from the PDBbind database of version 2016 (Liu  _et al._, 2017). This database was usually segmented into three overlapping subsets, namely the general set, the refined set and the core 2016 set."
> > > >    > "Here, we adopted the same manner in Pafnucy (Stepniewska-Dziubinska  _et al._, 2018) to partition the training and validation sets, i.e. 1000 complexes were randomly selected from the refined set to constitute the validation set, and a total of 11 906 complexes remaining in the general set constituted the training set."
> > > >    > "The core 2016 set and the CASF-2013 set were used as two benchmark test sets, and we named them Test2016_290 and Test2013_195, respectively."
> > > >
> > > > 6. **[Check Point] GAABind [3]**
> > > > GAABind used training and validation sets from the **PDBbind v2020 general set**, which comprises a larger number of samples and covers a broader range of protein-ligand conformations compared to PDBbind v2016.
> > > >    - **GAABind:** 16,563 training samples / 1,841 validation samples.
> > > >    -   GAABind evaluated their performances on the CASF2016 (=PDB v2016 core set).
> > > >    - GAABind explicitly states in its **"Dataset"** section that:
> > > >    > "Specifically, we used the general set of PDBbindv2020 for training GAABind, and the core set of PDBbindv2016, also known as CASF2016 [50], for evaluation."
> > > >    > "The remaining complexes were randomly divided into a training set (16563 complexes) and a validation set (1841 complexes) in a 9:1 ratio."
> > > >    > "The test dataset, CASF2016, consists of 285 protein–ligand complexes with high-quality crystal structures and reliable binding affinity measurements."

---

> > > > > ### Author Response · Authors · 2024-11-25
> > > > > **Response to Performance Comparison Fairness (Part 2/4)**
> > > > >
> > > > > 7. **[Check Point] DEAttentionDTA [4]**
> > > > > DEAttentionDTA also used training and validation sets from the **PDBbind v2020 general set**.
> > > > >    - **DEAttentionDTA:** 17,478 training samples / 1,942 validation samples.
> > > > >    -  DEAttentionDTA evaluated their performances on the PDB v2016 core set and the CASF-2013 set (=PDB v2013 core set).
> > > > >    - DEAttentionDTA explicitly states in its **"2.1 Datasets"** section that:
> > > > >    > "We utilized the 2020 version of the PDBbind database, which comprises 19 420 protein–ligand complexes."
> > > > >    > "Additionally, two high-quality datasets, CASF2016 (Su et al. 2019) and CASF2013 (Li et al. 2018), comprising 285 and 196 protein–ligand complexes, were used for validation.'"
> > > > >    >
> > > > >    - We carefully reviewed the provided source code (https://github.com/whatamazing1/DEAttentionDTA) and identified the exact split ratio and the number of samples in the training and validation datasets. Additionally, we clarified that the term 'used for validation' in the referenced manuscript refers to the use of CASF2016 (PDB v2016 core) and CASF2013 (PDB v2013 core) as test datasets, as stated in the above sentence.
> > > > >
> > > > > ---
> > > > >
> > > > > ### [Summary] Test Set Reporting in Table 1 and Table A.3
> > > > > In the revised manuscript, we ensured that comparisons across all models in Table 1 and Table A.3 use the **same test datasets**, namely:
> > > > > - **PDB v2013 core set** (N=107),
> > > > > - **PDB v2016 core set** (N=285), and
> > > > > - **PDB v2019 holdout set** (N=4366) (except CAPLA, GAABind, DEAttentionDTA).
> > > > >
> > > > > These datasets were consistently used to evaluate all 19 models' generalization capabilities. However, differences exist in the training and validation datasets:
> > > > > - **For 16 models (including CheapNet):** Both the training (N=11,904) and validation (N=1,000) sets were drawn from the PDBbind v2016 general set, following the protocol established in GIGN [1].
> > > > > - **For CAPLA, GAABind, and DEAttentionDTA:** These models used pre-trained checkpoints based on training and validation sets derived from the PDBbind v2020 general set (CAPLA: v2016 general+ refined set).
> > > > > The **PDB v2019 holdout set** was excluded for CAPLA, GAABind, and DEAttentionDTA because their respective original papers limited performance evaluations to the PDB v2013 or PDB v2016 datasets. As a result, we report results for these models only on the PDB v2013 and PDB v2016 core sets, maintaining consistency with their original experimental protocols.
> > > > >
> > > > > ---
> > > > >
> > > > > ### Consistency and Fairness in Comparisons
> > > > > We acknowledge that while the test datasets are identical across all models, differences in training and validation datasets (specifically for CAPLA, GAABind, and DEAttentionDTA) could lead to variations in model performance. However, as the evaluation protocols strictly adhere to those defined in the respective original studies and leverage identical test datasets, we believe the performance comparisons remain **transparent and meaningful**.
> > > > >
> > > > > [1] Yang, Z., Zhong, W., Lv, Q., Dong, T., & Yu-Chian Chen, C. (2023). Geometric interaction graph neural network for predicting protein–ligand binding affinities from 3d structures (gign). _The journal of physical chemistry letters_, _14_(8), 2020-2033.
> > > > >
> > > > > [2] Jin, Z., Wu, T., Chen, T., Pan, D., Wang, X., Xie, J., ... & Lyu, Q. (2023). CAPLA: improved prediction of protein–ligand binding affinity by a deep learning approach based on a cross-attention mechanism. _Bioinformatics_, _39_(2), btad049.
> > > > >
> > > > > [3] Tan, H., Wang, Z., & Hu, G. (2024). GAABind: a geometry-aware attention-based network for accurate protein–ligand binding pose and binding affinity prediction. _Briefings in Bioinformatics_, _25_(1), bbad462.
> > > > >
> > > > > [4] Chen, X., Huang, J., Shen, T., Zhang, H., Xu, L., Yang, M., ... & Yan, J. (2024). DEAttentionDTA: Protein-ligand binding affinity prediction based on dynamic embedding and self-attention. _Bioinformatics_, btae319.

---

> ### Author Response · Authors · 2024-11-25
> **Response to Performance Comparison Fairness (Part 3/4)**
>
> ## **Section 4.1 Ligand Binding Affinity  / Diverse Protein evaluation (Table 2 & Table A.4)**
> We carefully reviewed the original papers to ensure consistency in datasets and evaluation protocols.
>
> - **HoloProt [1]**([Section 5.1 "Dataset"]):
> > "The PDBBIND database (version 2019) [Liu et al., 2017] is a collection of the experimentally measured binding affinity data .... "
> > "We split the dataset into training, test and validation splits based on the scaffolds of the corresponding ligands (scaffold), or a 30% and a 60% sequence identity threshold (identity 30%, identity 60%) to limit homologous ligands or proteins appearing in both train and test sets."
> - **Atom3D [2]** ([Section 3.5, "Ligand Binding Affinity - Split"]):
> > "We split protein-ligand complexes such that no protein in the test dataset has more than 30% sequence identity with any protein in the training dataset."
> - **ProNet [3]**([Section 6.3 "Ligand Binding Affinity"]):
> > "we use the dataset curated from PDBbind (Wang et al., 2004;Liu et al., 2015) and experiment settings in Somnath et al. (2021) **(=HoloProt)**. We adopt dataset split with 30% and 60% sequence identity thresholds  ..."
> - **ProFSA [4]**([Section 4.3 "Ligand Binding Affinity Prediction - Experimental Configuration"]):
> > " We are utilizing the well-acknowledged PDBBind dataset(v2019) for the ligand binding affinity (LBA) prediction task, and we follow strict 30% or 60% protein sequence-identity data split and preprocessing procedures from the **Atom3D (Townshend et al., 2022)**."
> - **BindNet [5]**([Section 4.1 & 4.1.1 "Ligand Binding Affinity - Data"]):
> > " We assess the performance of BindNet on two binding affinity prediction related tasks, LBA and LEP, as originally proposed in **Atom3D (Townshend et al., 2020)**."
> > "The dataset is partitioned using a protein sequence identity threshold, resulting in two distinct splits: LBA 30% (with a protein sequence identity threshold of 30%) and LBA 60% (with a protein sequence identity threshold of 60%)."
> - **GET [6]**([Section 4.2 "Comparison to Vanilla Unified Representations - Dataset - Ligand-Binding Affinity (LBA),"]):
> > "we use the LBA dataset and its splits in **Atom3D benchmark (Townshend et  al., 2020)**, where there are 3507, 466, and 490 complexes in the training, the validation, and the test sets."
>
> ---
>
> ### Dataset Consistency
> To ensure dataset consistency, we conducted a thorough review based on the following steps:
>
> 1. **Dependency Mapping**
>    - *ProNet* references the dataset protocol from *HoloProt*.
>    - *ProFSA*, *BindNet*, and *GET* follow the dataset splits established by *Atom3D*.
>
> 2. **Verification of HoloProt and Atom3D Consistency**
>    Using the publicly available datasets from [HoloProt](https://zenodo.org/records/8102783) and [Atom3D](https://zenodo.org/records/4914718), we downloaded and compared the protein-ligand complexes. Our analysis confirmed that the datasets are identical, with the following splits:
> - Sequence identity 30%
>    - **Training Set:** 3,507 samples
>    - **Validation Set:** 466 samples
>    - **Test Set:** 490 samples
> - Sequence identity 60%
>    - **Training Set:** 3,563 samples
>    - **Validation Set:** 448 samples
>    - **Test Set:** 452 samples
> ---
>
> ### Baseline Results
> The baseline results were directly adopted from the following sources:
> - **HoloProt**: DeepDTA, SSA, TAPE, IEConv, MaSIF, Holoprot-Full Surface, Holoprot-Superpixel, ProtTrans
> - **Atom3D**: Atom3D-3DCNN, Atom3D-ENN, Atom3D-GNN,
> - **ProNet**: ProNet-Amino Acid, ProNet-Backbone, ProNet-All-Atom
> - **ProFSA**: EGNN-PLM, ProFSA
> - **BindNet**: DeepAffinity, SMT-DTA, GeoSSL, Uni-Mol, BindNet
> - **GET**: SchNet, GemNet, Equiformer, TorchMD-Net, MACE, LEFTNet,  GET
>
>
> [1] Somnath, V. R., Bunne, C., & Krause, A. (2021). Multi-scale representation learning on proteins. _Advances in Neural Information Processing Systems_, _34_, 25244-25255.
>
> [2] Townshend, R. J. L., Vögele, M., Suriana, P. A., Derry, A., Powers, A., Laloudakis, Y., ... & Dror, R. O. ATOM3D: Tasks on Molecules in Three Dimensions. In _Thirty-fifth Conference on Neural Information Processing Systems Datasets and Benchmarks Track (Round 1)_.
>
> [3] Wang, L., Liu, H., Liu, Y., Kurtin, J., & Ji, S. Learning Hierarchical Protein Representations via Complete 3D Graph Networks. In _The Eleventh International Conference on Learning Representations_.
>
> [4] Gao, B., Jia, Y., Mo, Y., Ni, Y., Ma, W. Y., Ma, Z. M., & Lan, Y. Self-supervised Pocket Pretraining via Protein Fragment-Surroundings Alignment. In _The Twelfth International Conference on Learning Representations_.
>
> [5] Feng, S., Li, M., Jia, Y., Ma, W. Y., & Lan, Y. Protein-ligand binding representation learning from fine-grained interactions. In _The Twelfth International Conference on Learning Representations_.
>
> [6] Kong, X., Huang, W., & Liu, Y. Generalist Equivariant Transformer Towards 3D Molecular Interaction Learning. In _Forty-first International Conference on Machine Learning_.

---

> > ### Author Response · Authors · 2024-11-25
> > **Response to Performance Comparison Fairness (Part 4/4)**
> >
> > ## Section 4.2 Ligand Efficacy Prediction (Table 3 & Table A.5)
> >
> > All 17 models were evaluated using the same training, validation, and test splits defined in the Atom3D benchmark [1]. For instance:
> >
> > - **ProNet [2]** ([Appendix F2]):
> >   >  "We also conduct experiments on additional datasets from **Atom3D (Townshend et al., 2021)**, specifically on Protein Structure Ranking (PSR) and Ligand Efficacy Prediction (LEP) datasets"
> > - **ProFSA [3]** ([Appendix C.5]):
> >   > "The result is shown in Table 11. We follow the similar setting used in **ATOM3D (Townshend et al.,2020)** ."
> >  - **BindNet [4]** ([Section 4.1.2 "Data"]):
> >   > "We follow the split defined in **Atom3D** based on the protein function."
> > - **GET [5]** ([Appendix E, "Dataset"]):
> >   > "We follow the LEP dataset and its splits in the **Atom3D benchmark (Townshend et al., 2020)**,"
> >
> > ---
> >
> > ### Baseline Results
> > The baseline results were directly adopted from the following sources:
> > - **Atom3D**: Atom3D-3DCNN, Atom3D-ENN, Atom3D-GNN
> > - **ProNet**: GVP-GNN, ProNet-Amino Acid, ProNet-Backbone, ProNet-All-Atom
> > - **ProFSA**: ProFSA
> > - **BindNet**: DeepDTA, GeoSSL, Uni-Mol, BindNet
> > - **GET**: SchNet, EGNN, TorchMD-Net, GET
> >
> >
> > This consistent use of Atom3D-defined datasets and splits ensures fair and reliable comparisons across all models.
> >
> > ---
> > [1] Townshend, R. J. L., Vögele, M., Suriana, P. A., Derry, A., Powers, A., Laloudakis, Y., ... & Dror, R. O. ATOM3D: Tasks on Molecules in Three Dimensions. In _Thirty-fifth Conference on Neural Information Processing Systems Datasets and Benchmarks Track (Round 1)_.
> >
> > [2] Wang, L., Liu, H., Liu, Y., Kurtin, J., & Ji, S. Learning Hierarchical Protein Representations via Complete 3D Graph Networks. In _The Eleventh International Conference on Learning Representations_.
> >
> > [3] Gao, B., Jia, Y., Mo, Y., Ni, Y., Ma, W. Y., Ma, Z. M., & Lan, Y. Self-supervised Pocket Pretraining via Protein Fragment-Surroundings Alignment. In _The Twelfth International Conference on Learning Representations_.
> >
> > [4] Feng, S., Li, M., Jia, Y., Ma, W. Y., & Lan, Y. Protein-ligand binding representation learning from fine-grained interactions. In _The Twelfth International Conference on Learning Representations_.
> >
> > [5] Kong, X., Huang, W., & Liu, Y. Generalist Equivariant Transformer Towards 3D Molecular Interaction Learning. In _Forty-first International Conference on Machine Learning_.
> >
> >
> > ----
> > ----
> > We hope this clarification addresses the reviewer's concerns regarding the fairness of the comparisons. Please feel free to let us know if further details or additional clarifications are required.

---

> > > ### Author Response · Authors · 2024-11-27
> > >
> > > **Dear Reviewer 2uzU,**
> > >
> > > Thank you for your thoughtful feedback, which has been invaluable in improving our work. As mentioned in our previous response, we conducted a thorough review of the relevant literature, datasets, and source codes to ensure fairness in our comparisons, adhering to standard data splits and evaluation settings wherever specified.
> > >
> > > With only one day remaining for updates to the manuscript itself, we kindly ask if you could share any further feedback by December 2nd, as the peer review discussion remains open until then. Your insights would greatly help us refine the manuscript further and address any remaining concerns.
> > >
> > > Thank you once again for your time and valuable input.
> > >
> > > **Sincerely,**
> > > The Authors

---

> > > > ### Comment · Reviewer_2uzU · 2024-12-02
> > > >
> > > > Thank you for your efforts. After a thorough review of the full paper and responses, I still maintain my score.

---

> ### Author Response · Authors · 2024-12-02
>
> Dear Reviewer 2uzU,
>
> Thank you for your thorough review of our paper and responses. We deeply appreciate the time and effort you dedicated to providing thoughtful feedback, which has been instrumental in refining and strengthening our work.
>
> Your insights have significantly enhanced the clarity and rigor of our study, and we are grateful for the opportunity to engage with your comments throughout the review process. If there are any remaining areas where further clarification is needed, we would be happy to address them.
>
> Once again, we sincerely thank you for your dedication and engagement.
>
> Sincerely,
> The Authors

---

### Official Review · Reviewer_VFuE · 2024-11-03

**Soundness:** 3
**Presentation:** 3
**Contribution:** 3
**Rating:** 8
**Confidence:** 3

**Summary:**

Predicting protein-ligand binding affinity is essential for drug discovery. Due to the complexity of protein-ligand interactions, traditional prediction models, which mainly rely on the atom-level interactions, are often computational intensive and unable to capture the complex and higher-order interactions. This paper proposes a deep learning-based model, CheapNet, for protein-ligand binding affinity prediction. CheapNet uses a cross-attention mechanism on hierarchical representations to capture intricate molecular interactions while maintaining computational efficiency.

**Strengths:**

1. CheapNet integrates both atom-level and cluster-level representations of protein-ligand complexes, this can significantly enhance the model's ability to predict protein-ligand binding affinity. The idea is novel and meaningful。

2. CheapNet employs the DiffPool method to cluster atoms in both the protein and ligand, reducing complexity while retaining the critical protein-ligand interaction patterns. The use of the cross-attention mechanism between protein and ligand clusters highlights the most relevant inter-molecular interactions, filtering out less impactful interactions and reducing the computational costs.

3. The authors utilized the PDBbind and  CSAR NRC-HiQ datasets to benchmark CheapNet against different types of protein-ligand binding affinity prediction models, They evaluated the model performance on ligand binding affinity and ligand efficacy prediction using different performance metrics. Subsequently, ablation studies were conducted to evaluated the model's effectiveness, focusing on adaptability of cluster-attention, hierarchical representations and attention mechanism, and cluster size. The experiments are comprehensive, and demonstrate CheapNet's superior performance.

**Weaknesses:**

1. CheapNet relies on high-quality three-dimensional structural data. However, many proteins lack experimentally crystallized structures, which limits CheapNet's ability to make predictions for proteins without available three-dimensional structural data.

2. In the section 'Permutation Invariance of Clusters for Cross Attention', the authors demonstrate that CheapNet’s cross-attention mechanism ensures permutation invariance for protein and ligand cluster-level representations. However, in protein-ligand interactions, three types of symmetries—translation, rotation, and permutation—should be considered. In my opinion, discussing whether and how the model achieves rotation and permutation invariance in local coordinates, as well as translation, rotation, and permutation equivariance in global coordinates, is essential. Only focusing on discussing the permutation invariance is insufficient.

**Questions:**

1. Discuss how to deal with the proteins which do not have the experimentally crystallized structures. For instance, combine some AI protein prediction models, or use alternative representations for the proteins without three-dimensional structures

2. Extend the discussion on whether and how CheapNet handle the symmetries of protein-ligand complexes. If CheapNet is not able to address other types of symmetries,  then discuss howthis might impact the model's performance or generalizability, and the further improvement.

---

> ### Author Response · Authors · 2024-11-21
> **Response to Reviewer VFuE**
>
> We thank the reviewer for the examination of our work and the thoughtful comments provided. Kindly find our responses to the raised comments and questions below.
>
> >**Q1**:  CheapNet relies on high-quality three-dimensional structural data. However, many proteins lack experimentally crystallized structures, which limits CheapNet's ability to make predictions for proteins without available three-dimensional structural data.
> >
> >**Q3**:  Discuss how to deal with the proteins which do not have the experimentally crystallized structures. For instance, combine some AI protein prediction models, or use alternative representations for the proteins without three-dimensional structures
>
> **A1+A3**: We thank the reviewer for highlighting this important limitation of our approach. We agree that CheapNet’s reliance on high-quality three-dimensional structural data restricts its applicability to proteins with experimentally crystallized structures. To address this limitation, we propose leveraging recent advances in AI-based protein structure prediction models, such as AlphaFold3, to generate high-confidence 3D structures for proteins. These predicted structures can serve as inputs to CheapNet, enabling predictions for a broader range of proteins.
>
> Additionally, we note that CheapNet’s core mechanism—cluster-attention mechanism—is flexible and not strictly tied to 3D-structure-based encoders. While the model benefits significantly from using 3D structural data, as shown with our encoder, it can also be integrated with encoders that do not require 3D structural information, such as GCN. As demonstrated in Table 4, combining CheapNet’s mechanisms with non-3D encoders still yields performance improvements, highlighting its adaptability to scenarios where 3D structures are unavailable.
>
> For cases where predicted 3D structures are unavailable or unreliable, we could also explore alternative representations of proteins, such as sequence-based embeddings (e.g., ESM3 or ProtT5). These could complement or replace structural data, further expanding the applicability of CheapNet while maintaining its interpretability.
>
> We have added a discussion on these potential extensions and the flexibility of CheapNet’s mechanisms to the revised manuscript to address this limitation and provide directions for future work.
>
> ---
>
> >**Q2**: In the section 'Permutation Invariance of Clusters for Cross Attention', the authors demonstrate that CheapNet’s cross-attention mechanism ensures permutation invariance for protein and ligand cluster-level representations. However, in protein-ligand interactions, three types of symmetries—translation, rotation, and permutation—should be considered. In my opinion, discussing whether and how the model achieves rotation and permutation invariance in local coordinates, as well as translation, rotation, and permutation equivariance in global coordinates, is essential. Only focusing on discussing the permutation invariance is insufficient.
> >
> >**Q4**: Extend the discussion on whether and how CheapNet handle the symmetries of protein-ligand complexes. If CheapNet is not able to address other types of symmetries, then discuss how this might impact the model's performance or generalizability, and the further improvement.
>
> **A2+A4**: We thank the reviewer for this insightful comment and for pointing out the need for a more comprehensive discussion on symmetry properties in protein-ligand interactions. We realize that our explanation in the manuscript may not have fully clarified the different types of symmetries involved.
>
> The permutation invariance addressed in Section 3.4 of CheapNet specifically refers to the invariance of cluster assignments during the clustering process—i.e., the order of clusters does not affect the final representation. However, we understand that the reviewer’s comment pertains to symmetries in 3D space, including rotation, translation, and permutation invariance at the atomic coordinate level.
>
> In our current implementation, these 3D symmetries are addressed at the atom embedding stage through the Geometric Interaction Graph Neural Network (GIGN), which explicitly enforces rotation and translation invariance. These invariance properties propagate through to the subsequent stages of CheapNet. However, the cluster-attention mechanism itself operates on graph representations rather than directly processing 3D coordinates, and as such, it does not explicitly enforce additional symmetries.
>
> We have clarified these distinctions in the revised manuscript and extended the discussion to explore how incorporating SE(3)-equivariant encoders, such as EGNN or SE(3)-Transformer, could further enhance CheapNet’s ability to handle global and local 3D symmetries. This flexibility highlights CheapNet’s adaptability to diverse tasks and datasets.
>
> We sincerely thank the reviewer for raising this point, which has allowed us to better articulate CheapNet’s current capabilities and potential extensions.

---

> > ### Author Response · Authors · 2024-11-25
> >
> > Dear Reviewer VFuE,
> >
> > Thank you very much for your insightful comments and feedback. We have uploaded our response to your comments and hope it adequately addresses your concerns.
> >
> > If you have any further questions or feedback regarding our response, we would be delighted to discuss them. We are committed to improving our manuscript based on your input and will do our best to respond promptly within the remaining 45 hours of the discussion period ends.
> >
> > Best regards,
> >
> > Authors

---

> > > ### Author Response · Authors · 2024-11-28
> > >
> > > **Dear Reviewer VFuE,**
> > >
> > > We hope this message finds you well. Following up on our previous response, we wanted to kindly remind you of the updates we have made in the revised manuscript based on your insightful comments. Specifically, we have:
> > >
> > > - Expanded the discussion on **symmetry handling** in CheapNet, addressing how it handles translation, rotation, and permutation symmetries in both local and global coordinates.
> > > - Clarified the distinctions between the symmetries addressed by the **cluster-attention mechanism** and those provided by the **atom-level encoder**, highlighting the modularity of CheapNet and its potential for further improvements.
> > >
> > > The updated details are now incorporated in the revised manuscript in **Section 3.4** and **Appendix A.3 (p. 19, L1010)**, where we discuss the current capabilities and potential extensions of CheapNet in handling symmetries.
> > >
> > > If there are any additional points or clarifications you wish to discuss, we would be delighted to address them before the discussion period ends. Your insights have been invaluable in improving our work, and we are committed to refining the manuscript based on your feedback.
> > >
> > > Thank you once again for your time and thoughtful input.
> > >
> > > **Best regards,**
> > > *The Authors*

---

> > ### Comment · Reviewer_VFuE · 2024-12-01
> >
> > Dear authors,
> > Thank you for addressing my previous questions and for providing the revised article. After reviewing it, I find the content both mathematically and logically rigorous. The cross-attention clustering method you propose is indeed novel and effectively.
> >
> > I still have one concern regarding your approach. For protein-ligand binding affinity, the main focus is on how the molecule interacts with the protein binding pocket. The attention mechanism you implemented should capable of filtering out irrelevant clusters of the protein, which is beneficial. However, your approach also clusters the atoms of the ligand. Considering that ligands are normally small molecules with relatively few atoms—some even fewer than 20—I am afraid of that some molecular information might be lost during the clustering process. This could potentially limit CheapNet's applicability in real-world scenarios.
> >
> > Could you address this concern further? Specifically, do you think a hybrid approach—treating the ligand at the atomic level while clustering the protein—might yield better results?
> >
> > Best regards.

---

> > > ### Author Response · Authors · 2024-12-02
> > > **Response to Small Molecules (Part 1/2)**
> > >
> > > We sincerely thank the reviewer for raising this insightful concern. To evaluate the potential impact of clustering ligand atoms in CheapNet, we conducted additional experiments focusing on ligands with fewer than 20 atoms and evaluated performance across all test datasets (PDB v2013 core set, PDB v2016 core set, and PDB v2019 holdout set). Specifically, we analyzed CheapNet’s performance on this subset and compared it to GIGN, the atom-level encoder used in our implementation.
> > >
> > > #### **Findings for Small Ligands (Fewer than 20 Atoms)**
> > > The results, summarized below, show that CheapNet outperforms GIGN across all three test datasets for this subset. Notably, CheapNet achieves substantial improvements in both RMSE (lower is better) and Pearson R (higher is better), demonstrating its ability to effectively capture meaningful interactions, even for small ligands:
> > >
> > > | **Dataset fewer than 20 atoms** | **v2013 core set** | **v2016 core set** | **v2019 holdout set** |
> > > |-------------------------|-----------------|-----------------|-----------------|
> > > | **RMSE (↓) Improvement (GIGN-CheapNet)** | +0.256 (+18.20%) | +0.174 (+13.87%) | +0.128 (+8.67%) |
> > > | **Pearson R (↑) Improvement (CheapNet - GIGN)** | +0.075 (+10.06%) | +0.038 (+4.72%) | +0.053 (+9.07%) |
> > >
> > > These results suggest that CheapNet’s **soft-clustering mechanism** dynamically groups ligand atoms based on their embeddings, offering greater flexibility compared to existing methods relying on geometric constraints or pre-defined structures. The **cluster-level cross-attention mechanism** further focuses on critical clusters involved in protein-ligand interactions, improving the model's ability to represent these interactions.
> > >
> > > #### **Visualization for Small Ligands Case**
> > > To further validate CheapNet’s ability to capture key interactions for small ligands, we referred to visualizations included in the revised manuscript. As shown in Figure 4 (main manuscript, PDB ID: 4kz6, ligand length: 15) and Appendix A.3 (b)-(c) (PDB IDs: 1uto and 1r5y, ligand lengths: 9 and 13, respectively), CheapNet effectively identifies biologically meaningful interactions between protein and ligand atoms.
> > >
> > > CheapNet leverages its soft-clustering and cross-attention mechanisms to **compute cluster-level attention scores and maps them back to atom-level scores** using Equation (17) (Appendix A.17). Across the visualized cases, CheapNet consistently identifies high-attention regions (marked in red boxes) corresponding to known binding sites. Additionally, CheapNet demonstrates more precise binding affinity predictions compared to GIGN, further validating its robustness.

---

> > > > ### Author Response · Authors · 2024-12-02
> > > > **Response to Small Molecules (Part 2/2)**
> > > >
> > > > #### **Discussion of Hybrid Approach**
> > > > Building on the reviewer's valuable suggestion, we tested a hybrid approach where ligand atoms are treated at the atom-level while protein atoms are clustered. Similarly, we conducted the comparison with the original CheapNet and the hybrid approach on ligands with fewer than 20 atoms.
> > > >
> > > > The results, summarized below, show that while the hybrid approach performs similarly on the PDB v2019 dataset, CheapNet consistently outperforms it across the PDB v2013 and PDB v2016 datasets:
> > > >
> > > > | Dataset fewer than 20 atoms| **CheapNet RMSE ↓** | **Hybrid Approach RMSE ↓** | **RMSE Difference (Hybrid - CheapNet)** | **CheapNet R ↑** | **Hybrid Approach R ↑** | **R Difference (CheapNet - Hybrid)** |
> > > > |--------------------|---------------------|----------------------------|------------------------------------------|------------------|--------------------------|---------------------------------------|
> > > > | **v2013 core set** | 1.151              | 1.244                     | **+0.093 (+7.44%)**                      | 0.821            | 0.788                   | **+0.032 (+4.10%)**                   |
> > > > | **v2016 core set** | 1.077              | 1.154                     | **+0.076 (+6.60%)**                      | 0.848            | 0.819                   | **+0.028 (+3.48%)**                   |
> > > > | **v2019 holdout set** | 1.348           | 1.352                     | **+0.005 (+0.34%)**                      | 0.637            | 0.636                   | **+0.001 (+0.21%)**                   |
> > > >
> > > > These findings highlight the effectiveness of CheapNet’s soft-clustering mechanism for ligand atoms, which dynamically captures meaningful atomic groupings without relying solely on atomic-level embeddings. This mechanism likely enhances CheapNet’s ability to model complex interactions for datasets with diverse ligand sizes and properties.
> > > >
> > > > #### **Future Directions**
> > > > Building on the reviewer’s valuable suggestion, future work could explore hybrid strategies that combine atom-level and cluster-level embeddings for ligands. Techniques such as size-aware gating networks or dual-awareness mechanisms (aligning with suggestions from Reviewer 4kHp Q3) could further enhance CheapNet’s adaptability across diverse ligand sizes and real-world tasks while maintaining its current strengths.
> > > >
> > > > We deeply appreciate the reviewer’s thoughtful comments and constructive suggestions, which have significantly helped improve our work.

---

> ### Comment · Reviewer_VFuE · 2024-12-02
>
> Dear authors,
>
> Thank you for your efforts on this topic. Your answer has addressed my concerns, and I think soft clustering is a promising approach to handle molecular information. Therefore, I would like to raise my point on your paper.
>
> For future work, I would still recommend to deal with the ligand at the atomistic level, as this may be better suited to real-world applications. Wish you good luck.
>
> Best regards.

---

> > ### Author Response · Authors · 2024-12-02
> >
> > Dear Reviewer VFuE,
> >
> > We are sincerely grateful for your thoughtful feedback, kind words, and for raising your score on our paper. It is deeply rewarding to know that our efforts to address your concerns were effective and that you recognize the promise of the soft clustering approach in handling molecular information.
> >
> > Your recommendation to explore atomistic-level handling for ligands in future work is invaluable, and we genuinely appreciate your insightful suggestion. We fully agree that incorporating such approaches may further enhance CheapNet’s applicability to real-world scenarios, and we are excited to pursue this direction in future research.
> >
> > Thank you once again for your constructive comments and encouragement throughout this process. Your input has greatly contributed to improving the quality and potential impact of our work.
> >
> > Sincerely,
> > The Authors

---

### Official Review · Reviewer_4kHp · 2024-11-04

**Soundness:** 3
**Presentation:** 2
**Contribution:** 2
**Rating:** 5
**Confidence:** 4

**Summary:**

This paper proposes a new solution to the protein-ligand binding problem, namely modeling molecules and proteins at a higher level than the atom (cluster level). This motivation comes from the fact that modeling only at the atomic level can easily lead to computational burden and reduced accuracy. Experiments on the ligand affinity prediction and ligand efficacy prediction tasks demonstrate the effectiveness of proposed method.

**Strengths:**

1. The motivation is reasonable, and it is possible to enhance the generalization of the model by trying to model at levels other than atoms.

2. The proposed method is simple and easy to understand.

3. Code is provided and is executable.

4. The experimental results are satisfactory in terms of both accuracy and computational efficiency.

**Weaknesses:**

1. The writing logic of the article is not smooth, making it less readable. Two examples: (1) In the first paragraph in Introduction, a better presentation would be to first introduce the task, then talk about the wet lab approach and limitations, and finally analyze the challenges of deep learning models in solving this problem. Then, the purpose of the sentence describing DTI is also unclear and can be deleted. (2) Why does line 047 begin with "however"? Didn't you just talk about the limitations of atom-level modeling?

2. The motivation is reasonable, that is, the entire functional group may interact with a certain protein region. However, the pooling method used does not seem to guarantee this. Can the author consider, at least, adding additional loss to ensure that clusters represent the functional group?

3. The significance of hierarchical representation is usually to allow the model to adaptively learn and select features from different information channels. I also agree that some interaction cases come from the entire functional group rather than the atom, but this is not absolute. Therefore, I prefer dual awareness at the atom-level and cluster-level. Although cluster-level representations are derived from atom encoders, this only complies with the strong assumption that individual atoms do not participate in interactions. The framework I suggest is to use atom selectors (such as attention selection or gating algorithms) to filter important atom representations to merge with cluster representations.

4. Let's analyze the title. This paper's greatest contribution seems to be to propose a specialized adaptive attention mechanism for hierarchical representation learning. However, the proposed cross-attention algorithm seems to be only for the cluster level. In addition, if this is the case, what is the difference between the proposed method and directly adopting the cross attention module in [1]?

5. This is a point I would like to discuss with the author. In this paper, the author allows complex input, but only uses this information in atom embedding computation. Perhaps there is a more explicit way to use this information. Since we already know the rough range of where the interaction occurs, why not use it to guide the coefficient of cross attention. For example, assuming that cl:1 and cp:2 are clusters of ligand and protein at the interface, the correlation coefficient between cl:1 and cp:2 should be much higher than the others.


[1] Learning Harmonic Molecular Representations on Riemannian Manifold. ICLR, 2023.

**Questions:**

1. How to determine the number of clusters for ligand and protein?

2. In fact, the proposed method still needs to learn effective atom-level representation to obtain pooling results and cluster representation. What is the advantage in computational efficiency?

3. The method in this paper does not seem to be limited to processing protein and ligand interactions, but can also handle protein-protein related tasks (please correct me if I am wrong). If the authors can perform additional experiments such as protein-protein interaction, protein-protein docking or protein-protein interface prediction, it will further prove the scope of the proposed method.

---

> ### Author Response · Authors · 2024-11-21
> **Response to Reviewer 4kHP (Part 1/4)**
>
> We sincerely thank the reviewer for their thorough examination of our work and for providing such thoughtful and constructive comments. Your feedback has been invaluable in helping us refine and improve the manuscript. Please find our detailed responses to the raised comments and questions below.
>
> >**Q1**: The writing logic of the article is not smooth, making it less readable. Two examples: (1) In the first paragraph in Introduction, a better presentation would be to first introduce the task, then talk about the wet lab approach and limitations, and finally analyze the challenges of deep learning models in solving this problem. Then, the purpose of the sentence describing DTI is also unclear and can be deleted. (2) Why does line 047 begin with "however"? Didn't you just talk about the limitations of atom-level modeling?
>
> **A1**:  We thank the reviewer for their insightful feedback regarding the writing logic of the introduction. In the revised manuscript, we have restructured the introduction to improve its flow and readability as follows:
>
> 1. **Task-Wet Lab-Computational Challenges Structure:** We revised the first paragraph to follow a logical progression by first introducing the task of predicting protein-ligand binding affinity, emphasizing its importance in drug discovery, and then discussing the limitations of wet-lab methods. This is followed by an analysis of the challenges faced by computational approaches, particularly deep learning models, in solving this problem.
> 2.  **Removed the DTI Reference:** The sentence describing drug-target interaction (DTI) prediction has been removed, as it did not directly relate to the focus on binding affinity and may have caused confusion.
> 3.  **Clarified the Transition:** We adjusted the transitions to ensure smooth logical progression. Specifically, the use of "however" in line 047 has been replaced with a more appropriate placement earlier in the introduction, where it transitions from discussing wet-lab limitations to computational challenges. This avoids the inconsistency highlighted by the reviewer and ensures that each paragraph builds coherently on the previous one.
>
> We sincerely thank the reviewer for their constructive suggestions, which have helped us improve the clarity, structure, and logical flow of the introduction.
>
> ----
>
> >**Q2**:  The motivation is reasonable, that is, the entire functional group may interact with a certain protein region. However, the pooling method used does not seem to guarantee this. Can the author consider, at least, adding additional loss to ensure that clusters represent the functional group?
>
> **A2**: We thank the reviewer for this thoughtful comment. We agree that the current pooling mechanism in CheapNet does not explicitly enforce clustering of predefined functional groups. Instead, CheapNet dynamically learns clusters through end-to-end training, guided by the task-specific loss, with the goal of identifying groups of atoms that contribute significantly to binding interactions.
>
> To explore the potential of enforcing clustering, we conducted additional experiments incorporating auxiliary losses, including a link prediction loss and an entropy regularization loss, as described in Appendix A.11. These losses were designed to encourage clustering based on geometric proximity. While these losses provided a marginal improvement on smaller datasets (e.g., the PDBbind v2013 core set), they tended to degrade performance on larger datasets, such as v2016 and v2019. This suggests that clustering atoms solely based on geometric proximity may be less effective compared to CheapNet’s current dynamic clustering approach, particularly when paired with the cross-attention mechanism.
>
> In the revised manuscript, we have toned down claims about functional clustering to better reflect CheapNet’s current capabilities. We have also included a discussion of these experimental results to justify our design choices and highlight potential future work, such as developing auxiliary losses that directly align clusters with functional groups.

---

> > ### Author Response · Authors · 2024-11-21
> > **Response to Reviewer 4kHP (Part 2/4)**
> >
> > >**Q3**:  The significance of hierarchical representation is usually to allow the model to adaptively learn and select features from different information channels. I also agree that some interaction cases come from the entire functional group rather than the atom, but this is not absolute. Therefore, I prefer dual awareness at the atom-level and cluster-level. Although cluster-level representations are derived from atom encoders, this only complies with the strong assumption that individual atoms do not participate in interactions. The framework I suggest is to use atom selectors (such as attention selection or gating algorithms) to filter important atom representations to merge with cluster representations.
> >
> > **A3**:
> > We thank the reviewer for their insightful suggestion regarding dual awareness at both the atom-level and cluster-level. To investigate this, we conducted experiments using atom selectors such as TopKPooling [1] (considering only single node embeddings) and ASAPooling [2] (considering representations of local cluster), which compute node scores to filter important atom representations. Additionally, we combined TopKPooling/ASAPooling with CheapNet to implement a dual-awareness framework, as suggested by the reviewer.
> >
> > The results below show that combining TopKPooling/ASAPooling with CheapNet (dual awareness) achieves better performance than using atom selectors alone (e.g., TopKPooling or ASAPooling). Notably, CheapNet alone still achieves the best overall results, but we acknowledge that the dual-awareness approach shows significant potential. Given the limited revision timeline, we were unable to fully explore and optimize the dual-awareness framework, and we agree that it represents a promising direction for future work.
> >
> > | Model  | Params # | RMSE ↓  | Pearson ↑  | Spearman  ↑  |
> > |---------------|--------|-----------------|-----------------|-----------------|
> > | TopKPooling  | 1.03M  |  1.478 ± 0.046  | 0.578 ± 0.013  |  0.574 ± 0.030  |
> > | ASAPooling  | 1.16M  |  1.419 ± 0.040  |0.592 ± 0.017  |  0.594 ± 0.020  |
> > | (Dual) TopKPooling + CheapNet  | 1.46M  | 1.417 ± 0.007 | 0.589 ± 0.012 | 0.587 ± 0.010 |
> > | (Dual) ASAPooling + CheapNet  | 1.59M  | 1.394 ± 0.032 | 0.618 ± 0.013 | 0.619 ± 0.017 |
> > | CheapNet (ours) | 1.39M  | **1.311 ± 0.003** | **0.642 ± 0.001** | **0.639 ± 0.010** |
> >
> > We have added a discussion of these findings to the revised manuscript and highlighted dual awareness as a potential avenue for further exploration.
> >
> > ----
> >
> > >**Q4**: Let's analyze the title. This paper's greatest contribution seems to be to propose a specialized adaptive attention mechanism for hierarchical representation learning. However, the proposed cross-attention algorithm seems to be only for the cluster level. In addition, if this is the case, what is the difference between the proposed method and directly adopting the cross attention module in [1]?
> > > [1] Learning Harmonic Molecular Representations on Riemannian Manifold. ICLR, 2023.
> >
> > **A4**: We thank the reviewer for their thoughtful analysis of the title and for highlighting the importance of clarifying our contributions in the context of hierarchical representation learning. Our approach aligns with the hierarchical framework described in HERN [3], which involves atom-level message passing, pooling, and subsequent block-level message passing. CheapNet adopts this structure by performing message passing in the GNN encoder, pooling through a differentiable soft-assignment mechanism, and learning protein-ligand interactions at the cluster level via a cross-attention mechanism.
> >
> > We acknowledge that clustering and cross-attention mechanisms have been utilized in prior models, such as GemNet [4], Equiformer [5], LEFTNet [6], GET [7], and HMR [8]. However, CheapNet combines **soft clustering with cross-attention** to represent cluster-level interactions in a dynamic and adaptive manner. Unlike approaches that rely on geometric constraints or domain-specific knowledge for clustering, CheapNet’s differentiable pooling mechanism flexibly assigns atoms to clusters, enabling the cross-attention mechanism to focus on interactions at the cluster level. This combination allows CheapNet to selectively attend to molecular groups contributing to binding interactions, improving both accuracy and computational efficiency.

---

> > > ### Author Response · Authors · 2024-11-21
> > > **Response to Reviewer 4kHP (Part 3/4)**
> > >
> > > >**Q5**:  This is a point I would like to discuss with the author. In this paper, the author allows complex input, but only uses this information in atom embedding computation. Perhaps there is a more explicit way to use this information. Since we already know the rough range of where the interaction occurs, why not use it to guide the coefficient of cross attention. For example, assuming that cl:1 and cp:2 are clusters of ligand and protein at the interface, the correlation coefficient between cl:1 and cp:2 should be much higher than the others.
> > >
> > > **A5**:
> > > We thank the reviewer for this insightful comment and for suggesting the use of protein-ligand distances to guide cross-attention coefficients. While CheapNet currently learns these interactions in the atom embedding computation during the graph encoding stage, we agree that it could be leveraged more explicitly in later stages, such as during the cross-attention mechanism.
> > >
> > > A potential approach involves pre-computing atom-level edges based on distances between ligand and protein atoms within a threshold (e.g., 5 Å). These edges could then be aggregated into cluster-level weights using the soft clustering assignments of atoms to clusters. The resulting cluster-level weights would represent the likelihood of interaction based on atom-level proximity and could be integrated into the cross-attention mechanism as biases to guide attention scores. This approach preserves CheapNet’s end-to-end differentiability while incorporating biologically meaningful priors into the interaction modeling.
> > >
> > > We have added a detailed description of this approach, including the relevant equations, in Appendix A.19 of the revised manuscript.  We hope this addition addresses the reviewer’s suggestion and welcome further discussion on this topic to refine and enhance the model’s design.
> > >
> > > ---
> > >
> > > >**Q6**: How to determine the number of clusters for ligand and protein?
> > >
> > > **A6**: We thank the reviewer for this question. The number of clusters for the ligand and protein is treated as a hyperparameter in CheapNet and is determined through hyperparameter tuning, as detailed in Appendix A.10.  In our experiments, we found that setting the number of clusters to approximately the median number of atoms in the training set for each molecule type (ligand or protein) achieves a good balance between overfitting and generalizability.
> > >
> > > ---
> > >
> > > >**Q7**: In fact, the proposed method still needs to learn effective atom-level representation to obtain pooling results and cluster representation. What is the advantage in computational efficiency?
> > >
> > > **A7**:  We thank the reviewer for this comment. While CheapNet does require learning atom-level representations, its computational efficiency arises from aggregating these representations into a smaller number of clusters via differentiable pooling. This significantly reduces the complexity of subsequent operations, such as cross-attention, which operates at the cluster level rather than on all atom pairs.
> > >
> > > To further demonstrate this efficiency, we refer to the memory footprint analysis in Section 4.5 and Figure 3, which shows that CheapNet maintains consistently low memory usage across varying batch and complex sizes. In comparison, models like GAABind and DEAttentionDTA, which rely on atom-level or residue-to-atom attention, exhibit significantly higher memory consumption. These results highlight CheapNet's scalability and suitability for handling large protein-ligand interactions.

---

> > > > ### Author Response · Authors · 2024-11-21
> > > > **Response to Reviewer 4kHP (Part 4/4)**
> > > >
> > > > >**Q8**: The method in this paper does not seem to be limited to processing protein and ligand interactions, but can also handle protein-protein related tasks (please correct me if I am wrong). If the authors can perform additional experiments such as protein-protein interaction, protein-protein docking or protein-protein interface prediction, it will further prove the scope of the proposed method.
> > > >
> > > > **A8**: We thank the reviewer for their thoughtful comment. You are correct that the proposed method is not inherently limited to processing protein-ligand interactions and could be extended to protein-protein related tasks, such as protein-protein affinity prediction (PPA).
> > > >
> > > > We are currently in the process of securing benchmark datasets (e.g., Protein-Protein Affinity Benchmark Version2 [9]) and establishing experimental settings for PPA tasks. While conducting these experiments within the revision process timeline is challenging, we will make every effort to update the manuscript with results if possible.
> > > >
> > > > ---
> > > >
> > > > [1] Gao, H., & Ji, S. (2019, May). Graph u-nets. In _international conference on machine learning_ (pp. 2083-2092). PMLR.
> > > >
> > > > [2] Ranjan, E., Sanyal, S., & Talukdar, P. (2020, April). Asap: Adaptive structure aware pooling for learning hierarchical graph representations. In _Proceedings of the AAAI conference on artificial intelligence_ (Vol. 34, No. 04, pp. 5470-5477).
> > > >
> > > > [3] Jin, W., Barzilay, R., & Jaakkola, T. (2022). Antibody-antigen docking and design via hierarchical equivariant refinement. arXiv preprint arXiv:2207.06616.
> > > >
> > > > [4] Gasteiger, J., Becker, F., & Günnemann, S. (2021). Gemnet: Universal directional graph neural networks for molecules. _Advances in Neural Information Processing Systems_, _34_, 6790-6802.
> > > >
> > > > [5] Liao, Y. L., & Smidt, T. (2022). Equiformer: Equivariant graph attention transformer for 3d atomistic graphs. _arXiv preprint arXiv:2206.11990_.
> > > >
> > > > [6] Du, Y., Wang, L., Feng, D., Wang, G., Ji, S., Gomes, C. P., & Ma, Z. M. (2024). A new perspective on building efficient and expressive 3D equivariant graph neural networks. _Advances in Neural Information Processing Systems_, _36_.
> > > >
> > > > [7] Kong, X., Huang, W., & Liu, Y. (2023). Generalist equivariant transformer towards 3d molecular interaction learning. _arXiv preprint arXiv:2306.01474_.
> > > >
> > > > [8] Wang, Y., Shen, Y., Chen, S., Wang, L., Ye, F., & Zhou, H. (2023). Learning harmonic molecular representations on Riemannian manifold. _arXiv preprint arXiv:2303.15520_.
> > > >
> > > > [9] Vreven, T., Moal, I. H., Vangone, A., Pierce, B. G., Kastritis, P. L., Torchala, M., ... & Weng, Z. (2015). Updates to the integrated protein–protein interaction benchmarks: docking benchmark version 5 and affinity benchmark version 2. Journal of molecular biology, 427(19), 3031-3041.

---

> > > > > ### Author Response · Authors · 2024-11-24
> > > > > **Follow-Up on A8: Response to Reviewer 4kHP**
> > > > >
> > > > > **A8 (Continued)**:  We thank the reviewer for their insightful comment and are pleased to provide updates on our ongoing experiments for Protein-Protein Affinity (PPA) prediction.
> > > > >
> > > > > To evaluate CheapNet’s applicability to protein-protein interactions, we followed the GET [7] setup using the Protein-Protein Affinity Benchmark Version 2 [9], which contains 176 complexes categorized as Rigid, Medium, or Flexible based on conformational changes during binding. Flexible cases are particularly challenging due to significant structural rearrangements. For training, we followed GET’s protocol, using 2,500 protein-protein complexes from PDBbind [10] with annotated affinities ($K_i$ or $K_d$), split by 30% sequence identity.
> > > > >
> > > > >  **Analysis of Results**
> > > > >
> > > > > The results, shown in the tables below, indicate that CheapNet outperforms all baselines across all difficulty levels, particularly excelling in the most challenging Flexible category. For instance, CheapNet achieves Pearson and Spearman correlations of 0.392 and 0.387 in the Flexible setting, surpassing GET, the prior state-of-the-art. This highlights the effectiveness of CheapNet’s cluster-attention mechanism in capturing complex protein-protein interactions, even under conditions of significant conformational change.
> > > > >
> > > > > In the Rigid and Medium settings, CheapNet also demonstrates competitive or superior performance compared to GET and consistently outperforms other baselines such as MACE [13] and LEFTNet [6]. Notably, models like GemNet [4] and Equiformer [5] encountered out-of-memory issues in this benchmark, emphasizing CheapNet’s scalability and efficiency.
> > > > >
> > > > > **Significance of Findings**
> > > > >
> > > > > These findings demonstrate CheapNet’s adaptability and generalizability to diverse interaction-related tasks, extending beyond protein-ligand binding to protein-protein affinity prediction. Furthermore, the ability to achieve these results without extensive hyperparameter tuning highlights its practicality for real-world applications.
> > > > >
> > > > > These findings are discussed in Section 4.4, “Evaluation on External Benchmarks,” with additional details provided in Appendix A.20.
> > > > >
> > > > > We sincerely thank the reviewer for their suggestion, which has enabled us to demonstrate the broader applicability of our method.
> > > > >
> > > > > ---
> > > > >
> > > > > **[Metric: Pearson Correlation]**
> > > > >
> > > > > | Model    |  Params # | Rigid           | Medium          | Flexible        | All            |
> > > > > |------------|-----------------|-----------------|-----------------|-----------------|----------------|
> > > > > | SchNet [11] | 0.37M   | 0.542 ± 0.028  | 0.507 ± 0.020  | 0.098 ± 0.011  | 0.438 ± 0.017  |
> > > > > | GemNet [4]  |2.64M  | OOM | OOM | OOM | OOM |
> > > > > | TorchMD-NET     [12]  |1.00M  | 0.572 ± 0.051  | 0.498 ± 0.025  | 0.101 ± 0.093  | 0.438 ± 0.026  |
> > > > > | MACE  [13]   |25.7M  | 0.616 ± 0.069  | 0.461 ± 0.050  | 0.275 ± 0.032  | 0.466 ± 0.020  |
> > > > > | Equiformer [5] |1.10M| OOM | OOM | OOM | OOM |
> > > > > | LEFTNet  [6] |3.10M | 0.533 ± 0.059  | 0.494 ± 0.026  | 0.165 ± 0.031  | 0.445 ± 0.024  |
> > > > > | GET [7] |2.50M| 0.670 ± 0.017  | 0.512 ± 0.010  | 0.381 ± 0.014  | 0.514 ± 0.011  |
> > > > > | CheapNet (Ours) |2.72M| **0.680 ± 0.016**  | **0.518 ± 0.008**  | **0.390 ± 0.004**  | **0.529 ± 0.002**  |
> > > > >
> > > > > **[Metric: Spearman Correlation]**
> > > > >
> > > > > | Model    |  Params # | Rigid           | Medium          | Flexible        | All            |
> > > > > |------------|-----------------|-----------------|-----------------|-----------------|----------------|
> > > > > | SchNet [11] | 0.37M   | 0.476 ± 0.017  | 0.523 ± 0.014  | 0.072 ± 0.021  | 0.424 ± 0.016  |
> > > > > | GemNet [4]    |2.64M  |  OOM | OOM | OOM | OOM |
> > > > > | TorchMD-NET     [12]   |1.00M  | 0.547 ± 0.045  | 0.516 ± 0.019  | 0.100 ± 0.111  | 0.438 ± 0.029  |
> > > > > | MACE  [13]     |25.7M  | 0.580 ± 0.075  | 0.476 ± 0.048  | 0.282 ± 0.036  | 0.470 ± 0.016  |
> > > > > | Equiformer [5] |1.10M| OOM | OOM | OOM | OOM |
> > > > > | LEFTNet  [6]  |3.10M | 0.476 ± 0.082  | 0.494 ± 0.037  | 0.151 ± 0.019  | 0.446 ± 0.029  |
> > > > > | GET [7] |2.50M| 0.622 ± 0.030  | 0.533 ± 0.014  | 0.363 ± 0.017  | 0.533 ± 0.011  |
> > > > > | CheapNet (Ours) |2.72M| **0.640 ± 0.005**  |**0.535 ± 0.008**  | **0.387 ± 0.017**  | **0.542 ± 0.002**  |
> > > > >
> > > > > ---
> > > > >
> > > > > [10] Wang, R., Fang, X., Lu, Y., & Wang, S. (2004). The PDBbind database: Collection of binding affinities for protein− ligand complexes with known three-dimensional structures. Journal of medicinal chemistry, 47(12), 2977-2980.
> > > > >
> > > > > [11] Schütt, K., Kindermans, P. J., Sauceda Felix, H. E., Chmiela, S., Tkatchenko, A., & Müller, K. R. (2017). Schnet: A continuous-filter convolutional neural network for modeling quantum interactions. Advances in neural information processing systems, 30.
> > > > >
> > > > > [12] Thölke, P., & De Fabritiis, G. (2022). Torchmd-net: equivariant transformers for neural network based molecular potentials. arXiv preprint arXiv:2202.02541.
> > > > >
> > > > > [13] Batatia, I., Kovacs, D. P., Simm, G., Ortner, C., & Csányi, G. (2022). MACE: Higher order equivariant message passing neural networks for fast and accurate force fields. Advances in Neural Information Processing Systems, 35, 11423-11436.

---

> > > > > > ### Author Response · Authors · 2024-11-25
> > > > > >
> > > > > > Dear Reviewer 4kHp,
> > > > > >
> > > > > > Thank you very much for your insightful comments and feedback. We have uploaded our response to your comments and hope it adequately addresses your concerns.
> > > > > >
> > > > > > If you have any further questions or feedback regarding our response, we would be delighted to discuss them. We are committed to improving our manuscript based on your input and will do our best to respond promptly within the remaining 45 hours of the discussion period ends.
> > > > > >
> > > > > > Best regards,
> > > > > >
> > > > > > Authors

---

> > > > > > > ### Comment · Reviewer_4kHp · 2024-11-26
> > > > > > > **response**
> > > > > > >
> > > > > > > Thanks to the authors for their responses. I think the authors addressed most of my concerns, including the motivation, writing, and potential application extensions. So I will consider improving my score. In addition, authors are encouraged to consider adding all the rebuttal content to revised manuscripts.

---

> > > > > > > > ### Author Response · Authors · 2024-11-26
> > > > > > > >
> > > > > > > > Dear Reviewer 4kHp,
> > > > > > > >
> > > > > > > > We sincerely thank you for your thoughtful feedback and for taking the time to carefully review our responses. We deeply appreciate your acknowledgment of the improvements in the motivation, writing, and potential application extensions. Your suggestions, especially regarding the Introduction, have been invaluable in enhancing the clarity and impact of our work.
> > > > > > > >
> > > > > > > > We will ensure that all the rebuttal content is fully integrated into the revised manuscript, as per your recommendation, to provide a comprehensive and transparent presentation of our work.
> > > > > > > >
> > > > > > > > Thank you once again for your constructive feedback and for considering an improved score. Your insights have greatly contributed to the refinement of our submission.
> > > > > > > >
> > > > > > > > Best regards,
> > > > > > > > The authors

---

### Author Response · Authors · 2024-11-21
**Summary of the revised manuscript**

We sincerely thank the reviewers for their thoughtful feedback and constructive suggestions, which have significantly improved the quality and clarity of our manuscript. Below, we summarize the key revisions made in response to the reviewers' comments:

1. **Clustering and Cross-Attention Mechanisms**
   - Clarified the explanation of Table 5 in Section 4.3.2 to better illustrate the impact of hierarchical representations and cross-attention mechanisms on CheapNet’s performance.
   - Analyzed methodological differences between CheapNet and existing cluster-level approaches, such as LEFTNet and GET, in Section 2.2, and compared their performance on LBA and LEP tasks (Tables 2 and 3).

2. **Handling Symmetries in Protein-Ligand Interactions**
   - Clarified in Section 3.4 that CheapNet’s permutation invariance refers specifically to cluster order, ensuring consistent outputs regardless of cluster ordering.
   - Expanded the discussion to explain how CheapNet handles translation and rotation invariance at the atom embedding stage via GIGN, and discussed the potential benefits of integrating SE(3)-equivariant encoders (e.g., EGNN) for handling 3D symmetries.

3. **Lack of 3D Structural Data and Noise Robustness**
   - Discussed the challenge of limited availability of high-quality 3D structural data and the potential use of predicted structures, such as those from AlphaFold3 (see Appendix A.18 for details).
   - Evaluated CheapNet’s robustness to coordinate noise through experiments, showing its ability to maintain strong performance even with noisy predicted structures (see Tables A13 and A14 in Appendix A.18).

4. **Application to Virtual Screening**
   - Discussed CheapNet’s ability to predict protein-ligand interactions through a virtual screening task using the DUD-E dataset (see Appendix A.15 for details).
   - Conducted a case study on Tyrosine Protein Kinase SRC (SRC) to demonstrate CheapNet’s interpretability and accuracy in identifying critical interaction regions (Figure A2 in Appendix A.15).

5. **Extending 3D Information and Exploring Dual-Awareness**
   - Discussed the potential to incorporate 3D information into later stages, such as the cross-attention mechanism, by aggregating atom-level edges into cluster-level weights to guide attention scores (see Algorithm A2 in Appendix A.19).
   - Explored the idea of a dual-awareness framework combining atom- and cluster-level representations with atom-level selectors like TopKPooling and ASAPooling. Preliminary analysis (Table A.15 in Appendix A.19) highlights this as a promising direction for future work.

6. **Statistical Analysis**
   - Conducted Z-tests to assess the statistical significance of CheapNet’s performance improvements compared to baseline models. Results are summarized in Tables A3, A4, and A5 in Appendices A.6 and A.7, confirming that CheapNet consistently outperforms baseline models across multiple metrics.

7. **Clarifications and Revisions**
   - Refined the **Introduction** to improve the writing logic, clearly presenting the motivation, problem statement, and contributions of the study.
   - Conducted a thorough review of the manuscript to enhance clarity, conciseness, and overall readability, ensuring that all sections align cohesively with the study’s objectives and contributions.

---

Please find the new experiments and key revisions highlighted in **blue** in the revised manuscript.

We hope that the added experiments, along with our detailed point-to-point responses, have addressed the reviewers’ concerns. Should there be any additional questions or points requiring further clarification, we would be more than happy to address them.

Thank you again for your valuable time and thoughtful feedback in reviewing our work.

**Best regards,**
_The authors_

---

### Public Comment · ~yang_zhang28 · 2024-11-26
**Questions Regarding Results and Data Preprocessing**

Dear Author(s),

I am deeply interested in the field of binding affinity prediction and have read your paper with great enthusiasm. Currently, I am attempting to understand and study your paper, but I have a few questions regarding the results and data preprocessing. I would greatly appreciate your insights on the following:

1. Regarding GCN Results: In Table 4 (Ablation Study) of your paper, I noticed that the RMSE results of GCN on PDBbind v2013, v2016, and the v2019 holdout set are reported as 1.419, 1.280, and 1.463, respectively. These results are significantly better than those mentioned in the GIGN [1] paper (1.749, 1.513, 1.763) and even surpass those of EGNN. To my understanding, for the same model, given consistent data and experimental configurations, the results are expected to be comparable. Could you please provide more information on whether any additional processing was applied when using GCN?

2. Regarding Ablation Study Results: In the rebuttal of "Response to Reviewer vcDV (Part 2/2)", you presented ablation study results indicating that CheapNet without cluster and cross-attention achieved RMSEs of 1.345, 1.189, and 1.360, respectively, which outperform recent SOTA method like GIGN (1.380, 1.190, 1.393). Since CheapNet without cluster and cross-attention seems relatively straightforward, could you please share any additional details on whether any additional modules or data features were introduced?

3. Regarding Data Preprocessing: In the rebuttal, you provided details about the test dataset and mentioned that "This database was usually segmented into three overlapping subsets, namely the general set, the refined set, and the core 2016 set." There is an overlap between the general set (training dataset) and the core-set (test dataset).  Could you kindly elaborate on the data preprocessing process?

Thank you for your time, and I apologize for any inconvenience caused by my questions.

Sincerely,

[1] Yang, Z., Zhong, W., Lv, Q., Dong, T., & Yu-Chian Chen, C. (2023). Geometric interaction graph neural network for predicting protein–ligand binding affinities from 3d structures (gign). The journal of physical chemistry letters, 14(8), 2020-2033.

---

> ### Author Response · Authors · 2024-11-26
> **Response to "Questions Regarding Results and Data Preprocessing" (Part 1/2)**
>
> Dear yang zhang,
>
> Thank you for your interest in our work and for taking the time to read our paper. We are delighted to hear that you found it engaging and relevant to your studies in binding affinity prediction. We would be happy to address your questions and provide further clarity on the results and data preprocessing.
>
> ---
>
> > **Q1**: Regarding GCN Results: In Table 4 (Ablation Study) of your paper, I noticed that the RMSE results of GCN on PDBbind v2013, v2016, and the v2019 holdout set are reported as 1.419, 1.280, and 1.463, respectively. These results are significantly better than those mentioned in the GIGN [1] paper (1.749, 1.513, 1.763) and even surpass those of EGNN. To my understanding, for the same model, given consistent data and experimental configurations, the results are expected to be comparable. Could you please provide more information on whether any additional processing was applied when using GCN?
>
> **A1**:  We appreciate the observation and would like to clarify the distinction between the GCN model used in our work and that in GIGN [1].
>
> The GCN model in GIGN is an **interaction-free** method, originally adapted from GraphDTA [2]. It operates on the SMILES graph representation of the ligand and the protein sequence as separate inputs. In contrast, our GCN model processes a protein-ligand complex graph as input, identical to the input structure employed by GIGN's **interaction-based** methods. This key difference in input and modeling approach explains the discrepancy in performance between the two GCN models. Our GCN model leverages the interaction-based representation of the protein-ligand complex, which improves its predictive capability.
>
> While our GCN outperforms EGNN on the PDB v2013 core set, EGNN demonstrates superior performance on the PDB v2019 holdout set, which includes more complex and larger protein-ligand complexes. This suggests that EGNN's SE(3)-equivariance provides an advantage in handling datasets with higher structural variability and complexity. These findings demonstrate the importance of choosing an appropriate GNN architecture based on the dataset's characteristics and task requirements.
>
> ---
> > **Q2**: Regarding Ablation Study Results: In the rebuttal of "Response to Reviewer vcDV (Part 2/2)", you presented ablation study results indicating that CheapNet without cluster and cross-attention achieved RMSEs of 1.345, 1.189, and 1.360, respectively, which outperform recent baselines like GIGN (1.380, 1.190, 1.393). Since CheapNet without cluster and cross-attention seems relatively straightforward, could you please share any additional details on whether any additional modules or data features were introduced?
>
> **A2**: We thank the question and for highlighting the importance of providing additional details regarding the ablation study.
>
> CheapNet without cluster-level representations and cross-attention indeed resembles the structure of GIGN, as both operate on atom-level interactions. However, our implementation includes some modifications to the GNN encoder of GIGN to enhance its performance:
>
> 1. **Modified Nonlinear and Normalization Layers:**
>    - In GIGN, the order of the nonlinear and normalization layers is Dropout-LeakyReLU-BatchNorm.
>    - In our implementation, we adjusted this order to BatchNorm-Mish-Dropout. This modification leverages the Mish activation function, which has been shown to improve gradient flow and representation learning in GNNs.
>
> 2. **Incorporating Residual Connections:**
>    - We added a residual connection to the message-passing function of GIGN. This change helps preserve the node's original information across layers, mitigating potential oversmoothing and improving information propagation.
>
> No additional data features were used in our implementation; the input features in the ablation study are consistent with those employed by GIGN.
>
> These modifications likely explain why CheapNet, even without cluster and cross-attention mechanisms, achieves results that are slightly better than GIGN while maintaining a similar overall framework.

---

> > ### Author Response · Authors · 2024-11-26
> > **Response to "Questions Regarding Results and Data Preprocessing" (Part 2/2)**
> >
> > > **Q3**: Regarding Data Preprocessing: In the rebuttal, you provided details about the test dataset and mentioned that "This database was usually segmented into three overlapping subsets, namely the general set, the refined set, and the core 2016 set." Could you kindly elaborate on the data preprocessing process?
> >
> > **A3**: We would like to clarify the context of the statement, *"This database was usually segmented into three overlapping subsets, namely the general set, the refined set, and the core 2016 set."* This particular statement is not from our manuscript, but is instead directly cited from the CAPLA [3] paper. Below, we provide further elaboration on the data segmentation and preprocessing process, drawing from CAPLA.
> >
> > > [From Section 2.1 "Datasets" in CAPLA paper]
> > > "The commonly used dataset of protein–ligand binding affinity was derived from the PDBbind database of version 2016 (Liu  _et al._, 2017). This database was usually segmented into three overlapping subsets, namely the general set, the refined set and the core 2016 set. Specifically, the general set contains all available data, and now a total of 13 285 protein–ligand complexes are included. The refined set is a subset of the general set, which contains 4057 high-quality complexes in total. The core 2016 set comprises 290 complexes by carefully selecting from the refined set, and this set is usually designed as a high-quality benchmark for evaluating protein–ligand binding affinity prediction methods."
> >
> > 1. **Segmentation of PDBBind Data**:
> >    As detailed in the Section 2.1 "Datasets" of CAPLA [3] paper, the authors of CAPLA considered the PDBBind v2016 dataset into three overlapping subsets:
> >    - **General Set**: Includes all available data of PDBBind v2016, totaling 13,285 protein–ligand complexes.
> >    - **Refined Set**: A high-quality subset of the general set, containing 4,057 complexes.
> >    - **Core Set**: A carefully curated benchmark subset of 290 complexes, selected from the refined set. This subset is commonly used to evaluate binding affinity prediction methods.
> >
> > 2. **Core Set Details**:
> >    While CAPLA [3] utilized 290 complexes as the PDB v2016 core set (details provided in Supplementary Table S1), our study followed the GIGN [1] protocol and used the PDBbind database’s CASF-2016 benchmark set, which contains 285 complexes (subset of the 290 complexes), as the test data.
> >
> > ---
> >
> > We hope this detailed clarification addresses your question. Please let us know if further elaboration or additional information is needed.
> >
> > Best regards,
> > The authors
> >
> > ----
> >
> > [1] Yang, Z., Zhong, W., Lv, Q., Dong, T., & Yu-Chian Chen, C. (2023). Geometric interaction graph neural network for predicting protein–ligand binding affinities from 3d structures (gign). _The journal of physical chemistry letters_, _14_(8), 2020-2033.
> >
> > [2]Nguyen, T., Le, H., Quinn, T. P., Nguyen, T., Le, T. D., & Venkatesh, S. (2021). GraphDTA: predicting drug–target binding affinity with graph neural networks. Bioinformatics, 37(8), 1140-1147.
> >
> > [3] Jin, Z., Wu, T., Chen, T., Pan, D., Wang, X., Xie, J., ... & Lyu, Q. (2023). CAPLA: improved prediction of protein–ligand binding affinity by a deep learning approach based on a cross-attention mechanism. Bioinformatics, 39(2), btad049.

---

> ### Public Comment · ~yang_zhang28 · 2024-11-26
>
> Thank you a lot for your response.

---

> > ### Author Response · Authors · 2024-11-27
> >
> > Thank you for your kind comment. We are glad to hear that our responses addressed your questions. Please do not hesitate to reach out if you have any further inquiries or need additional clarification.
> >
> > Sincerely,
> >
> > The authors

---

### Author Response · Authors · 2024-12-03
**Summary of Discussion Period**

We sincerely thank the reviewers for their constructive feedback, which has greatly improved our work. CheapNet introduces a novel cluster-attention mechanism that uses soft clustering of protein-ligand complexes, combined with cross-attention, to identify biologically meaningful interactions for binding affinity prediction.

During the discussion period, the reviewers **recognized the novelty, motivation, and strong performance of CheapNet** and provided insightful feedback on its methodology, experimental evidence, and potential applications.

---
Below, we summarize how the primary concerns raised were addressed:
1. **Impact of Cluster-Attention Mechanism (Reviewer vcDV)**
   - **Concern**: Clarify why the clustering approach improves performance and provide experimental evidence.
   - **Response**: We clarified the reasoning behind clustering and cross-attention mechanisms through ablation studies (Table 5). **The Soft clustering dynamically identifies biologically meaningful clusters based on atom embeddings, while cross-attention refines key protein-ligand interactions**. To further investigate, we experimented with auxiliary losses (Appendix A.11), such as link prediction and entropy regularization. However, these losses tended to group atoms based on geometric proximity rather than embedding similarity, and therefore did not contribute to performance improvement.
2. **Discussion with Recent Cluster-Level Approaches (Reviewer NPNb)**
   - **Concern**: Compare how CheapNet aligns with recent cluster-level approaches, such as LEFTNet and GET, both methodologically and empirically.
   - **Response**: CheapNet’s novelty lies in its **soft clustering of atoms** combined with **cross-attention**, which dynamically captures meaningful interactions without being limited by geometric constraints. This flexibility allows CheapNet to group atoms based on embeddings, distinguishing it from methods relying on predefined geometric or domain-specific knowledge. As shown in the LBA 30% results (Table 2), CheapNet achieves **better performance**, demonstrating its effectiveness in the protein-binding affinity prediction.
3. **Broader Applicability Across Interaction-Related Tasks (Reviewer 4kHp, 2uzU)**
   - **Concern**: Explore CheapNet’s applicability to more diverse tasks (e.g., virtual screening) or tasks beyond protein-ligand binding (e.g., protein-protein affinity).
   - **Response**: CheapNet demonstrated **versatility in virtual screening (DUD-E, Appendix A.15) and protein-protein affinity prediction (PPA Benchmark v2, Appendix A.20)**. In both tasks, CheapNet outperformed baselines, achieving higher AUROC and EF 0.5% in virtual screening and excelling in challenging cases like the PPA Flexible category. These results highlight CheapNet’s adaptability across interaction-related tasks.
4. **Fairness in Performance Comparisons (Reviewer 2uzU)**
   - **Concern**: Ensure that all baseline comparisons use consistent training, validation, and test splits.
   - **Response**: We carefully reviewed the baseline methods to confirm adherence to standard data splits and evaluation protocols. In **"Response to Performance Comparison Fairness (Part 1/4~4/4)"**, we provided detailed explanations of the splits used for each model, referencing their respective papers and source code. This information is summarized in Appendix A.5 of the revised manuscript, ensuring transparency and consistency across all comparisons. We are confident that these efforts demonstrate the fairness and reliability of the reported results.
5. **Symmetries in CheapNet (Reviewer VFuE)**
   - **Concern**: Address how CheapNet handles translation, rotation, and permutation symmetries.
   - **Response**: We **clarified that CheapNet’s ability to handle translation, rotation, and permutation invariance** relies on the properties of its atom-level encoder. The cluster-attention mechanism itself operates on graph representations and does not enforce additional symmetries, ensuring modularity and flexibility in adapting to various GNN encoders. To enhance symmetry-awareness, we explored integrating (S)E(3)-equivariant encoders (e.g., EGNN), as outlined in Section 3.4 and Appendix A.3.


**Future Directions**

Building on the reviewers’ insightful comments, future work could explore **hybrid strategies** that integrate atom- and cluster-level embeddings to enhance CheapNet’s flexibility and performance. Additionally,  leveraging predicted 3D structures (e.g., AlphaFold3) could further expand CheapNet’s applicability in real-world scenarios where experimental 3D structures are unavailable.

---

We hope this summary demonstrates that the reviewers’ concerns have been thoroughly addressed, and that the manuscript is now more robust, clear, and complete. We are deeply grateful to the reviewers for their invaluable insights, which greatly improved the quality and impact of this work.

---

### Meta-Review · Area_Chair_o3mN · 2024-12-20

**Metareview:**

The paper introduces a novel cross-attention mechanism for molecular data based on soft-clustering.

Amonth the strenghts of the method reviewers emphasized clear writing, simplicity of the method and consistent improvements across various benchmarks.

One of the key weaknesses of the paper is its limited novelty. Prior works such as GemNet and LEFTNet also utilize clustering in cross-atention. Authors propose a novel learnable clustering, which delivers consistent but modest improvements over the closest method (as shows during the rebuttal phase).

Three reviewers voted for acceptance and two voted to reject the paper.

During the rebuttal, reviewers raised concerns about the reliance on high-quality 3D structural data, the model's handling of rotational and translational symmetries, and comparison to other clustering-based approaches. Most importantly, the authors have conducted ablation studies demonstrating the performance advantages of CheapNet's soft clustering and cross-attention. Given the broad application of such soft clustering to biological and chemical data, it clears the bar for acceptance.

To summarize, the work is well executed though has limited novelty. Given the soundness of the work and the generality of the self-attention mechanism, the paper clears the bar for acceptance. It is my pleasure to recommend accepting the work.

**Additional Comments On Reviewer Discussion:**

Summarized in the meta-review. Beyond conducting ablation studies, the authors addressed comments by discussing the integration of AI-predicted structures like AlphaFold3 and discussing invariance of the proposed attention mechanism.

---

### Decision · Program_Chairs · 2025-01-22

Accept (Poster)